# Transformative or Conservative?
# Conservation laws for ResNets and Transformers

**Sibylle Marcotte** [1]   **Rémi Gribonval** [2]   **Gabriel Peyré** [1,3]

## Abstract

While conservation laws in gradient flow training dynamics are well understood for (mostly shallow) ReLU and linear networks, their study remains largely unexplored for more practical architectures. This paper bridges this gap by deriving and analyzing conservation laws for modern architectures, with a focus on convolutional ResNets and Transformer networks. For this, we first show that basic building blocks such as ReLU (or linear) shallow networks, with or without convolution, have easily expressed conservation laws, and no more than the known ones. In the case of a single attention layer, we also completely describe all conservation laws, and we show that residual blocks have the same conservation laws as the same block without skip connection. We then introduce the notion of conservation laws that depend only on *a subset* of parameters (corresponding e.g. to a pair of consecutive layers, to a residual block, or to an attention layer). We demonstrate that the characterization of such laws can be reduced to the analysis of the corresponding building block in isolation. Finally, we examine how these newly discovered conservation principles, initially established in the continuous gradient flow regime, persist under discrete optimization dynamics, particularly in the context of Stochastic Gradient Descent (SGD).

## 1. Introduction

Understanding the behavior of neural networks during training remains a fundamental challenge in deep learning. A particularly insightful approach to this challenge involves studying conserved functions - quantities that remain invariant throughout the training process. These conserved functions reveal important geometric properties of the training dynamics and serve a dual purpose. First, they provide valuable insights into the implicit bias induced by both the training algorithm and network architecture, by revealing properties that persist from initialization to the final solution (Saxe et al., 2013; Bah et al., 2022; Arora et al., 2018; Tarmoun et al., 2021; Min et al., 2021). Second, they have emerged as crucial tools in theoretical analyses, playing a key role in convergence studies (Du et al., 2018; Arora et al., 2019; Bah et al., 2022; Chizat & Bach, 2020; Ji & Telgarsky, 2019a; Min et al., 2021). Understanding these conservation laws can also be applied to designing new optimization schemes, which no longer preserve these laws but instead enforce them to reach a desired value (e.g., a balanced condition for ReLU networks) in order to potentially accelerate convergence (Saul, 2023; Stock et al., 2019).

**Conservation laws.** In the context of Euclidean gradient flow training dynamics, conservation laws in the form of "balancedness conditions" have been established for ReLU and linear networks (Saxe et al., 2013; Du et al., 2018; Arora et al., 2019). Subsequently, (Marcotte et al., 2023) demonstrated the "completeness" of these laws: no additional conservation laws exist for these architectures in the shallow case under Euclidean gradient flows. For these network architectures, (Marcotte et al., 2024) unveiled novel conservation laws when considering alternative optimization algorithms – particularly non-Euclidean gradient flows, as employed in ICNN or NMF, or momentum-based dynamics, also demonstrating their completeness. Furthermore, (Marcotte et al., 2024) revealed that conservation laws under momentum dynamics exhibit fundamentally different characteristics compared to simple gradient flows: these laws are *time and velocity-dependent*, and are generally fewer in number than in the gradient flow case. For feed-forward networks with single-channel convolutions, conservation laws were also identified under gradient flow dynamics (Du et al., 2018). While the investigation of conservation laws for more sophisticated neural architectures has remained largely unexplored, this paper addresses this gap by extending the analysis to more complex network architectures.

[1]ENS-PSL Univ. [2]Inria, CNRS, ENS de Lyon, Université Claude Bernard Lyon 1, LIP, UMR 5668, 69342, Lyon cedex 07, France [3]CNRS. Correspondence to: Sibylle Marcotte <sibylle.marcotte@ens.fr>.

*Proceedings of the 42$^d$ International Conference on Machine Learning*, Vancouver, Canada. PMLR 267, 2025.

**Residual networks (ResNets)** constitute a fundamental class of deep learning architectures that revolutionized the field of computer vision through their groundbreaking performance (He et al., 2016). The distinguishing feature of these networks—the incorporation of skip connections—has since become a cornerstone principle in other deep learning architectures, most notably exemplified in Transformer models (Vaswani, 2017). In (Marion et al., 2023), under specific initialization assumptions and incorporating a rescaling operation, the authors demonstrate that the solution reached during traing (i.e. the trained neural network) corresponds to a discretization of a Neural ODE (Chen et al., 2018), thus revealing an implicit bias. This enables leveraging ODE (ordinary differential equation) theory to analyze the trained network.

**Transformers.** Since their introduction (Vaswani, 2017), Transformers and their multi-head attention mechanism (Bahdanau et al., 2015) have achieved unprecedented performance across domains from natural language processing (Brown et al., 2020) to computer vision (Dosovitskiy et al., 2021). In (Vasudeva et al., 2024), the authors shows that when training self-attention layers with gradient descent, the key-query matrix naturally converges to a hard-margin SVM solution, revealing an implicit bias similar to that observed in linear logistic regression on separable data (Soudry et al., 2018; Ji & Telgarsky, 2019b).

**Our contributions.** After proving that conservation laws of gradient flows *with weight decay* are determined by their equivalent without weight decay (Theorem 2.1), we discover new conservation laws and show that these new laws are complete for several basic building blocks of modern networks with skip connections: shallow multi-channel convolution layers (Theorem 3.6), self-attention layers (Corollary 3.9, Corollary 3.10), cross-entropy classification layer (Proposition 3.11), and plain MLP with skip connections (Proposition 3.2). We then explain how these results can be used to analyze deeper networks, via a generic analysis under the lens of the new taylored analysis of conservation laws that only depend on a given subset of parameters (Proposition 4.3). Notable results (Theorem 4.6) include the formal proof that such laws exactly match the laws of the classical blocks considered in isolation. Besides, in the case of residual convolutional networks we show (Theorem 4.7) the absence of conservation law associated to consecutive linear layers that "overlap" two residual blocks. To complete the theoretical analysis we finally show that the conservation laws of gradient flows are also approximately preserved during actual (discrete) SGD dynamics, under appropriate assumptions (Proposition 5.1), with an error bound (27) scaling as $O(\text{step-size}^2)$ that we showcase in our numerical experiments, Figure 1 and Figure 5.

## 2. Conservation laws for Gradient Flows

In this paper, we consider learning problems where the features are represented as $x_i \in \mathbb{R}^m$ and the targets or labels are denoted by $y_i \in \mathcal{Y}$. In regression tasks, $\mathcal{Y}$ is typically defined as $\mathbb{R}^n$, while in classification contexts, $y_i$ represents categorical labels. In scenarios involving unsupervised or self-supervised learning, $y_i$ can be treated as a constant. We denote $z_i \coloneqq (x_i, y_i)$ and $Z = (x_i, y_i)_i$.

Predictions are generated through a parametric function $g(\theta, \cdot) : \mathbb{R}^m \to \mathbb{R}^n$ (such as a neural network). This function is trained by empirically minimizing a **cost** function with respect to the parameters $\theta \in \Theta \subseteq \mathbb{R}^D$.

$$L_Z(\theta) \coloneqq \frac{1}{N} \sum_i \ell(g(\theta, x_i), y_i), \tag{1}$$

with $\ell$ a **loss** function. In practice, the training dynamics are realized using a gradient descent algorithm with weight decay:

$$\theta_{k+1} = \theta_k - \tau \nabla L_Z(\theta_k) - \lambda_k \tau \theta_k. \tag{2}$$

To facilitate mathematical analysis, the training dynamics is simplified by considering a gradient flow (**GF**) with weight decay (**WD**). This approach represents the continuous counterpart of Equation (2) as $\tau$ approaches zero and can be expressed as the first-order ODE where $\lambda(t) \geq 0$:

$$\dot{\theta}(t) + \underbrace{\lambda(t)\theta(t)}_{\text{weight-decay}} = -\nabla L_Z(\theta(t)), \tag{3}$$

We aim to understand what quantities are preserved during the dynamic (3) for a variety of neural networks $g$. The mathematical study of what happens in discrete dynamics is done in Section 5. The analogue of the transition from (2) to (3) for a simplified version of Adam algorithm is analyzed in Section 2.3.

### 2.1. Conservation laws of neural networks

A function $h(t, \theta)$ is conserved if for each solution $\theta(t)$ of the ODE (3) with any initialization and any data-set, one has $h(t, \theta(t))$ that remains constant over time.

**Time-dependency: with *vs* without weight-decay.** Our first contribution is the following theorem, which clarifies a fundamental distinction in the temporal dependence of conserved quantities of a dynamic system, whether it includes WD or not. It also demonstrates how the conservation laws with WD can be readily derived from those from the system without WD. See Appendix A for a proof.

**Theorem 2.1** (Structure theorem). *Let $h(t, \theta)$ be a conserved function for the ODE (3). If for every $\theta$,*

*there exists a data-set $Z$ such that $\nabla L_Z(\theta) = 0$, then the function $H(a) := h(0, a)$ satisfies $h(t, \theta) = H\big(\theta \exp(\int_0^t \lambda(s)\mathrm{d}s)\big)$, $\forall t, \theta$. Thus, conserved functions can be expressed with $D$ variables (instead of $D+1$). Moreover, $\tilde{h}(t, \theta) := H(\theta)$ is a conservation law of (3) without WD (i.e. with $\lambda(t) \equiv 0$).*

*Remark 2.2.* In particular, when considering (3) with $\lambda(t) \equiv 0$, one has $h(t, \theta) = h(0, \theta)$ for all $t$ and $\theta$.

Given this established correspondence between conserved functions with and without weight-decay, we can now restrict our analysis to time-independent conserved functions $h(\theta)$ in the case of gradient descent without WD:

$$\dot{\theta}(t) = -\nabla L_Z(\theta(t)). \tag{4}$$

**Definition and characterization of conservation laws.** Here we recall the main ingredients of the framework for conservation laws from (Marcotte et al., 2023), introducing some formal definitions of intermediate objects and results that hopefully streamline the corresponding reasoning. A function $h(\theta)$ is a conservation law for $g$ if for each solution $\theta(t)$ of the ODE (4) with any initialization and any data-set, one has $h(\theta(t)) = h(\theta(0))$, $\forall t$. The formal definition of a conservation law in that case is given in (Marcotte et al., 2023, Definition 2.4) (corresponds to the notion of being locally conserved on $\Theta$ for any dataset), and we recall an orthogonal characterization of a smooth conservation law (Marcotte et al., 2023, Corollary 2.6, Proposition 2.7):

**Proposition 2.3.** *Assume that for each $y \in \mathcal{Y}$, the loss $\ell(z, y)$ is $\mathcal{C}^2$-differentiable with respect to $z \in \mathbb{R}^n$. A function $h \in \mathcal{C}^1(\Theta, \mathbb{R})$ is a conservation law for $g$ with respect to the loss $\ell$ if and only if $\nabla h(\theta) \perp \mathcal{W}_\theta^{g,\ell}$ for all $\theta \in \Theta$ where:*

$$
\begin{aligned}
\mathcal{W}_\theta^{g,\ell} &:= \operatorname*{span}_{Z=(x_i, y_i) \in (\mathcal{X}_\theta \times \mathcal{Y})^N} \{\nabla L_Z(\theta)\} \\
&= \operatorname*{span}_{(x,y) \in \mathcal{X}_\theta \times \mathcal{Y}} \{\partial_\theta g(\theta, x)^\top \nabla_z \ell(g(\theta, x), y)\} \subseteq \mathbb{R}^D,
\end{aligned}
$$

*with $\mathcal{X}_\theta$ the set of data points $x \in \mathbb{R}^m$ such that $g(\cdot, x)$ is $\mathcal{C}^2$-differentiable in the neighborhood of $\theta$.*

**Assumption 2.4.** Consider a loss $\ell(z, y)$. We assume that for every $y$ it is differentiable with respect to $z \in \mathbb{R}^n$. We define for all $z \in \mathbb{R}^n$ the subspace

$$\mathcal{V}_\ell(z) := \operatorname{span}_y \nabla_z \ell(z, y) \subseteq \mathbb{R}^n.$$

We also assume that $\mathcal{V}_\ell(z)$ **does not depend** on $z \in \mathbb{R}^n$, so we rewrite $\mathcal{V}_\ell(z) = \mathcal{V}_\ell$.

In particular, this assumption is satisfied for all classical losses (e.g. mean-square error, Kullback-Leibler divergence, cross-entropy loss) as stated in (Marcotte et al., 2023, Lemma C.2, Remark C.3), and is known to imply the following direct corollary:

**Corollary 2.5.** *Consider a loss $\ell(z, y)$ that satisfies Assumption 2.4. Then for all $\theta \in \Theta$:*

$$\mathcal{W}_\theta^{g,\ell} = \operatorname*{span}_{x \in \mathcal{X}_\theta, w \in \mathcal{V}_\ell} \{\partial_\theta g(\theta, x)^\top w\}.$$

Another useful assumption is the following.

**Assumption 2.6** (Local reparameterization)**.** There exists $d$ and $\phi \in \mathcal{C}^2(\Theta, \mathbb{R}^d)$ such that: for each parameter $\theta_0$ in the open set $\Theta \subseteq \mathbb{R}^D$, for each $x \in \mathcal{X}$ such that $\theta \mapsto g(\theta, x)$ is $\mathcal{C}^2$ in a neighborhood of $\theta_0$[1], there is a neighborhood $\Omega$ of $\theta_0$ and $f(\cdot, x) \in \mathcal{C}^2(\phi(\Omega), \mathbb{R}^n)$ such that

$$\forall \theta \in \Omega, \quad g(\theta, x) = f(\phi(\theta), x). \tag{5}$$

Such a factorization $g(\theta, x) = f(\phi(\theta), x)$ is always possible but never unique: $\phi = \mathrm{id}$ and $f = g$ yield a trivial factorization, and starting from an arbitrary factorization any diffeomorphism $\psi$ yields another one $g(\theta, x) = \tilde{f}(\tilde{\phi}(\theta), x)$ with $\tilde{f}(a, x) := f(\psi(a), x)$ and $\tilde{\phi} := \psi^{-1} \circ \phi$. The corresponding space $\mathcal{W}_\theta^{g,\ell}$ can be characterized with any such factorization (Marcotte et al., 2023, Proposition 2.12).

**Proposition 2.7.** *Assume that for every $y$ the loss $\ell(z, y)$ is $\mathcal{C}^2$-differentiable with respect to $z$. Under Assumption 2.6, for all $\theta \in \Theta$:*

$$\mathcal{W}_\theta^{g,\ell} = \partial\phi(\theta)^\top \mathcal{W}_{\phi(\theta)}^{f,\ell} \tag{6}$$

*with $\partial\phi(\theta) \in \mathbb{R}^{d \times D}$ the Jacobian of $\phi$ and*

$$\mathcal{W}_{\phi(\theta)}^{f,\ell} := \operatorname*{span}_{(x,y) \in \mathcal{X}_\theta \times \mathcal{Y}} \{\partial f^x(\phi(\theta))^\top \nabla_z \ell(g(\theta, x), y)\},$$

*where $f^x(\cdot) := f(\cdot, x)$.*

Under Assumption 2.6, Proposition 2.7 directly rewrites as:

**Corollary 2.8.** *Consider a loss $\ell(z, y)$ that satisfies Assumption 2.4. Under Assumption 2.6, then for all $\theta \in \Theta$*

$$\mathcal{W}_{\phi(\theta)}^{f,\ell} = \operatorname*{span}_{(x,w) \in \mathcal{X}_\theta \times \mathcal{V}_\ell} \{[\partial f^x(\phi(\theta))]^\top w\}. \tag{7}$$

*and, denoting $P_\ell(\phi(\theta)) \in \mathbb{R}^{d \times d}$ the matrix of the projection on this finite-dimensional vector subspace, we have*

$$\mathcal{W}_\theta^{g,\ell} = \operatorname{range}\big(\partial\phi(\theta)^\top P_\ell(\phi(\theta))\big). \tag{8}$$

With plain shallow ReLU and linear networks, when $\mathcal{V}_\ell = \mathbb{R}^n$ (this holds with the Euclidean loss or the Kullback-Leibler loss), there happens to a be a "distinguished" choice of $\phi$ (Marcotte et al., 2023, Lemma 2.13) such that the projection matrix $P_\ell$ is simply the identity, so that all the

---

[1]i.e., $x$ belongs to the set $\mathcal{X}_{\theta_0}$, as defined in Proposition 2.3.

needed information about $\mathcal{W}_{\theta}^{g,\ell}$ is captured in $\partial\phi$. The formalism with $P_{\ell}$ enhances the flexibility of the framework to encompass the variety of possible factorizations via $f$ and $\phi$.

In light of (8) we introduce the vectors fields:

$$\chi_i^{\ell} : \theta \mapsto \partial\phi(\theta)^{\top} P_{\ell}(\phi(\theta)) e_i, \quad 1 \le i \le d \qquad (9)$$

and the linear function space:

$$\mathbb{W}^{g,\ell} := \operatorname{span}\{\chi_1^{\ell}, \cdots, \chi_d^{\ell}\},$$

so that we have for any $\theta \in \Theta$, $\mathcal{W}_{\theta}^{g,\ell} = \mathbb{W}^{g,\ell}(\theta)$, where the *trace* at $\theta \in \Theta$ of any set $\mathbb{W} \subset \mathcal{C}^1(\Theta, \mathbb{R}^D)$ of vector fields is defined as the linear space

$$\mathbb{W}(\theta) := \operatorname{span}\{\chi(\theta) : \chi \in \mathbb{W}\} \subseteq \mathbb{R}^D. \qquad (10)$$

In particular $h \in \mathcal{C}^1$ is a conservation law of $g$ with respect to the loss $\ell$ if, and only if, its gradient is orthogonal to $\chi_i^{\ell}(\theta)$ for every $i$ and $\theta$. This property is stable by Lie brackets (for self-containedness see a reminder on the underlying calculus in Appendix B), leading to a new orthogonal characterization of conservation laws (proved in Appendix B).

**Proposition 2.9.** *If $\ell(z, y)$ satisfies Assumption 2.4 then $h \in \mathcal{C}^1(\Theta, \mathbb{R})$ is a conservation law for $g$ with respect to $\ell$ if and only if $\nabla h(\theta) \perp \operatorname{Lie}(\operatorname{span}_i\{\chi_i^{\ell}\})(\theta)$ for all $\theta \in \Theta$.*

### 2.2. Existence and "number" of conservation laws

Having characterized the conservation laws, we now seek to ascertain the quantity of such laws. However, to comprehend the number that exists, it is essential to establish a notion of independence that eliminates all functional redundancies. We recall the definition of independency from (Marcotte et al., 2023, Definition 2.18):

**Definition 2.10.** A family of functions $h_i$, $1 \le i \le N$ is said to be *independent* if the vectors $\nabla h_i(\theta)$, $1 \le i \le N$ are linearly independent for all $\theta \in \Omega$.

The following fundamental theorem (Marcotte et al., 2023, Theorem 3.3) provides a formula for the exact number of independent conservation laws.

**Theorem 2.11.** *If $\mathbb{W}^{g,\ell} \subseteq \mathcal{C}^{\infty}(\Theta, \mathbb{R}^D)$ and $\dim(\operatorname{Lie}(\mathbb{W}^{g,\ell})(\theta))$ is locally constant (equal to some $k$) then each $\theta \in \Theta \subseteq \mathbb{R}^D$ admits a neighborhood $U_0$ such that there are $D - k$ smooth ($\mathcal{C}^{\infty}$) independent conservation laws $h_{k+1}, \cdots, h_D$ of $g$ with respect to $\ell$ on $U_0$.*

The proof of this theorem in (Marcotte et al., 2023, Theorem 3.3) relies on Frobenius Theorem (recalled in Theorem C.1) and necessitates a reasoning by contradiction, along with the use of an intermediate functional space. In this article, we present a significantly simplified proof (see Appendix C)

that is based solely on the Frobenius Theorem and the characterization of Proposition 2.9.

We now demonstrate why Definition 2.10 effectively eliminates all functional redundancies. The following proposition states (see Appendix D for a proof) that any conservation law can be expressed locally as a function of the independent reference conservation laws obtained in Theorem 2.11.

**Proposition 2.12.** *Consider a smooth conservation law $h : \Theta \mapsto \mathbb{R}$ of $g$ with respect to $\ell$, and $\theta \in \Theta$ around which $\dim(\operatorname{Lie}(\mathbb{W}^{g,\ell})(\theta))$ is locally constant (equal to some integer $k$). Then, on the neighborhood $U_0$ of $\theta$ given by Theorem 2.11, $h$ can be expressed as a function of the $D - k$ smooth independent conservation laws $h_{k+1}, \cdots, h_D$ of $g$ given by Theorem 2.11.*

Marcotte et al. (2023) developed a code (detailed in their Section 3.3) that calculates the dimension of the trace of the Lie algebra generated by a finite set of vector fields.

### 2.3. Conservation laws for Adam flows

A simplified version of Adam algorithm (Kingma & Ba, 2014) (full batch, without bias correction and $\varepsilon, \beta_1, \beta_2 = 0$)[2] updates the parameters $\theta$, with an estimate of the first and second moments $m_t, v_t$ using the following equations:

$$
\begin{aligned}
m_t &= -\nabla_{\theta} L_Z(\theta_t), \\
v_t &= -\left(\nabla_{\theta} L_Z(\theta_t)\right)^2, \\
\theta_{t+1} &= \theta_t - \eta \frac{m_t}{\sqrt{v_t}} = \theta_t - \eta \operatorname{sign}(\nabla_{\theta} L_Z(\theta_t))
\end{aligned}
\qquad (11)
$$

where the square, square-root, and division are done element-wise.

The discrete dynamic (11) corresponds to the explicit Euler scheme of the ODE (informally corresponds to (11) when $\eta \to 0$):

$$\dot{\theta} = -\operatorname{sign}\left(\nabla_{\theta} L_Z(\theta)\right). \qquad (12)$$

*Remark* 2.13. Connections between the Adam algorithm and variants of sign gradient descent (referred to as "variance-adapted sign descent") have been established in (Balles & Hennig, 2018) in the discrete dynamic.

Thus one can adapt the results of Section 2.1 by considering the ODE (12) instead of (4). In particular under the same assumptions as in Proposition 2.3, a conservation law $h$ for $g$ for the Adam flow (12) with respect to the loss $\ell$ is now characterized by $\nabla h(\theta) \perp \mathcal{W}_{\theta}^{g,\ell}$, $\forall\theta$ where

$$\mathcal{W}_{\theta}^{g,\ell} := \operatorname*{span}_{Z=(x_i,y_i)\in(\mathcal{X}_{\theta}\times\mathcal{Y})^N} \{\operatorname{sign}\left(\nabla L_Z(\theta)\right)\}. \qquad (13)$$

---

[2]The continuous-time version of the Adam algorithm (Kingma & Ba, 2014) has also been studied, including the bias correction steps, with any $\epsilon, \beta_1, \beta_2$ and any batch size in (Barakat & Bianchi, 2021).

In particular, the associated space $\mathbb{W}^{g,\ell}$ is locally constant with respect to $\theta$. Thus, the trace of the generated Lie algebra at $\theta$ is directly equal to the one of $\mathbb{W}^{g,\ell}$ at $\theta$: by using Theorem 2.11, it suffices to determine the dimension of the trace of $\mathbb{W}^{g,\ell}$ to know the number of independent conservation laws. In the case of a 2-layer linear neural network parameterized by $\phi : (U, V) \in \mathbb{R}^{n \times r} \times \mathbb{R}^{m \times r} \mapsto UV^\top$ (and similarly for an attention layer), we empirically find that there are no conservation laws, except in the special case $n = m = r = 1$, where there is exactly one conservation law, given by $|U| - |V|$, as detailed in Appendix R.

# 3. The case of shallow neural networks

Equipped with the general results of the previous section we now provide characterizations of the conservation laws of several elementary networks that serve as building blocks of standard modern network architectures such as ResNets and Transformers. After showing that a basic block has the same conservation laws with or without skip connections, we remind existing laws for shallow ReLU and linear networks, before providing our main contributions of this section : a characterization of the conservation laws of shallow ReLU networks with convolutions, of attention layers, and of cross-entropy classification layers.

*Remark* 3.1 (Conservation and invariances)*.* In this section we characterize *all* conservation laws for elementary networks (except multihead layers where the completeness of the identified laws is left to future work). It is noteworthy that these laws in most cases are intrinsically connected to network invariances (as detailed in Appendix J.1). When considering weight decay regularization and applying the structure theorem Theorem 2.1, a particular consequence of our results is that the conservation laws from the non-regularized case (GF without weight decay) *vanish at the optimum*. This finding not only aligns with partially known "balancedness" properties ((Yang et al., 2022, Theorem 1) applies here in the case of a ReLU activation, as the associated elementary networks $g_\theta$ are homogeneous with respect to hidden neuron rescaling, and our results show that it is also true for linear networks and for cross-entropy classification layers (20)); it also provides additional insight on the *dynamics of the convergence* to balanced parameters.

## 3.1. With *vs* without residual connections

We first establish a simple but generic result for conservation laws in the presence of skip connexions. Given any neural network $g(\theta, \cdot)$ with $n = m$, we consider the residual neural network $\tilde{g}(\theta, \cdot)$ defined by:

$$\tilde{g}(\theta, \cdot) : x \in \mathbb{R}^n \mapsto x + g(\theta, x) \in \mathbb{R}^n. \tag{14}$$

**Proposition 3.2.** *With respect to any loss satisfying Assumption 2.4 $g$ and $\tilde{g}$ have the same conservation laws.*

*Proof.* Use Proposition 2.3 and Corollary 2.5, and notice that since $\partial_\theta g = \partial_\theta \tilde{g}$ we have $\mathcal{W}_\theta^{g,\ell} = \mathcal{W}_\theta^{\tilde{g},\ell}$. $\qquad \square$

*Remark* 3.3. It is worth noticing that this result does not require the assumption $\mathcal{V}_\ell = \mathbb{R}^n$.

## 3.2. ReLU and linear networks: known results

Let us consider $U \in \mathbb{R}^{n \times r}$, $V \in \mathbb{R}^{m \times r}$, and we denote $\theta := (U, V)$ the parameters and $u_k, v_k$ the columns of $U$ and $V$. In that case, the neural network writes:

$$g(\theta, x) = U\sigma(V^\top x), \tag{15}$$

where $\sigma = \texttt{id}$ (*resp.* $\sigma = \texttt{ReLU}$) in the linear case case (*resp.* ReLU case). The number of parameters is $D = (n + m)r$.

We recall here the result demonstrated in (Marcotte et al., 2023, Lemma 2.13, Theorem 2.14) which shows that, through an appropriate parameterization $\phi$, the study of $\mathbb{W}^{g,\ell}$ can be reduced to the study of a Lie algebra generated by a finite number of 'well-behaved' vector fields.

**Theorem 3.4.** *Under Assumption 2.4, if $\mathcal{V}_\ell = \mathbb{R}^n$, then considering $\Theta = \mathbb{R}^D$ and $\phi(\theta) := UV^\top$ for linear neural networks, one has: $\mathcal{W}_{\phi(\theta)}^{f,\ell} = \mathbb{R}^d$ and $\mathcal{W}_\theta^{g,\ell} = range(\partial\phi(\theta)^\top)$.*

*The same result holds for 2-layer ReLU networks (with or without bias $b_k \in \mathbb{R}$, $1 \leq k \leq r$) with $\Theta \subseteq \mathbb{R}^D$ the set of all parameters $\theta$ such that hidden neurons define pairwise distinct hyperplanes $H_k := \{x \in \mathbb{R}^d, v_k^\top + b_k = 0\}$, and $\phi(\theta) := (u_k v_k^\top)_{k=1}^r$*

Indeed, by Theorem 3.4, computing the associated generated Lie algebra and finally applying Theorem 2.11, the authors are able to determine *all* conservation laws in these settings (Marcotte et al., 2023, Corollary 4.4):

**Proposition 3.5.** *With the assumptions of Theorem 3.4, in the ReLU (resp. linear) case, we have: for any $\theta \in \Theta$ (resp. $\theta \in \Theta$ such that $\binom{U}{V}$ has full rank), there is a neighborhood of $\theta$ in which all conservation laws for (15) are functions of $\|u_k\|^2 - \|v_k\|^2$ (resp. of $\langle u_k, u_l \rangle - \langle v_k, v_l \rangle$), $1 \leq k, l \leq r$.*

## 3.3. ReLU neural networks with convolutions

We now consider the case of a basic block of a one-hidden layer ReLU neural network *with convolutions*. This means that the input vector $x \in \mathbb{R}^m$ is considered as the concatenation of channels $x^{(i)} \in \mathbb{R}^p$, $1 \leq i \leq c_0$ each with $p$ pixels (for images), or $p$ samples (in case of time series), so that $m = c_0 p$. Accordingly the output of the network is given by (15) where the matrices $V$ and $U$ are made of circulant blocks respectively corresponding to convolutions with filters $v_{j,i}, u_{k,j}$, $1 \leq i \leq c_0, 1 \leq j \leq c_1$ ($c_1$ is the number of channels of the hidden layer), and $1 \leq k \leq c_2$ ($c_2$ is the number of channels of the output $y = g(\theta, x) \in \mathbb{R}^n$, considered as the concatenation of channels $y^{(k)} \in \mathbb{R}^{n_1}$, so that $n = c_2 \times n_1$.

More explicitly this corresponds to

$$g(\theta, x) := \Big( \sum_{j=1}^{c_1} u_{k,j} \star \sigma \big( \sum_{i=1}^{c_0} v_{j,i} \star x^{(i)} \big) \Big)_{k=1}^{c_2}. \quad (16)$$

Assuming that the filters all satisfy $u_{k,j} \in \mathbb{R}^{n_u}$ (resp. $v_{i,j} \in \mathbb{R}^{n_v}$) the network parameters $\theta := ((u_{k,j})_{k,j}, (v_{j,i})_{j,i})$, are of dimension $D := c_2 c_1 n_u + c_1 c_0 n_v = c_1 (c_2 n_u + c_0 n_v)$.

We define $\Theta_{\texttt{conv}}$ as the set of all $\theta$ such that the matrix $V$ has all its columns that define pairwise distinct hyperplanes.

A conservation law for this network has already been established in (Du et al., 2018, Theorem 2.3) for a *single-channel* networks ($c_0 = c_1 = c_2 = 1$). We find that similar functions are preserved in the multi-channel case, and we demonstrate that there are no others by characterizing the trace of the associated Lie algebra as well as its dimension. The following theorem presents *all* conservation laws of the network (16). A more general version of this result is proved in Appendix E, which allows for instance to consider strided convolutions (Zhang, 2019) (i.e. with zeros inserted in the filter) in order to define translation invariant CNNs.

**Theorem 3.6.** *Under Assumption 2.4, if $\mathcal{V}_\ell = \mathbb{R}^n$, then in the neighborhood of each $\theta \in \Theta_{\texttt{conv}}$ there are exactly $c_1$ independent conservation laws for* (16) *given by*

$$h_j(\theta) := \sum_{k=1}^{c_2} \|u_{k,j}\|^2 - \sum_{i=1}^{c_0} \|v_{j,i}\|^2, \quad 1 \le j \le c_1. \quad (17)$$

*Remark* 3.7. The formulation of the neural network (16) in the multi-channel convolutive case generalizes the one (15) of a simple 2-layer ReLU network without convolution. Specifically, setting $p = 1$ and identifying $c_0$ and $n_v$ with $m$, $c_1$ with $r$, and $c_2$ and $n_u$ with $n$, yields (15) and the conservation laws obtained in Proposition 3.5 coincide with the ones given in Theorem 3.6.

### 3.4. One attention layer

For an attention layer, the input $X \in \mathbb{R}^{N \times \dim}$ is reshaped as the concatenation $x \in \mathbb{R}^m$ of $N$ tokens $x^{(i)} \in \mathbb{R}^{\dim}$, with $m = N\dim$. The layer output is

$$g(\theta, x) = \text{softmax}(XQ^\top K X^\top) X V^\top O \in \mathbb{R}^{N \times \dim} \quad (18)$$

(reshaped row by row as a $n$-dimensional vector with $n = N \times \dim$ to fit our generic notations), and where:

$$\text{softmax}(A)_i = \frac{\exp(A_i)}{\sum_{k=1}^N \exp(A_{ik})},$$

and with $Q, K, V, O \in \mathbb{R}^{d_1 \times \dim}$. We assume that all the columns of $O^\top V$ are non equal to zero. We consider $\Theta_{\texttt{att}}$ the set of all parameters that satisfy this condition.

We define $\phi(\theta) = (\phi_1, \phi_2)$ with $\phi_1 = Q^\top K$ and $\phi_2 = V^\top O$ the reparametrization such that (up to flattening of matrices into vectors) $g(\theta, x) = f(\phi, x) = \text{softmax}(X\phi_1 X^\top)X\phi_2$. The following theorem (see Appendix F for a proof) demonstrates that the parameterization $\phi$ is, in a some sense, sufficiently rich and allows for reduction to the study of a Lie algebra generated by the vector fields $(\theta \mapsto \partial\phi(\theta)^\top e_k)_k$.

**Theorem 3.8.** *Under Assumption 2.4, if $\mathcal{V}_\ell = \mathbb{R}^n$ and $N \ge 2$ then*

$$\mathcal{W}_{\phi(\theta)}^{f,\ell} = \mathbb{R}^d, \text{ and } \mathcal{W}_\theta^{g,\ell} = \text{range}\{\partial\phi(\theta)^\top\}, \; \forall \theta \in \Theta_{\texttt{att}}.$$

Thanks to this theorem, the analysis boils down to a similar problem as for Proposition 3.5 and allows us to determine *all* conservation laws. See Appendix G for a proof.

**Corollary 3.9.** *Under the assumptions of Theorem 3.8 for each $\theta \in \Theta_{\texttt{att}}$ such that both horizontally concatenated matrices $(Q, K)$ and $(V, O)$ have full rank, there is a neighborhood of $\theta$ in which all conservation laws for (18) are functions of $QQ^\top - KK^\top$, and $VV^\top - OO^\top$ and vice-versa.*

For more than one head, the neural network writes

$$g(\theta, X) = \sum_{h=1}^H \text{softmax}(XQ_h^\top K_h X^\top)XV_h^\top O_h, \quad (19)$$

up to matrix flattening, with $Q_h, K_h, V_h, O_h \in \mathbb{R}^{\frac{d_1}{H} \times \dim}$. In the case of multi-head attention, we can partially extend our results to obtain a set of conserved quantities (see Corollary 3.10, with proof detailed in Appendix G). However, determining whether this set of conservation laws is complete remains an open problem.

**Corollary 3.10.** *For any $h = 1, \cdots, H$, the functions*

$$Q_h Q_h^\top - K_h K_h^\top, \quad V_h V_h^\top - O_h O_h^\top,$$

*define conservation laws for (19) with respect to any loss $\ell$.*

### 3.5. Cross-Entropy Classification Layer

For classification tasks (which we consider in the numerical part Section 5.2), the final layer typically combines a linear transformation with the evaluation of a cross-entropy. This corresponds to using a Kullback-Leibler loss over $n$ classes $\ell(y, y') := \sum_{i=1}^n y_i \log(y_i/y_i') - y_i + y_i'$, together with a soft-max layer $g(\theta, \cdot)$ where $\theta \in \mathbb{R}^{n \times m}$:

$$g(\theta, x) := \text{softmax}(\theta x), \; \text{softmax}(z)_i := \frac{e^{z_i}}{\sum_j e^{z_j}}. \quad (20)$$

Note that this loss satisfies Assumption 2.4 and $\mathcal{V}_\ell = \mathbb{R}^n$. The following proposition (proved in Appendix H) shows that the softmax layer induces a new set of conservation laws with respect to this loss.

**Proposition 3.11.** *With respect to any loss $\ell$ such that $\mathcal{V}_\ell = \mathbb{R}^n$, there are exactly $m$ independent conservation laws for the classification layer given by* (20)*: $h_j(\theta) := \sum_i \theta_{i,j}, j = 1, \ldots, m$.*

## 4. Deeper ResNets and Transformers

In this section, we examine conservation laws for deep networks $g(\theta, \cdot)$ (denoted here as $g_\theta(x)$) composed of $q$ residual blocks. Specifically, each block $l$ consists of parameters $\theta_l$ such that $\theta = (\theta_1, \ldots, \theta_q)$ and corresponds to an elementary network denoted $g_{\theta_l}^l(x)$. Thus, the global network $g_\theta$ can be written as the composition of $q$ elementary networks:

$$g_\theta : x \in \mathbb{R}^m \mapsto g_{\theta_q}^q \circ \cdots \circ g_{\theta_1}^1(x) \in \mathbb{R}^m.$$

In case of a classification task using a cross-entropy loss, we consider a last block constitued of a linear (fully-connected) layer and a softmax activation $g_{\theta_{q+1}}^{q+1}$ as in (20), so that $\theta := (\theta_1, \cdots, \theta_{q+1})$ and the global network then writes:

$$g_\theta : x \in \mathbb{R}^m \mapsto g_{\theta_{q+1}}^{q+1} \circ g_{\theta_q}^q \circ \cdots \circ g_{\theta_1}^1(x) \in \mathbb{R}^n.$$

**Example 4.1** (Convolutive ResNet). A convolutive ResNet corresponds to $g_\theta : x \in \mathbb{R}^m \mapsto g_\theta(x) \in \mathbb{R}^n$ with $q$ residual blocks (we recall that here $m = c_0 p$). Each block $l \leq q$ has parameters $(u_{k,j}^l, v_{j,i}^l)$ and consists of a plain 2-layer convolutive ReLU network (16) with a skip connection:

$$g_{\theta_l}^l : x \mapsto x + U^l \sigma\left(V^l x\right), \tag{21}$$

with matrices $V^l$ and $U^l$ made of circulant blocks respectively corresponding to convolutions with filters $v_{j,i}^l, u_{k,j}^l$.

**Example 4.2** (Transformer). We consider a transformer architecture where $g_\theta : x \in \mathbb{R}^m \mapsto g_\theta(x) \in \mathbb{R}^n$(here $m = N \times \dim$) consists of alternating residual blocks with either an attention layer with one head, or a 2-layer MLP. We notably omit normalization layers, and restrict to a single head. In that case, up to appropriate harmless matricizations or flattening operations associating the vector $x$ to the token matrix $X$, each block $l \leq q$ writes either:

$$g_{\theta_l}^l : x \mapsto \text{vec}(X^\top + U_l \sigma(U_l' X^\top)), \tag{22}$$

where $\theta_l = (U_l, U_l')$ corresponds to the two weight matrices of a 2-layer MLP (15), or:

$$g_{\theta_l}(x) = \text{vec}([X + \text{softmax}(X Q_l^\top K_l X^\top) X V_l^\top O_l]^\top), \tag{23}$$

where $\theta_l = (Q_l, K_l, V_l, O_l)$ corresponds to the parameters of a single-head attention layer (18).

### 4.1. Characterization of "block" conservation laws

To analyze conservation laws in this context we focus on laws that depend only on a subset (or block) of parameters.

Considering $\theta \in \Theta$ the parameters of a global network $g_\theta$, we focus on $\theta_T$ a *subset* of the parameter entries (typically we will consider $\theta_T = \theta_l$ for some $l$, but other scenarios will also be covered), so we can write $\theta = (\theta_T, \theta_{T^c})$, where $\theta_{T^c}$ gathers the remaining entries of the parameters $\theta$.

The following proposition (see Appendix I) characterizes smooth conservation laws of $g$ that only depend on $\theta_T$.

**Proposition 4.3.** *Consider a function $h \in \mathcal{C}^1(\Theta, \mathbb{R})$ that only depends on the coordinates $\theta_T$, and for each $\theta \in \Theta$ denote*

$$\Theta_{T^c}(\theta_T) := \{\eta \in \mathbb{R}^{T^c} : (\theta_T, \eta) \in \Theta\} \tag{24}$$

*Consider a loss that satisfies the assumptions of Proposition 2.3 as well as Assumption 2.4. The function $h$ is a conservation law of $g$ with respect to $\ell$ if, and only if, for every $\theta \in \Theta$ one has $\nabla_{\theta_T} h(\theta) \perp R_{\theta_T}(\mathbb{W}^{g,\ell})$, where:*

$$R_{\theta_T}(\mathbb{W}^{g,\ell}) := \operatorname*{span}_{\substack{\eta \in \Theta_{T^c}(\theta_T) \\ w \in \mathcal{V}_\ell}} \operatorname*{span}_{x \in \mathcal{X}_{(\theta_T, \eta)}} \{\partial_{\theta_T} g((\theta_T, \eta), x)^\top w\}. \tag{25}$$

For concrete examples below, a technical challenge that arises when studying conservation laws that depend solely on $\theta_T$, and in comparing them with those of "internal" shallow networks involving only $\theta_T$, is the analysis of the set $\mathcal{X}_\theta$ of input $x$ *of the global* network $g_\theta$ around which it is smooth enough (cf the definition of $\mathcal{X}_\theta$ in Proposition 2.3), rather than the set of inputs of the considered "internal" shallow network. Overall, the important property for our purposes is the density of this set, proved in Appendix K.

**Lemma 4.4.** *Denote $\Theta = \Theta_q \times \ldots \times \Theta_1$ (or $\Theta = \Theta_{q+1} \times \Theta_q \times \ldots \times \Theta_1$ with $\Theta_{q+1} = \mathbb{R}^{n \times m}$ when there is a last softmax layer) where for each layer $1 \leq l \leq q$, $\Theta_l$ is the set of parameters $\theta_l$ such that*

1. *$g_{\theta_l}^l$ is an open map[3];*

2. *all the rows of the matrix $V^l$ (resp. $U_l'$) from (21) (resp. (22)) are nonzero in the convolutive ResNet case (resp. in the Transformer case).*

*For every $\theta \in \Theta$ we have $\overline{\mathcal{X}_\theta} = \mathbb{R}^m$.*

*Remark* 4.5. Obviously Item 2 only excludes a lower-dimensional set of parameters. We discuss in Appendix L why Item 1 is also a generic condition on the parameters for the example of a residual block associated with a 2-layer ReLU network.

### 4.2. Block laws for natural residual blocks

Consider $\theta_T := \theta_l$ where $l \in \{1, \cdots, q\}$. The following theorem (See Appendix N for a proof) demonstrates that

---

[3]*i.e.*, it sends an open set to an open set

the conservation laws of the global network $g$ that depend exclusively on $\theta_l$ are precisely those of the shallow network $g_{\theta_l}^l$ of the $l$-th residual block.

**Theorem 4.6.** *With $\Theta$ as in Lemma 4.4, consider the l-th residual block of Example 4.1 (resp. Example 4.2), and denote $\theta_T := \theta_l$ and $\theta_{T^c}$ the parameters of all other residual blocks. A function $H : \theta = (\theta_T, \theta_{T^c}) \in \Theta \mapsto h(\theta_T)$ that only depends on $\theta_T$ is a conservation law of g with respect to a loss $\ell$ such that $\mathcal{V}_\ell = \mathbb{R}^n$ if and only if h is a conservation law of the shallow residual network $g^l(\theta_l, x) := g_{\theta_l}^l(x)$ with respect to the Euclidean loss. The same result holds for $\theta_T := \theta_{q+1}$ when considering a last block (20).*

Thus by using Proposition 3.2, the conservation laws of $g$ that only depends on $\theta_l$ are exactly the ones of (16) (resp. (18)), which are described in Theorem 3.6 (resp. Corollary 3.9) for a residual block defined with a 2-layer ReLU networks (resp. with an attention layer).

### 4.3. Case of blocks overlapping a residual connection

This section focuses exclusively on the case of ResNet (with or without) convolutions as defined in Example 4.1. In the previous section Section 4.2, we examined the conservation laws of the global network $g$ that depend only on the parameters $\theta_l$ of the $l$-th block, and we just show that they exactly correspond to the ones of the shallow network $g_{\theta_l}^l$ corresponding to the $l$-th block. However, it is also possible to recast the global network as a composition of elementary networks that maintain parameter separation while involving a subset of parameters located before *and* after a residual connection as described in Figure 2 in Appendix O.

Specifically, let us consider $l \in \{1, \cdots, q-1\}$ and $\theta_T = (v_{j,i}^{l+1}, u_{k,j'}^l)$. In that case, $\theta_T$ corresponds to two consecutive parameter blocks that overlap a skip connection (as described in Figure 2 in Appendix O). Denoting $x_1$ (resp. $y_1$) the input of the intermediate layer of the $l$-th (resp. $l+1$-th) block, $x_2$ (resp. $y_2$) the copy of the input of the corresponding block that is transferred via the skip connection, $g$ can be written as a composition of $g_1$, $g_2$ and $g_3$ with

$$\left(\begin{smallmatrix} y_1 \\ y_2 \end{smallmatrix}\right) = g_2\left(\tilde{\theta}_2, \left(\begin{smallmatrix} x_1 \\ x_2 \end{smallmatrix}\right)\right) := \left(\begin{smallmatrix} V^{l+1} \\ I_d \end{smallmatrix}\right)\left(U^l \ I_d\right)\left(\begin{smallmatrix} \sigma \\ id \end{smallmatrix}\right)\left(\begin{smallmatrix} x_1 \\ x_2 \end{smallmatrix}\right),$$

$$\left(\begin{smallmatrix} x_1 \\ x_2 \end{smallmatrix}\right) = g_3(\tilde{\theta}_3, x) := \left(\begin{smallmatrix} V^l \\ I_d \end{smallmatrix}\right)(g_{\theta_{l-1}}^{l-1} \circ \ldots \circ g_{\theta_1}^1)(x),$$

and $g_1\left(\tilde{\theta}_1, \left(\begin{smallmatrix} y_1 \\ y_2 \end{smallmatrix}\right)\right) = g_{\theta_q}^q \circ \ldots \circ g_{\theta_{l+2}}^{l+2}\left(\left(U^{l+1} \ I_d\right)\left(\begin{smallmatrix} \sigma \\ id \end{smallmatrix}\right)\left(\begin{smallmatrix} y_1 \\ y_2 \end{smallmatrix}\right)\right)$

where $\tilde{\theta}_1, \tilde{\theta}_3$ gather all parameters appearing in $g_1$ and $g_3$. The following proposition (See Appendix P for a proof) shows that there exist no conserved functions of the global network that depend exclusively on $\tilde{\theta}_2 = \theta_T = (V^{l+1}, U^l)$.

**Theorem 4.7.** *Consider a layer index $1 \le l \le q-1$ and $\Theta$ defined as in Lemma 4.4 with the exception that for each*

$\theta_{l+1} \in \Theta_{l+1}$, *we further require that the rows of $V^{l+1}$ are pairwise non-colinear. If $n_v = p$ then any conservation law of g with respect to the Euclidean loss that only depends on $\tilde{\theta}_2$ is a constant function.*

### 4.4. Numerical confirmation

In the general case of a residual network composed of $q$ blocks, there could potentially exist conservation laws that depend on a larger subset of parameters than those previously analyzed, which helped us demonstrate that we recover exactly the same conservation laws as those of elementary blocks when considering the global network. However, by numerically computing the dimension of $\text{span}\{L_Z(\theta) : Z\} \subseteq \mathcal{W}_\theta^g$ with a sufficiently large batch size to adequately explore the space, we find that there is no additional conservation law when $m > 1$ for a ResNet with $q = 2$ residual blocks (see code in our GitHub repository).

When $m = 1$, numerical results suggest that there are additional conservation laws. It might be that this specific case enables as in (Marcotte et al., 2024, Theorem 4.1) to shed the light on new invariances that give rise to new conservation laws. This is confirmed by the following example: we exhibit a domain $\Omega$ where there are more conservation laws than the "block" ones.

**Example 4.8.** Consider a ReLU neural network with two-residual blocks, $g((u,v,s,t),x) = x + u\sigma(vx) + s\sigma(t(x + u\sigma(vx))$ with $(u,v,s,t) \in \Omega \subseteq \mathbb{R}^4$ and $x \in \mathbb{R}$. While there are two "block" conservation laws: $u^2-v^2$ and $s^2-t^2$, there are three conservation laws on the set $\Omega$ of all parameters such that $\text{sgn}(t) = \text{sgn}(v) = \text{sgn}(u)$. Indeed, for every $x$,

1. either $vx, tx \le 0$, hence $g(\theta, x) = x$ and $\nabla_\theta g(\cdot) = 0$;

2. or $vx, tx \ge 0$, hence $g(\theta, x) = x + uvx + stx + stuvx$ as $vx, tx, tu \ge 0$, and thus $\frac{1}{x}\nabla_\theta g(\cdot) =: \chi_1(\cdot)$ is a vector field that does not depend on $x$.

As the space $\mathbb{W}^{g,\ell} = \mathbb{R}\chi_1$ is spanned by a single non-null vector field, its Lie algebra is itself, and by Theorem 2.11, there are exactly $4 - 1 = 3$ conservation laws as claimed.

## 5. Conservation laws for *discrete* dynamics

In practice optimization is performed with a discrete dynamics associated to stochastic gradient descent. To what extent do conservation laws still apply in this context? This is the object of this section.

### 5.1. (Stochastic) gradient descent as a training dynamic

We consider the ERM problem (1). Instead of using gradient descent (2), we consider the stochastic gradient descent

(SGD) method on mini-batches:

$$\theta_{k+1} = \theta_k - \tau_k \nabla L_{Z_k}(\theta_k),$$

where the sequence of mini-batches $(Z_k)_k$ is drawn i.i.d. from the data distribution. The following proposition (see Appendix Q for a proof) shows that a conservation law is approximately preserved by SGD.

**Proposition 5.1.** *Let $h(\theta)$ be a conservation law of the gradient flow with a bounded Hessian $\forall \theta, \quad \|\partial^2 h(\theta)\| \leq C_h$. Suppose that the gradients remain bounded in expectation throughout the algorithm:*

$$\mathbb{E}_{Z_k, \theta_k} \left\| \nabla L_{Z_k}(\theta_k) \right\|_2^2 \leq C_L. \tag{26}$$

*Then, we have* $\quad \mathbb{E}\left| h(\theta_k) - h(\theta_0) \right| \leq \dfrac{C_h C_L}{2} \sum_{i=0}^{k-1} \tau_i^2. \tag{27}$

For a constant step size $\tau_k = \tau$, this proposition shows that the conservation error grows as $|h(\theta_k) - h(\theta_0)| = O(\tau^2 k)$, which increases linearly with the number of iterations. If one uses a decaying step size, such as $\tau_k = \tau_0/(k+1)$, which ensures convergence of the method, then the error remains bounded: $|h(\theta_k) - h(\theta_0)| = O(\tau_0^2)$. The requirement that $h$ has a bounded Hessian holds in the case considered in this paper since the conservation laws we examine are quadratic. The key assumption for the result to hold is the bound on the gradient magnitude in (26). This condition is met if the loss function $\ell$ and the network $g_\theta$ are uniformly Lipschitz, though this is not generally true for deep networks. It also holds with an explicit constant $C_L$ for smooth convex losses (Bach, 2024), but this setting is quite restrictive. More generally, such bounds hold for smooth losses when the variance of $\nabla L_{Z_k}(\theta)$ is bounded (Garrigos & Gower, 2023), because the iterates are bounded in expectations, $\mathbb{E}(\|\theta_k\|^2) < +\infty$, though the constant $C_L$ may not be explicitly determined. In the numerical experiments presented in Section 5.2, we empirically evaluate the constants to show that, in practice, they remain relatively small, ensuring approximate conservation.

### 5.2. Numerical experiments

In Figure 1, we train a ResNet-18 on CIFAR-10 (Krizhevsky, 2009) while tracking the difference between the squared Frobenius norms of consecutive convolutional layers in the first residual block, considering the conservation law $h(\theta_T) := \sum_{j=1}^{c_1} h_j(\theta_T)$, where $h_j$ is defined in (17), with $\theta_T$ representing the parameters of the first residual block. We vary the learning rate between $10^{-3}$ and $5 \times 10^{-3}$, using stochastic gradient descent (SGD) without momentum or weight decay. For each learning rate, we train 10 models for 50 steps with 10 different random seeds, recording both the loss evolution (bottom) and the evolution of the conservation error $|(h(\theta_k) - h(\theta_0))/h(\theta_0)|$ (top). The dotted

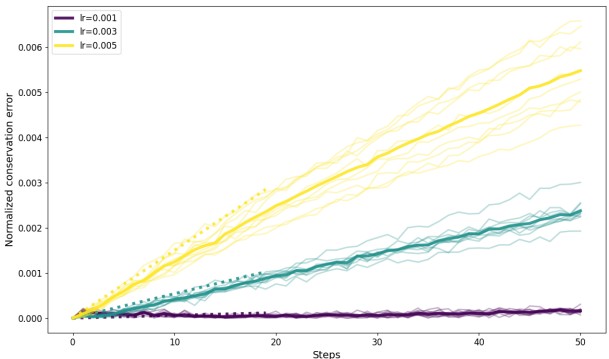

*Figure 1.* Tracking a conserved function during ResNet-18 training on CIFAR-10.

lines show the theoretical slopes $C\tau^2$ derived from (27), confirming that the function is approximately conserved and that the slope coefficient maintains proportionality with $\tau^2$. Our code is available at our GitHub repository.

In another experiment, we train a transformer model on the IMDb sentiment analysis dataset (Maas et al., 2011) using SGD optimization. We track the evolution of the Frobenius norm of the conserved matrix identified in Corollary 3.10, specifically examining the query and key matrices from the first attention head in the first layer. Consistent with our ResNet training results presented in Figure 1, we observe that the conservation error scales as $O(\text{step-size}^2)$ throughout training, which confirms our theoretical bound (27). Furthermore, it is worth noting that the numerical behavior is unchanged whether masking is applied or not. See Appendix O for the associated figures. Our code is available at our GitHub repository

### Conclusion

This paper investigates conservation laws in deep networks (ResNet and Transformer architectures) within gradient flow dynamics and examines their behavior under discrete SGD dynamics. Our analysis does not currently account for transformer normalization layers, and the multi-head attention mechanism is only partially addressed. The integration of these components as well as max-pooling layers presents promising avenues for future research.

### Acknowledgements

The work of G. Peyré was supported by the European Research Council (ERC project WOLF) and the French government under management of Agence Nationale de la Recherche as part of the "Investissements d'avenir" program, reference ANR-19-P3IA-0001 (PRAIRIE 3IA Institute). The work of R. Gribonval was partially supported by the AllegroAssai ANR project ANR-19-CHIA-0009 and the

SHARP ANR Project ANR-23-PEIA-0008 of the PEPR IA, funded in the framework of the France 2030 program.

## Impact Statement

This paper presents work whose goal is to advance the field of Machine Learning. There are many potential societal consequences of our work, none which we feel must be specifically highlighted here.

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

# A. Proof of Theorem 2.1 (Structure theorem)

To prove Theorem 2.1, it will be essential to juggle with certain properties of conserved quantities when they depend on $(t, \theta)$. Here we recall definitions and properties directly derived from (Marcotte et al., 2024), which are necessary to understand the proof of the structure theorem.

## A.1. Definitions of conservation

We recall definitions and properties from (Marcotte et al., 2024).

**Definition A.1** (Conservation through a (family of) flow(s)). Consider an open subset $\Omega \subseteq \mathbb{R} \times \mathbb{R}^D$ and a function $F \in \mathcal{C}^1(\Omega, \mathbb{R}^D)$. By the Cauchy-Lipschitz theorem, for each initial condition $\texttt{init} := (t_{\text{init}}, \theta_{\text{init}}) \in \Omega$, there exists a unique maximal solution $t \in (t_{\text{init}} - \eta_{\text{init}}, t_{\text{init}} + \eta_{\text{init}}) \mapsto \theta(t, \texttt{init})$ of the ODE $\dot{\theta}(t) = F(t, \theta(t))$ with $\theta(t_{\text{init}}) = \theta_{\text{init}}$. A function $h : \Omega \subseteq \mathbb{R}^{D+1} \to \mathbb{R}$ is *conserved on $\Omega$ through the flow $F$* if $h(t, \theta(t, \texttt{init})) = h(t_{\text{init}}, \theta_{\text{init}})$ for each choice of $\texttt{init} \in \Omega$ and every $t \in (t_{\text{init}} - \eta_{\text{init}}, t_{\text{init}} + \eta_{\text{init}})$. It is *conserved on $\Omega$ through a family of flows* if $h$ is conserved on $\Omega$ through all flows.

**Proposition A.2** (Smooth functions conserved through a family of flows). *Let $\mathcal{I}$ be any set. Given $F_i \in \mathcal{C}^1(\Omega, \mathbb{R}^D)$ for any $i \in \mathcal{I}$, a function $h \in \mathcal{C}^1(\Omega, \mathbb{R})$ is conserved through the family of flows induced by all $F_i$ if and only if $\langle \nabla h(\alpha), (1, F_i(\alpha)^\top)^\top \rangle = 0$ for all $\alpha \in \Omega$ and for all $i \in \mathcal{I}$. Moreover by denoting $\chi_i$ the vector field on $\Omega$ defined by*

$$\chi_i : \alpha \in \Omega \mapsto \begin{pmatrix} 1 \\ F_i(\alpha) \end{pmatrix} \in \mathbb{R}^{D+1}, \tag{28}$$

*and by denoting*

$$\mathbb{W} := \operatorname{span}_i \{\chi_i\} \subset \mathcal{C}^1\left(\Omega, \mathbb{R}^{D+1}\right), \tag{29}$$

*this exactly means that for all $\alpha \in \Omega$, $\langle \nabla h(\alpha), \mathbb{W}(\alpha) \rangle = 0$, where the* trace *of $\mathbb{W}$ is defined by*

$$\mathbb{W}(\alpha) := \operatorname{span}\{\chi(\alpha) : \chi \in \mathbb{W}\} = \operatorname{span}_i \{\chi_i(\alpha)\}. \tag{30}$$

In the context of the dynamic (3), we consider $\alpha = (t, \theta) \in \mathbb{R} \times \Theta \subseteq \mathbb{R}^{D+1}$ and we consider the flow $F_Z$ defined by

$$F_Z(t, \theta) = -M(\theta) \nabla L_Z(\theta) - \lambda(t) \theta. \tag{31}$$

*We keep in that section the matrix $M(\theta)$ (that allows to deal with non-euclidean metric as in (Marcotte et al., 2024)). However, we do not focus in that paper on the non-euclidean setting.*

**Definition A.3** (Conservation during the flow (3) with a given dataset). Consider an open subset $\Omega = \Omega_t \times \Omega_\theta \subseteq \mathbb{R} \times \Theta$ and a dataset $Z = (x_i, y_i)_i$ such that $L_Z \in \mathcal{C}^2(\Omega_\theta, \mathbb{R})$. A function $h : \Omega \subseteq \mathbb{R}^{D+1} \to \mathbb{R}$ is *conserved on $\Omega$ during the flow* (3) if it is conserved through the flow induced by $F_Z$ defined in (31).

To identify conserved functions that do not depend on a specific dataset, we focus on a more precise class of conserved functions: those that remain conserved during all flows defined by the ordinary differential equation (ODE) (3). This leads us to the following definition. The goal is to derive universal laws that hold true regardless of any particular dataset.

**Definition A.4** (Conservation during the flow (3) with "any" dataset). Consider an open subset $\Omega = \Omega_t \times \Omega_\theta \subset \mathbb{R} \times \Theta$ and a loss $\ell(z, y)$ such that $\ell(\cdot, y)$ is $\mathcal{C}^2$-differentiable for all $y \in \mathcal{Y}$. A function $h : \Omega \subseteq \mathbb{R}^{D+1} \to \mathbb{R}$ is *conserved on $\Omega$ for any data set* if, for each data set $(X, Y)$ such that $g(\cdot, x_i) \in \mathcal{C}^2(\Omega_\theta, \mathbb{R}^n)$ for each $i$, the function $h$ is conserved on $\Omega$ during the flow (3). This leads us to introduce the family of vector fields:

$$\mathbb{W}_\Omega{}^g := \left\{ \chi(\cdot) : \exists Z, \forall i \ g(\cdot, x_i) \in \mathcal{C}^2(\Omega_\theta, \mathbb{R}^n), \ \chi(\cdot) = (1, F_Z(\cdot)^\top)^\top \right\} \subseteq \mathcal{C}^1(\Omega, \mathbb{R}^{D+1}) \tag{32}$$

so that being conserved on $\Omega$ for any dataset is the same as being conserved on $\Omega$ through $\mathbb{W}_\Omega{}^g$.

The definitions provided above are local and contingent upon the selection of an open set of parameters $\Omega_\theta \subset \Theta$. However, our primary interest lies in functions defined across the entire parameter space $\Theta$; thus, we present the following definition.

**Definition A.5** (**Conservation law** for a neural network). A function $h : \mathbb{R} \times \Theta \mapsto \mathbb{R}$ is *a conservation law* if for each open subset $\Omega \subseteq \mathbb{R} \times \Theta$, $h$ is conserved on $\Omega$ for any data set.

Therefore, by using the "orthogonality" characterization of a conserved function Proposition A.2 and Definition A.5, the object of interest to study locally conserved functions is the union of the traces

$$\mathcal{W}_\alpha^g := \bigcup \left\{ \mathbb{W}_\Omega{}^g(\alpha) \ : \ \Omega \subseteq \mathbb{R} \times \Theta \text{ with } \Omega \text{ a neighborhood of } \alpha \right\}. \tag{33}$$

**Corollary A.6.** *A function $h : \mathbb{R} \times \Theta \mapsto \mathbb{R}$ is a conservation law if and only if $\nabla h(\alpha) \perp \mathcal{W}_\alpha^g$ for all $\alpha \in \mathbb{R} \times \Theta$.*

**Proposition A.7.** *Assume that for each $y \in \mathcal{Y}$ the loss $\ell(z, y)$ is $\mathcal{C}^2$-differentiable with respect to $z \in \mathbb{R}^n$. For each $\alpha = (t, \theta) \in \mathbb{R} \times \Theta$ we have:*

$$\mathcal{W}_\alpha^g = \operatorname*{span}_{Z=(x_i,y_i)\in(\mathcal{X}_\theta\times\mathcal{Y})^N} \left\{ (1, -[M(\theta)\nabla L_Z(\theta) + \lambda(t)\theta]^\top)^\top \right\}$$

*where $\mathcal{X}_\theta$ is the set of data points $x$ such that $g(\cdot, x)$ is $\mathcal{C}^2$-differentiable in the neighborhood of $\theta$.*

## A.2. Proof of Theorem 2.1

**Theorem 2.1** (Structure theorem). *Let $h(t, \theta)$ be a conserved function for the ODE (3). If for every $\theta$, there exists a data-set $Z$ such that $\nabla L_Z(\theta) = 0$, then the function $H(a) := h(0, a)$ satisfies $h(t, \theta) = H\big(\theta \exp(\int_0^t \lambda(s)\mathrm{d}s)\big)$, $\forall t, \theta$. Thus, conserved functions can be expressed with $D$ variables (instead of $D + 1$). Moreover, $\tilde{h}(t, \theta) := H(\theta)$ is a conservation law of (3) without WD (i.e. with $\lambda(t) \equiv 0$).*

To manipulate the first assumption we outline it as follows:

**Assumption A.8.** *For all $\theta \in \Theta$, there exists $Z \in \mathcal{Z}_\theta'$ such that $\nabla L_Z(\theta) = 0$, where we denote $\mathcal{Z}_\theta'$ the collection of all data set $Z = (x_i, y_i)_i$ such that for all $i$, $g(\cdot, x_i)$ is $\mathcal{C}^2$-differentiable in the neighborhood of $\theta$*

Assumption A.8 is satisfied for all classical losses e.g. for the mean-squared or the cross-entropy loss as shown in Lemma D.2 of (Marcotte et al., 2024).

*Proof.* Let $h(t, \theta)$ be a conservation law. Then by Corollary A.6, for each $\alpha = (t, \theta) \in \mathbb{R} \times \Theta$ we have: $\langle \nabla h(\alpha), \mathcal{W}_\alpha^g \rangle = 0$. In particular, thanks to Proposition A.7 and Assumption A.8, one has $\langle \nabla h(\alpha), (1, -[\lambda(t)\theta]^\top) \rangle = 0$. By Proposition A.2, $h$ is in particular conserved through the flow induced by $F(t, \theta) := -\lambda(t)\theta$. For each initialization $(0, \theta_0)$, the solution of the ODE $\dot{\theta}(t) = F(t, \theta(t))$ is $\theta(t) = \theta_0 \exp(-\int_0^t \lambda(s)\mathrm{d}s)$ for $t \in \mathbb{R}$ and by definition of a conserved function, one has $h(t, \theta(t)) = h(0, \theta(0)) = h(0, \theta(t) \exp(\int_0^t \lambda(s)\mathrm{d}s)) = H(\theta(t) \exp(\int_0^t \lambda(s)\mathrm{d}s))$. Thus for all $t$ and $\theta$, one has $h(t, \theta) = H(\theta \exp(\int_0^t \lambda(s)\mathrm{d}s))$.

Now, let us show that $(t, \theta) \mapsto H(\theta)$ is a conservation law of the GF without WD scenario ($\lambda(t) \equiv 0$). For any $\alpha = (t, \theta) \in \mathbb{R} \times \Theta$, one has

$$\nabla h(\alpha) = \begin{pmatrix} \partial_t h(\alpha) \\ \nabla_\theta h(\alpha) \end{pmatrix} = \begin{pmatrix} \langle \nabla H(\theta \exp(\int_0^t \lambda(s)\mathrm{d}s)), \lambda(t)\theta \exp(\int_0^t \lambda(s)\mathrm{d}s) \rangle \\ \exp(\int_0^t \lambda(s)\mathrm{d}s) \nabla H(\theta \exp(\int_0^t \lambda(s)\mathrm{d}s)) \end{pmatrix}.$$

Let us consider $\alpha = (t, \theta) \in \mathbb{R} \times \Theta$. By taking $Z \in \mathcal{Z}_\theta'$ (NB: here we do *not* choose it such that $\nabla L_Z(\theta) = 0$) and using the characterization (Corollary A.6) of a conservation law and Proposition A.7, one has:

$$0 = \left\langle \nabla h(\alpha), \begin{pmatrix} 1 \\ -M(\theta)\nabla L_Z(\theta) - \lambda(t)\theta \end{pmatrix} \right\rangle$$

$$= \exp\left(\int_0^t \lambda(s)\mathrm{d}s\right)\left(\langle \lambda(t)\theta, \nabla H(\theta \exp(\int_0^t \lambda(s)\mathrm{d}s)) \rangle\right) +$$

$$\langle \nabla H(\theta \exp(\int_0^t \lambda(s)\mathrm{d}s)), -M(\theta)\nabla L_Z(\theta) - \lambda(t)\theta \rangle\big)$$

$$= -\exp\left(\int_0^t \lambda(s)\mathrm{d}s\right)\langle \nabla H(\theta \exp(\int_0^t \lambda(s)\mathrm{d}s)), M(\theta)\nabla L_Z(\theta) \rangle.$$

In particular, this remains true for $t = 0$, and thus one has:

$$\langle \nabla H(\theta), M(\theta) \nabla L_Z(\theta) \rangle = 0.$$

Finally by denoting $H^0 : (t, \theta) \mapsto H(\theta)$, one has

$$\left\langle \nabla H^0(t, \theta), \begin{pmatrix} 1 \\ -M(\theta) \nabla L_Z(\theta) \end{pmatrix} \right\rangle = 0,$$

and thus $H^0$ is a conservation law for the GF without WD case by using Corollary A.6 and Proposition A.7. □

## B. Proof of Proposition 2.9 (Characterization of conservation laws via Lie algebras)

First, we recall that the **Lie bracket** $[\chi_1, \chi_2]$ of two vector fields $\chi_1, \chi_2 \in \mathcal{C}^\infty(\Omega, \mathbb{R}^D)$ is the vector field defined by:

$$[\chi_1, \chi_2] : \quad \theta \in \Omega \mapsto [\chi_1, \chi_2](\theta) := \partial \chi_1(\theta) \chi_2(\theta) - \partial \chi_2(\theta) \chi_1(\theta). \tag{34}$$

Let us consider $W \subseteq \mathcal{C}^\infty(\Omega, \mathbb{R}^D)$. The **generated Lie algebra** of $W$ is the smallest space $A \subseteq \mathcal{C}^\infty(\Omega, \mathbb{R}^D)$ that contains $W$ and that is stable by Lie brackets, i.e. for any $\chi_1, \chi_2 \in A$ one has $[\chi_1, \chi_2] \in A$. We denote $A := \mathrm{Lie}(W)$.

Proposition 2.9 is a direct consequence of this lemma:

**Lemma B.1.** *Let $h$ be a real-valued function defined on $\Omega$ and let consider two smooth vector fields $\chi_1$ and $\chi_2$ defined on $\Omega$ satisfying for all $\theta \in \Omega$:*

$$\langle \nabla h(\theta), \chi_1(\theta) \rangle = 0 \quad and \quad \langle \nabla h(\theta), \chi_2(\theta) \rangle = 0, \tag{35}$$

*then*

$$\langle \nabla h(\theta), [\chi_1, \chi_2](\theta) \rangle = 0. \tag{36}$$

*Proof.* Let us denote $\chi$ the Lie bracket of $\chi_1$ and $\chi_2$. By definition

$$\chi(\theta) := [\chi_1, \chi_2](\theta) = \partial \chi_1(\theta) \chi_2(\theta) - \partial \chi_2(\theta) \chi_1(\theta). \tag{37}$$

Differentiating (35) gives

$$\partial \chi_i(\theta)^\top \nabla h(\theta) = -\partial(\nabla h)(\theta)^\top \chi_i(\theta) = -\partial(\nabla h)(\theta) \chi_i(\theta), \tag{38}$$

as $\partial(\nabla h)(\alpha)$ is self-adjoint. Using this relation (38), we have

$$
\begin{aligned}
\langle \nabla h(\theta), \chi(\theta) \rangle &= \langle \partial \chi_1(\theta)^\top \nabla h(\theta), \chi_2(\theta) \rangle - \langle \partial \chi_2(\theta)^\top \nabla h(\theta), \chi_1(\theta) \rangle \\
&= -\langle \partial(\nabla h)(\theta) \chi_1(\theta), \chi_2(\theta) \rangle + \langle \partial(\nabla h)(\theta)^\top \chi_2(\theta), \chi_1(\theta) \rangle \\
&= 0. \quad\quad\quad\quad\quad\quad\quad\quad\quad\quad\quad\quad\quad\quad\quad\quad\quad\quad\quad\quad\quad\quad\quad\quad\quad\quad\quad\quad \square
\end{aligned}
$$

## C. A new and simplified proof of Theorem 2.11 (Marcotte et al., 2023, Theorem 3.3)

We first recall Frobenius theorem (see for example (Marcotte et al., 2023, Theorem E.1) with the same notations or see (Isidori, 1995, Section 1.4) for control theory practitioners).

**Theorem C.1** (Frobenius theorem). *Consider $\mathbb{W} \subseteq \mathcal{C}^\infty(\Omega, \mathbb{R}^D)$, and assume that the dimension of its trace $\mathbb{W}(\theta)$ is constant: $\dim \mathbb{W}(\theta) = k$ for every $\theta \in \Omega \subseteq \mathbb{R}^D$. Then the two following assertions are equivalent:*

1. *each $\theta \in \Omega$ admits a neighborhood $U_0$ such that there exists $D - k$ smooth ($\mathcal{C}^\infty$) real-valued functions $h_{k+1}, \cdots, h_D$ on $U_0$ such that for all $\theta' \in U_0$:*

$$\mathrm{span}\{\nabla h_{k+1}(\theta'), \cdots, \nabla h_D(\theta')\} = \mathbb{W}(\theta')^\perp; \tag{39}$$

2. *the following property holds:*

$$[\chi_1, \chi_2](\theta) \in \mathbb{W}(\theta), \ \forall \chi_1, \chi_2 \in \mathbb{W}, \forall \theta \in \Omega. \tag{40}$$

By using Proposition 2.9 and Theorem C.1, we derive the following corollary, which is already established in (Marcotte et al., 2023), Theorem 3.3. Nevertheless, this new proof, by using Proposition 2.9, is more intuitive and straightforward.

**Corollary C.2.** *Let us assume that $\mathbb{W}^{g,\ell} \subseteq \mathcal{C}^\infty\left(\Theta, \mathbb{R}^D\right)$. If $\dim(\mathrm{Lie}(\mathbb{W}^{g,\ell})(\theta)) = k$ is locally constant then each $\theta \in \Theta \subseteq \mathbb{R}^D$ admits a neighborhood $U_0$ such that there are $D - k$ smooth conservation laws $h_{k+1}, \cdots, h_D$ of $g$ on $U_0$ and such that for all $\theta' \in U_0$, the vectors $(\nabla h_{k+1}(\theta'), \cdots, \nabla h_D(\theta'))$ are linearly independent.*

Thus by using Definition 2.10, Corollary C.2 can be reformulated as

**Corollary C.3.** *Let us assume that $\mathbb{W}^{g,\ell} \subseteq \mathcal{C}^\infty\left(\Theta, \mathbb{R}^D\right)$. If $\dim(\mathrm{Lie}(\mathbb{W}^{g,\ell})(\theta)) = k$ is locally constant then each $\theta \in \Theta \subseteq \mathbb{R}^D$ admits a neighborhood $U_0$ such that there are $D - k$ smooth independent conservation laws $h_{k+1}, \cdots, h_D$ of $g$ on $U_0$.*

# D. Proof of Proposition 2.12 (role of *independent* conservation laws)

**Proposition 2.12.** *Consider a smooth conservation law $h : \Theta \mapsto \mathbb{R}$ of $g$ with respect to $\ell$, and $\theta \in \Theta$ around which $\dim(\mathrm{Lie}(\mathbb{W}^{g,\ell})(\theta))$ is locally constant (equal to some integer $k$). Then, on the neighborhood $U_0$ of $\theta$ given by Theorem 2.11, $h$ can be expressed as a function of the $D - k$ smooth independent conservation laws $h_{k+1}, \cdots, h_D$ of $g$ given by Theorem 2.11.*

*Proof.* We consider some $\theta \in \Theta$ such that $\dim(\mathrm{Lie}(\mathbb{W}^{g,\ell})(\theta')) = k$ is locally constant. By using Theorem 2.11, there exist $h_{k+1}, \cdots, h_D$ smooth conservation laws of $g$ on a neighborhood $U_0$, and thus by using Proposition 2.9 for any $\theta' \in U_0$:

$$\mathrm{span}\{\nabla h_{k+1}(\theta'), \cdots, \nabla h_D(\theta')\} = \mathrm{Lie}(\mathbb{W}^{g,\ell})(\theta')^\perp. \tag{41}$$

But as $h$ is also a conservation law of $g$, by using again Proposition 2.9, one has that for any $\theta' \in U_0$: $\nabla h(\theta') \in \mathrm{span}\{\nabla h_{k+1}(\theta'), \cdots, \nabla h_D(\theta')\}$. This exactly means that $\nabla h$ is linearly dependent of all gradients $\nabla h_i$ everywhere on $U_0$. If all conservation laws are ($\mathcal{C}^\infty$) smooth on $U_0$, this corresponds to the fact (cf (Newns, 1967, Theorem 1)) that $h$ is then a function of the $h_i$ on $U_0$. $\qquad\square$

# E. Proof of Theorem 3.6 (Conservation laws for convolutive two-layer ReLU networks)

**We first recall all notations.** Let $c_0$ be the number of channels for the input. Let $c_1$ be the number of channels for the hidden layer, and let $c_2$ be the number of channels for the final output. Let $n_u$ (resp. $n_v$) be the size of the filters for the second (resp. first) convolution, and let $p$ be the number of pixels of the input image. In that setting, we consider $\theta := ((u_{k,j})_{k,j}, (v_{j,i})_{j,i})$, so that $D := (n_u c_2 + n_v c_0) \times c_1$, and the parameter vector consists of the collection of all filters. We write the input and the output:

$$x = \begin{pmatrix} x^{(1)} \\ \cdots \\ x^{(c_0)} \end{pmatrix} \in \mathbb{R}^m, \quad \text{with } m = c_0 \times p \qquad y = \begin{pmatrix} y^{(1)} \\ \cdots \\ y^{(c_2)} \end{pmatrix} \in \mathbb{R}^n, \quad \text{with } n = c_2 \times n_1.$$

We denote $G(\theta, x)$ the convolutive 2-layer ReLU neural network defined by:

$$G(\theta, x) := \left[ \sum_{j=1}^{c_1} C_1(u_{k,j}) \sigma \left( \sum_{i=1}^{c_0} C_2(v_{j,i})^\top x^{(i)} \right) \right]_{k=1}^{c_2}, \tag{42}$$

where $C_1 : \mathbb{R}^{n_u} \mapsto \mathbb{R}^{n_1 \times p_1}$ and $C_2 : \mathbb{R}^{n_v} \mapsto \mathbb{R}^{p \times p_1}$ are linear operators.

*In particular, the formulation (42) is more general than (16). We explicit below the corresponding operators whenin the case of circular convolution.*

**Assumption E.1.** We assume that we can write $C_1(u) = (P_1 u, \cdots, P_{p_1} u)$ (resp. $C_2(v) = (Q_1 v, \cdots, Q_{p_1} v)$) where all matrices $P_i \in \mathbb{R}^{n_1 \times n_u}$ (resp. $Q_i \in \mathbb{R}^{p \times n_v}$) are injective.

**Link with the two formulations.** In case of a "full" circular convolution (i.e. $n_u = p$), $P_i$ is the circular shift operator $P_i u = u_{\cdot - i} \pmod{p}$, that corresponds to consider $P_i$ the $i$-cyclic shift matrix, and in particular $P_i$ is injective. In the case of filters of size $n_u \leq p$, then by considering $I$ the canonical injection $I : \mathbb{R}^{n_u} \mapsto \mathbb{R}^p$ that adds zeros, then $P_i u = \tilde{P}_i I(u)$, with $\tilde{P}_i$ the circular shift operator of the previous case $n_u = p$, in particular $P_i = \tilde{P}_i I$ is injective too.

**We now explain how** (42) **can be expressed with a structured** 2-**layer ReLU neural network** (15).

Obviously (42) can be rewritten as

$$G(\theta, x) = U\sigma(V^\top x) \tag{43}$$

where $U = U(\theta)$ and $V = V(\theta)$ are defined blockwise as

$$U := [C_1(u_{k,j})]_{1 \le k \le c_2, 1 \le j \le c_1} \quad V := [C_2(v_{j,i})]_{1 \le i \le c_0, 1 \le j \le c_1}$$

It will be convenient for further calculations to consider the vertical concatenation of blocks associated to a given index $1 \le j \le c_1$. For this reason we write $u^{(j)} := (u_{k,j})_{k=1}^{c_2} \in \mathbb{R}^{n_u c_2}$, $v^{(j)} := (v_{j,i})_{i=1}^{c_0} \in \mathbb{R}^{n_v c_0}$ and

$$\overline{C}_1(u^{(j)}) := \begin{pmatrix} C_1(u_{1,j}) \\ \cdots \\ C_1(u_{c_2,j}) \end{pmatrix} \in \mathbb{R}^{(n_1 c_2) \times p_1}, \quad \text{so that} \quad U(\theta) = \begin{pmatrix} \overline{C}_1(u^{(1)}) & \cdots & \overline{C}_1(u^{(c_1)}) \end{pmatrix} \in \mathbb{R}^{(n_1 c_2) \times (p_1 c_1)} \tag{44}$$

and

$$\overline{C}_2(v^{(j)}) := \begin{pmatrix} C_2(v_{j,1}) \\ \cdots \\ C_2(v_{j,c_0}) \end{pmatrix} \in \mathbb{R}^{(pc_0) \times p_1}, \quad \text{so that} \quad V(\theta) = \begin{pmatrix} \overline{C}_2(v^{(1)}) & \cdots & \overline{C}_2(v^{(c_1)}) \end{pmatrix} \in \mathbb{R}^{(pc_0) \times (p_1 c_1)}. \tag{45}$$

It will also be convenient to observe that by construction, using the notations of Assumption E.1 and the definition of the Kronecker product between matrices

$$\overline{C}_1(u^{(j)}) = \begin{pmatrix} P'_1 u^{(j)} & \cdots & P'_{p_1} u^{(j)} \end{pmatrix} \quad \text{with } P'_i = I_{c_2} \otimes P_i \tag{46}$$

$$\overline{C}_2(v^{(j)}) = \begin{pmatrix} Q'_1 v^{(j)} & \cdots & Q'_{p_1} v^{(j)} \end{pmatrix} \quad \text{with } Q'_i = I_{c_0} \otimes Q_i \tag{47}$$

Thus $G(\theta, x) = g((U, V), x)$ where $g$ is the 2-layer ReLU network defined in (15).

**Lemma E.2.** *Consider $G(\theta, x)$ as in* (42)*, $g((U, V), x)$ from* (15)*, and $\phi$ the reparametrization of $g$ defined in Theorem 3.4 in the (unstructured / nonconvolutive) ReLU case. Denote $\psi(\theta) := \phi(T\theta)$, where $T\theta := vec((\begin{smallmatrix} U \\ V \end{smallmatrix})) \in \mathbb{R}^{(n_1 c_2 + pc_0)p_1 c_1}$ with $U = U(\theta)$ and $V = V(\theta)$ defined blockwise as above. Denote $\Theta$ the set of all parameters such that the columns of $V(\theta)$ define pairwise disjoint hyperplanes. Under Assumption 2.4, if $\mathcal{V}_\ell = \mathbb{R}^n$, then for each $\theta \in \Theta$ we have*

$$\mathcal{W}_\theta^{G,\ell} = \operatorname*{span}_{\gamma \in \mathbb{R}^d} \{\partial\psi(\theta)^\top \gamma\} \quad and \quad \mathbb{W}^{G,\ell} = \operatorname{span}\{\nabla\psi_i(\cdot) : i = 1, \cdots, d\}.$$

*Proof.* We have $G(\theta, x) = g(T\theta, x) = f(\phi(T\theta), x) = f(\psi(\theta), x)$, hence the set $\mathcal{X}_\theta$ associated to the model $G(\cdot, x)$ (see within Proposition 2.3) coincides with the set $\mathcal{X}_{T\theta}$ associated to $g(\cdot, x)$, and the general definition (7) yields $\mathcal{W}_{\psi(\theta)}^{f,\ell} = \mathcal{W}_{\phi(T\theta)}^{f,\ell}$. Moreover, by definition of $\Theta$, each parameter $\theta \in \Theta$ is such $T\theta$ satisfies the assumptions of Theorem 3.4. By Theorem 3.4 it follows that $\mathcal{W}_{\psi(\theta)}^{f,\ell} = \mathcal{W}_{\phi(T\theta)}^{f,\ell} = \mathbb{R}^d$. Finally by using Proposition 2.7, one obtains: $\mathcal{W}_\theta^{G,\ell} = \operatorname*{span}_{\gamma \in \mathbb{R}^d} \{\partial\psi(\theta)^\top \gamma\}$ and $\mathbb{W}^{G,\ell} = \operatorname{span}\{\nabla\psi_i(\cdot) : i = 1, \cdots, d\}$. $\square$

**Proposition E.3.** *Consider any structured two-layer ReLU network model as in* (42) *where the linear operators $C_1$, $C_2$ satisfy Assumption E.1, and $\Theta$ defined as in Lemma E.2. For each $\theta \in \Theta$, there exists a neighborhood of $\theta$ in which there are exactly $c_1$ independent conservation laws for* (42)*. Such independent conservation laws are given, e.g., by $\theta \mapsto \sum_{k=1}^{c_2} \|u_{k,j}\|^2 - \sum_{i=1}^{c_0} \|v_{j,i}\|^2$, for $j = 1, \cdots c_1$.*

Before proving this proposition let us highlight that it yields Theorem 3.6 as a direct corollary by the very definition of $\Theta_{\text{conv}}$ and the fact that the matrices $P_i$, $Q_i$ associated to the convolutive model indeed satisfy Assumption E.1.

*Proof.* Recall that $P'_i = I_{c_2} \otimes P_i$ and $Q'_i = I_{c_0} \otimes Q_i$, so that $P'_i$ (resp. $Q'_i$ is injective) as $P_i$ (resp. $Q_i$) is injective. It will be useful to denote $\overline{P}_i := \begin{pmatrix} P'_i & 0 \\ 0 & Q'_i \end{pmatrix}$ and $\overline{P} = \begin{pmatrix} \overline{P}_1 \\ \cdots \\ \overline{P}_{p_1} \end{pmatrix}$ in order to get a compact explicit expression of $T$ (step 1 below). This will be used to characterize the trace of the Lie algebra of $\mathbb{W}^{G,\ell}$ and its dimension.

*1st step: We first give an explicit expression of $T$.*

Writing $\theta = \begin{pmatrix} u^{(1)} \\ v^{(1)} \\ \cdots \\ u^{(c_1)} \\ v^{(c_1)} \end{pmatrix}$, where we recall that for $j = 1, \cdots, c_1$, $u^{(j)} = \mathrm{vec}((u_{k,j})_k) \in \mathbb{R}^{n_u c_2}$ and $v^{(j)} = \mathrm{vec}((v_{j,i})_i) \in$

$\mathbb{R}^{n_v c_0}$, we get using once again the definition of the Kronecker product,

$$T\theta = \mathrm{vec} \begin{pmatrix} U(\theta) \\ V(\theta) \end{pmatrix} \overset{(44)-(45)=(46)-(47)}{=} \left( I_{c_1} \otimes \overline{P} \right) \theta. \tag{48}$$

*2d step: Characterization of $\mathbb{W}^{G,\ell}$ and of the trace of its Lie algebra.*

Recall that the reparameterization $\phi$ of two-layer ReLU networks from Theorem 3.4 reads $\phi : (U, V) \in \mathbb{R}^{(n_1 c_2) \times (p_1 c_1)} \times \mathbb{R}^{(p c_0) \times (p_1 c_1)} \mapsto (u_i v_i^\top)_{i=1,\cdots,p_1 c_1} \in (\mathbb{R}^{n_1 c_2 \times p c_0})^{p_1 c_1}$ with $u_i, v_i$ the columns of the matrices $U, V$. Let $\Delta := (\Delta_{i,j})_{i,j} \in (\mathbb{R}^{n_1 c_2 \times p c_0})^{p_1 c_1}$ and let us consider $\gamma := \mathrm{vec}(\Delta)$.

Straightforward computations show that $\partial \phi(U,V)^\top \gamma = \begin{pmatrix} S_{\Delta_{1,1}} & \cdots & 0 \\ 0 & S_{\Delta_{2,1}} & \cdots \\ & \cdots & \\ 0 & \cdots & S_{\Delta_{p_1,c_1}} \end{pmatrix} \mathrm{vec}(U;V)$, where: $S_{\Delta_{i,j}} :=$

$\begin{pmatrix} 0 & \Delta_{i,j} \\ \Delta_{i,j}^\top & 0 \end{pmatrix}$. Since $\psi(\theta) := \phi(T\theta)$ it follows that

$$\partial \psi(\theta)^\top \gamma = T^\top \partial \phi(T\theta)^\top \gamma = T^\top \begin{pmatrix} S_{\Delta_{1,1}} & \cdots & 0 \\ & \cdots & \\ 0 & \cdots & S_{\Delta_{p_1,c_1}} \end{pmatrix} T\theta$$

Using Equation (48) we thus obtain

$$\partial \psi(\theta)^\top \gamma = \mathtt{blockdiag}_j \left( \overline{P}^\top \left( \mathtt{blockdiag}_i(S_{\Delta_{i,j}}) \right) \overline{P} \right) \theta = \mathtt{blockdiag}_j \left( \sum_{i=1}^{p_1} \overline{P_i}^\top S_{\Delta_{i,j}} \overline{P_i} \right) \theta.$$

The above is valid for any $\Delta$, and we now exhibit particular choices of $\Delta$ to obtain particular constructions, in order to characterize $\mathbb{W}^{G,\ell}$. Given some $(i,j) \in \{1,\cdots,p_1\} \times \{1,\cdots,c_1\}$, imposing $\Delta_{i',j'} = 0$ for all $(i',j') \neq (i,j)$ yields

$$\partial \psi(\theta)^\top \gamma = \begin{pmatrix} 0 & \cdots & 0 \\ & \cdots & \\ 0 & \overline{P_i}^\top S_{\Delta_{i,j}} \overline{P_i} & 0 \\ & \cdots & \\ 0 & \cdots & 0 \end{pmatrix} \theta. \tag{49}$$

As $\overline{P_i}^\top S_{\Delta_{i,j}} \overline{P_i} = \begin{pmatrix} 0 & P_i'^\top \Delta_{i,j} Q_i' \\ Q_i'^\top \Delta_{i,j}^\top P_i' & 0 \end{pmatrix} \in \mathbb{R}^{(\tilde{n}_u + \tilde{n}_v) \times (\tilde{n}_u + \tilde{n}_v)}$ with $\tilde{n}_u := n_u c_2$ and $\tilde{n}_v := n_v c_0$, we obtain

$$\mathbb{W}^{G,\ell} \overset{\text{Lemma E.2}}{=} \mathrm{span}\{\theta \mapsto \partial \psi(\theta)^\top \gamma : \gamma\}$$

$$\subseteq \mathrm{span} \left\{ \theta \mapsto \begin{pmatrix} \begin{pmatrix} 0 & A_1 \\ A_1^\top & 0 \end{pmatrix} & \cdots & 0 \\ & \cdots & \\ 0 & \cdots & \begin{pmatrix} 0 & A_{c_1} \\ A_{c_1}^\top & 0 \end{pmatrix} \end{pmatrix} \theta : A_i \in \mathbb{R}^{\tilde{n}_u \times \tilde{n}_v}, 1 \leq i \leq c_1 \right\}. \tag{50}$$

We now show the converse inclusion. Let us consider some $i = 1, \cdots, c_1$ and some $A \in \mathbb{R}^{\tilde{n}_u \times \tilde{n}_v}$. To show the converse inclusion, it is enough to show that there exists $\Delta_{i,j} \in \mathbb{R}^{n_1 c_2 \times p c_0}$ such that $P_i'^\top \Delta_{i,j} Q_i' = A$. As $P_i'$ is injective by

hypothesis, then $P_i'^\top$ is surjective. Thus, there exists $B = \begin{pmatrix} b_1 & \cdots & b_{\tilde{n}_v} \end{pmatrix} \in \mathbb{R}^{(n_1 c_2) \times \tilde{n}_v}$ such that $P_i'^\top B = A$. Then as $Q_i' = \begin{pmatrix} q_1 & \cdots & q_{\tilde{n}_v} \end{pmatrix}$ is injective, the vectors $q_j$ are linearly independent. Let us define $\Delta_{i,j} \in \mathbb{R}^{n_1 c_2 \times p c_0}$ such that for all $k = 1, \cdots, \tilde{n}_v$, $\Delta_{i,j} q_k = b_k$. In particular, such a $\Delta_{i,j}$ satisfies $P_i'^\top \Delta_{i,j} Q_i' = A$.

Thus:

$$
\text{span} \left\{ \theta \mapsto \begin{pmatrix} \begin{pmatrix} 0 & A_1 \\ A_1^\top & 0 \end{pmatrix} & \cdots & 0 \\ & \cdots & \\ 0 & \cdots & \begin{pmatrix} 0 & A_{c_1} \\ A_{c_1}^\top & 0 \end{pmatrix} \end{pmatrix} \theta : A_i \in \mathbb{R}^{\tilde{n}_u \times \tilde{n}_v}, 1 \leq i \leq c_1 \right\} \subseteq \text{span}\{\theta \mapsto \partial\psi(\theta)^\top \gamma : \gamma\} = \mathbb{W}^{G,\ell}. \quad (51)
$$

Finally by using (50) and (51) we have characterized the space $\mathbb{W}^{G,\ell}$ as

$$
\mathbb{W}^{G,\ell} = \text{span} \left\{ \theta \mapsto \begin{pmatrix} \begin{pmatrix} 0 & A_1 \\ A_1^\top & 0 \end{pmatrix} & \cdots & 0 \\ & \cdots & \\ 0 & \cdots & \begin{pmatrix} 0 & A_{c_1} \\ A_{c_1}^\top & 0 \end{pmatrix} \end{pmatrix} \theta : A_i \in \mathbb{R}^{\tilde{n}_u \times \tilde{n}_v}, 1 \leq i \leq c_1 \right\},
$$

To conclude this step we need to compute the trace of its Lie algebra

$$
\text{Lie}(\mathbb{W}^{G,\ell}) = \text{Lie} \left( \text{span} \left\{ \begin{pmatrix} \begin{pmatrix} 0 & A_1 \\ A_1^\top & 0 \end{pmatrix} & \cdots & 0 \\ & \cdots & \\ 0 & \cdots & \begin{pmatrix} 0 & A_{c_1} \\ A_{c_1}^\top & 0 \end{pmatrix} \end{pmatrix} : A_i \in \mathbb{R}^{\tilde{n}_u \times \tilde{n}_v}, 1 \leq i \leq c_1 \right\} \right).
$$

For this we can reuse computations from (Marcotte et al., 2023, Proposition H.3) showing that the considered Lie algebra is indeed a space of linear operators that can be identified with matrices:

$$
\text{Lie} \left( \text{span} \left\{ \begin{pmatrix} \begin{pmatrix} 0 & A_1 \\ A_1^\top & 0 \end{pmatrix} & \cdots & 0 \\ & \cdots & \\ 0 & \cdots & \begin{pmatrix} 0 & A_{c_1} \\ A_{c_1}^\top & 0 \end{pmatrix} \end{pmatrix} : A_i \in \mathbb{R}^{\tilde{n}_u \times \tilde{n}_v} \right\} \right)
$$
$$
= \left\{ \left( I_{c_1} \otimes \begin{pmatrix} I_{\tilde{n}_u} & 0 \\ 0 & -I_{\tilde{n}_v} \end{pmatrix} \right) \times \begin{pmatrix} M_1 & \cdots & 0 \\ & \cdots & \\ 0 & \cdots & M_{c_1} \end{pmatrix} : M_i \in \mathcal{A}_{\tilde{n}_u + \tilde{n}_v} \right\},
$$

where $\mathcal{A}_{\tilde{n}_u + \tilde{n}_v} \subset \mathbb{R}^{(\tilde{n}_u + \tilde{n}_v)^2}$ is the space of skew symmetric matrices.

*3d step: Let us show that there are exactly $c_1$ independent conservation laws.*

Let us consider $\theta \in \Theta$ (in particular $\theta \neq 0$). To conclude, by using Theorem 2.11, we only need to compute $\dim(\text{Lie}(\mathbb{W}^G)(\theta))$ (as the computation is unchanged for $\theta'$ in a neighborhood of $\theta$, we will also get that this dimension is locally constant as needed).

For this, denoting $\theta^{(i)} = \begin{pmatrix} u^{(i)} \\ v^{(i)} \end{pmatrix}$, we consider the linear application:

$$
\Gamma : (M_1, \cdots, M_{c_1}) \in (\mathcal{A}_{\tilde{n}_u + \tilde{n}_v})^{c_1} \mapsto \left( I_{c_1} \otimes \begin{pmatrix} I_{\tilde{n}_u} & 0 \\ 0 & -I_{\tilde{n}_v} \end{pmatrix} \right) \times \begin{pmatrix} M_1 & \cdots & 0 \\ & \cdots & \\ 0 & \cdots & M_{c_1} \end{pmatrix} \begin{pmatrix} \theta^{(1)} \\ \cdots \\ \theta^{(c_1)} \end{pmatrix},
$$

As $\text{range}(\Gamma) := \Gamma((\mathcal{A}_{\tilde{n}_u + \tilde{n}_v})^{c_1}) = \text{Lie}\left(\mathbb{W}^G\right)(\theta)$, we only need to compute rank $\Gamma$ as in (Marcotte et al., 2023, Proposition H.5). Since, by the rank–nullity theorem, we have $\dim \ker \Gamma + \text{rank } \Gamma = c_1(\tilde{n}_u + \tilde{n}_v)(\tilde{n}_u + \tilde{n}_v - 1)/2$, it is equivalent to compute $\dim \ker \Gamma$. It is also easy to check that $\dim \ker \Gamma = \dim \ker \Gamma_1 + \cdots + \dim \ker \Gamma_{c_1}$, where

$$
\Gamma_i : M \in \mathcal{A}_{\tilde{n}_u + \tilde{n}_v} \mapsto \begin{pmatrix} I_{\tilde{n}_u} & 0 \\ 0 & -I_{\tilde{n}_v} \end{pmatrix} \times M \theta^{(i)}.
$$

By (Marcotte et al., 2023, Proposition H.5) (remember that $\theta^{(i)}$ is a vector in that case and not a matrix), one has $\dim \ker \Gamma_i = (\tilde{n}_u + \tilde{n}_v - 2)(\tilde{n}_u + \tilde{n}_v - 1)/2$, so that $\dim \ker \Gamma = c_1(\tilde{n}_u + \tilde{n}_v - 2)(\tilde{n}_u + \tilde{n}_v - 1)/2$, and thus

$$\text{rank } \Gamma = c_1(\tilde{n}_u + \tilde{n}_v - 1) = D - c_1$$

since the number of parameters is indeed $D = c_1(\tilde{n}_u + \tilde{n}_v)$. By Theorem 2.11, we conclude that there are exactly $c_1$ independent conservation laws.

*4th step: Finally, let us check that the claimed laws are indeed a set of $c_1$ independent conservation laws.*

Let us define for $j = 1, \cdots, c_1$: $h_j : \theta^{(j)} \mapsto \sum_{k=1}^{c_2} \|u_{k,j}\|^2 - \sum_{i=1}^{c_0} \|v_{j,i}\|^2 = \|u^{(j)}\|^2 - \|v^{(j)}\|^2$ and $\tilde{h}_j : \theta \mapsto h_j(\theta^{(j)})$.

By Proposition 2.9 and the expression of the gradient of $\tilde{h}_j$ in the terms of $\nabla h_j$, the function $\tilde{h}_j$ is a conservation law for (42) if and only if the function $h_j$ satisfies for all $\theta$ and for all $M \in \mathcal{A}_{\tilde{n}_u + \tilde{n}_v}$:

$$\nabla h_j(\theta^{(j)}) = \nabla_{\theta^{(j)}} \tilde{h}_j(\theta) \perp \begin{pmatrix} \mathrm{I}_{\tilde{n}_u} & 0 \\ 0 & -\mathrm{I}_{\tilde{n}_v} \end{pmatrix} \times M\theta^{(j)}. \tag{52}$$

Since $h_j : \theta^{(j)} \mapsto u^{(j)\top} u^{(j)} - v^{(j)\top} v^{(j)}$, by (Marcotte et al., 2023, Proposition H.1), this function indeed satisfies (52). Thus $\tilde{h}_j$ is a conservation law for (42). Finally, it is straightforward to check that the gradient of the functions $h_j$, $1 \leq j \leq c_1$ are nonzero vectors with disjoint supports, hence they are linearly independent. This shows that the considered conservation laws are indeed independent. $\square$

## F. Proof of Theorem 3.8 for attention layers

**Theorem 3.8.** *Under Assumption 2.4, if $\mathcal{V}_\ell = \mathbb{R}^n$ and $N \geq 2$ then*

$$\mathcal{W}_{\phi(\theta)}^{f,\ell} = \mathbb{R}^d, \text{ and } \mathcal{W}_\theta^{g,\ell} = \mathrm{range}\{\partial\phi(\theta)^\top\}, \ \forall\theta \in \Theta_{\mathtt{att}}.$$

*Proof.* Consider $\theta \in \Theta_{\mathtt{att}}$. By Proposition 2.7 it is sufficient to show that $\mathcal{W}_{\phi(\theta)}^{f,\ell} = \mathbb{R}^d$, since this implies that $\mathcal{W}_\theta^{g,\ell} = \mathrm{range}\{\partial\phi(\theta)^\top\}$. Since the considered model is smooth, here we have $\mathcal{X}_\theta = \mathbb{R}^m$, which can be identified (with the reshaping of $x \in \mathbb{R}^m$ into $X \in \mathbb{R}^{N \times \dim}$) to $\mathbb{R}^{N \times \dim}$. Recall for convenience that $\phi(\theta) = (\phi_1, \phi_2)$ with $\phi_1 = Q^\top K$, $\phi_2 = V^\top O$ and $f(\phi, x) = \mathrm{softmax}(X\phi_1 X^\top)X\phi_2$. As we assume $\mathcal{V}_\ell = \mathbb{R}^n$, the space we need to consider is thus

$$\mathcal{W}_{\phi(\theta)}^{f,\ell} = \operatorname*{span}_{X \in \mathbb{R}^{N \times \dim}, \ \Delta \in \mathbb{R}^{N \times \dim}} \{[\partial_\phi f(\phi(\theta), X)]^\top \cdot \Delta\}$$

Our goal is to show that $\mathcal{W}_{\phi(\theta)}^{f,\ell} = \mathbb{R}^{\dim \times (2\dim)}$. As a warmup given any $H \in \mathbb{R}^{\dim \times \dim}$ and $\Delta \in \mathbb{R}^{N \times \dim}$, we have:

$$\langle \partial_{\phi_2} f \cdot H, \Delta \rangle = \langle \mathrm{softmax}(X\phi_1 X^\top)XH, \Delta \rangle = \langle H, X^\top[\mathrm{softmax}(X\phi_1 X^\top)]^\top \Delta \rangle \tag{53}$$

hence

$$[\partial_{\phi_2} f]^\top \cdot \Delta = X^\top[\mathrm{softmax}(X\phi_1 X^\top)]^\top \Delta \tag{54}$$

*1st step: Considering any $j, l \in \{1, \cdots, \dim\}$, we first show that*

$$\begin{pmatrix} 0 \\ E_{j,l} \end{pmatrix} \in \operatorname*{span}_{X \in \mathbb{R}^{N \times \dim}, \ \Delta \in \mathbb{R}^{N \times \dim}} \{\partial_\phi f(\phi(\theta), X)^\top \cdot \Delta\} \tag{55}$$

*and*

$$\begin{pmatrix} E_{j,j} \\ 0 \end{pmatrix} \in \operatorname*{span}_{X \in \mathbb{R}^{N \times \dim}, \ \Delta \in \mathbb{R}^{N \times \dim}} \{\partial_\phi f(\phi, X)^\top \cdot \Delta\}. \tag{56}$$

Consider any $i \in \{1, \cdots, N\}$ and $X = E_{i,j}$. By (54) we have

$$[\partial_{\phi_2} f]^\top \cdot \Delta = E_{j,i}[\mathrm{softmax}(E_{i,j}\phi_1 E_{j,i})]^\top \Delta = e_j \, [\mathrm{softmax}(E_{i,j}\phi_1 E_{j,i})e_i]^\top \Delta. \tag{57}$$

Moreover denoting $\alpha_j = \alpha_j(\phi_1) := \langle e_j, \phi_1 e_j \rangle$ we have

$$\mathrm{softmax}(E_{i,j}\phi_1 E_{j,i})e_i = \tfrac{1}{N}\mathbf{1} + \left( \frac{\exp(\alpha_j)}{\exp(\alpha_j)+N-1} - \frac{1}{N} \right) e_i,$$

By straightforward calculus we obtain that for any $K \in \mathbb{R}^{\dim \times \dim}$

$$\partial_{\phi_1} f \cdot K = \langle e_j, K e_j \rangle [(\underbrace{\frac{\exp(\alpha_j)(N-1)}{(\exp(\alpha_j)+N-1)^2}}_{=:\beta_j}) e_i](\phi_2^\top e_j)^\top.$$

Thus for any $\Delta \in \mathbb{R}^{N \times \dim}$, one has:

$$\langle \partial_{\phi_1} f \cdot K, \Delta \rangle = \langle \langle e_j, K e_j \rangle \beta_j e_i (\phi_2^\top e_j)^\top, \Delta \rangle = \langle e_j, K e_j \rangle \langle \beta_j e_i (\phi_2^\top e_j)^\top, \Delta \rangle = \langle K, e_j e_j^\top \rangle \langle \beta_j e_i (\phi_2^\top e_j)^\top, \Delta \rangle$$

hence

$$[\partial_{\phi_1} f]^\top \cdot \Delta = e_j e_j^\top \langle \beta_j e_i (\phi_2^\top e_j)^\top, \Delta \rangle. \tag{58}$$

Combining (57) and (58) yields

$$[\partial_\phi f]^\top \cdot \Delta = \begin{pmatrix} [\partial_{\phi_1} f]^\top \cdot \Delta \\ [\partial_{\phi_2} f]^\top \cdot \Delta \end{pmatrix} = \begin{pmatrix} e_j e_j^\top \langle \beta_j e_i (\phi_2^\top e_j)^\top, \Delta \rangle \\ e_j \left( \frac{1}{N} \mathbf{1} + \left( \frac{\exp(\alpha_j)}{\exp(\alpha_j)+N-1} - \frac{1}{N} \right) e_i \right)^\top \Delta \end{pmatrix}.$$

Specializing to $\Delta = E_{k,l}$ for some $k \in \{1, \cdots, N\}$ we have

$$\left( \frac{1}{N} \mathbf{1} + \left( \frac{\exp(\alpha_j)}{\exp(\alpha_j)+N-1} - \frac{1}{N} \right) e_i \right)^\top \Delta = \begin{cases} \frac{1}{N} e_l^\top & \text{if } k \neq i \\ \frac{\exp(\alpha_j)}{\exp(\alpha_j)+N-1} e_l^\top & \text{if } k = i \end{cases}$$

and

$$\beta_j^{-1} \langle \beta_j e_i (\phi_2^\top e_j)^\top, \Delta \rangle = \langle e_i (\phi_2^\top e_j)^\top, e_k e_l^\top \rangle = \langle e_k^\top e_i (\phi_2^\top e_j)^\top, e_l^\top \rangle = \begin{cases} 0 & \text{if } k \neq i \\ \langle (\phi_2^\top e_j)^\top, e_l^\top \rangle & \text{if } k = i \end{cases}$$

hence

$$\partial_\phi f^\top \cdot \Delta = \begin{cases} \frac{1}{N} \begin{pmatrix} 0 \\ e_j e_l^\top \end{pmatrix} & \text{when } k \neq i \\ \begin{pmatrix} C_1 e_j e_j^\top \langle (\phi_2^\top e_j)^\top, e_l^\top \rangle \\ C_2 e_j e_l^\top \end{pmatrix}, & \text{when } k = i \end{cases} \tag{59}$$

where $C_1$ and $C_2$ are nonzero. The case $k \neq i$ is possible since we assume $N \geq 2$ and it yields (55) directly, while combining both cases yields

$$\langle (\phi_2^\top e_j)^\top, e_l^\top \rangle \begin{pmatrix} E_{j,j} \\ 0 \end{pmatrix} \in \operatorname*{span}_{X \in \mathbb{R}^{N \times \dim}, \Delta \in \mathbb{R}^{N \times \dim}} \{\partial_\phi f(\phi(\theta), X)^\top \cdot \Delta\}.$$

Finally as by hypothesis on $\Theta_{\texttt{att}}$ all columns of $O^\top V = \phi_2^\top$ are nonzero, we have $\phi_2^\top e_j \neq 0$, hence there exists some $l_1$ such that $\langle (\phi_2^\top e_j)^\top, e_{l_1}^\top \rangle \neq 0$. This directly implies (56), which concludes this step.

*2d step: Considering an arbitrary pair $j, l \in \{1, \cdots, \dim\}$, we now show that*

$$\begin{pmatrix} E_{j,l} \\ 0 \end{pmatrix} \in \operatorname*{span}_{X \in \mathbb{R}^{N \times \dim}, \Delta \in \mathbb{R}^{N \times \dim}} \{\partial_\phi f(\phi(\theta), X)^\top \cdot \Delta\}. \tag{60}$$

This is of course a trivial from (56) when $j = l$ (which is always the case when $\dim = 1$) so we focus on $j \neq l$.

Considering $X = E_{i,j} + E_{k,l}$, for some pair of indicex $i < k \in \{1, \cdots, N\}$ (possible as $N \geq 2$). we have

$$f(\phi, X) = \operatorname{softmax}(X \phi_1 X^\top) X \phi_2 = \left( \frac{1}{N} \mathbf{1} + (\alpha(\phi_1) e_i + \beta(\phi_1) e_k) (\phi_2^\top e_j)^\top + \left( \frac{1}{N} \mathbf{1} + \gamma(\phi_1) e_i + \delta(\phi_1) e_k \right) (\phi_2^\top e_l)^\top \right)$$

where

$$\alpha(\phi_1) := \frac{\exp(\langle e_j, \phi_1 e_j\rangle)}{N - 2 + \exp(\langle e_j, \phi_1 e_j\rangle) + \exp(\langle e_j, \phi_1 e_l\rangle)} - \frac{1}{N}$$

$$\beta(\phi_1) := \frac{\exp(\langle e_l, \phi_1 e_j\rangle)}{N - 2 + \exp(\langle e_l, \phi_1 e_j\rangle) + \exp(\langle e_l, \phi_1 e_l\rangle)} - \frac{1}{N}$$

$$\gamma(\phi_1) := \frac{\exp(\langle e_j, \phi_1 e_l\rangle)}{N - 2 + \exp(\langle e_j, \phi_1 e_j\rangle) + \exp(\langle e_j, \phi_1 e_l\rangle)} - \frac{1}{N}$$

$$\delta(\phi_1) := \frac{\exp(\langle e_l, \phi_1 e_l\rangle)}{N - 2 + \exp(\langle e_l, \phi_1 e_j\rangle) + \exp(\langle e_l, \phi_1 e_l\rangle)} - \frac{1}{N}$$

Straightforward calculus yields that for any $H \in \mathbb{R}^{\dim \times \dim}$

$$\partial_{\phi_1} f \cdot H = e_i \left[ a_1 \langle e_j, He_j\rangle + b_1 \langle e_j, He_l\rangle \right] (\phi_2^\top e_j)^\top + e_i \left[ c_1 \langle e_j, He_l\rangle + d_1 \langle e_j, He_j\rangle \right] (\phi_2^\top e_l)^\top$$
$$+ e_k \left[ a_2 \langle e_l, He_j\rangle + b_2 \langle e_l, He_l\rangle \right] (\phi_2^\top e_j)^\top + e_k \left[ c_2 \langle e_l, He_l\rangle + d_2 \langle e_l, He_j\rangle \right] (\phi_2^\top e_l)^\top,$$

with appropriate scalars $a_i, b_i, c_i, d_i$, where $b_1 \neq 0$. Thus for any $\Delta \in \mathbb{R}^{N \times \dim}$ and $H \in \mathbb{R}^{\dim \times \dim}$

$$\langle [\partial_{\phi_1} f]^\top \cdot \Delta, H\rangle = \langle \partial_{\phi_1} f \cdot H, \Delta\rangle = [a_1 \langle e_j, He_j\rangle + b_1 \langle e_j, He_l\rangle] \langle e_i(\phi_2^\top e_j)^\top, \Delta\rangle$$
$$+ [c_1 \langle e_j, He_l\rangle + d_1 \langle e_j, He_j\rangle] \langle e_i(\phi_2^\top e_l)^\top, \Delta\rangle$$
$$+ [a_2 \langle e_l, He_j\rangle + b_2 \langle e_l, He_l\rangle] \langle e_k(\phi_2^\top e_j)^\top, \Delta\rangle$$
$$+ [c_2 \langle e_l, He_l\rangle + d_2 \langle e_l, He_j\rangle] \langle e_k(\phi_2^\top e_l)^\top, \Delta\rangle$$

Since $\langle e_p, He_q\rangle = \langle e_p e_q^\top, H\rangle = \langle E_{p,q}, H\rangle$ for every $p, q$ it follows that

$$[\partial_{\phi_1} f]^\top \cdot \Delta = a_1 \langle e_i(\phi_2^\top e_j)^\top, \Delta\rangle E_{j,j} + b_1 \langle e_i(\phi_2^\top e_j)^\top, \Delta\rangle E_{j,l}$$
$$+ c_1 \langle e_i(\phi_2^\top e_l)^\top, \Delta\rangle E_{j,l} + d_1 \langle e_i(\phi_2^\top e_l)^\top, \Delta\rangle E_{j,j}$$
$$+ a_2 \langle e_k(\phi_2^\top e_j)^\top, \Delta\rangle E_{l,j} + b_2 \langle e_k(\phi_2^\top e_j)^\top, \Delta\rangle E_{l,l}$$
$$+ c_2 \langle e_k(\phi_2^\top e_l)^\top, \Delta\rangle E_{l,l} + d_2 \langle e_k(\phi_2^\top e_l)^\top, \Delta\rangle E_{l,j}.$$

By (56), we have $E_{j,j}, E_{l,l} \in \operatorname{span}_{X', \Delta'} \{[\partial_{\phi_1} f(\phi(\theta), X')]^\top \cdot \Delta'\}$ hence we obtain

$$\underbrace{\left[ b_1 \langle e_i(\phi_2^\top e_j)^\top, \Delta\rangle + c_1 \langle e_i(\phi_2^\top e_l)^\top, \Delta\rangle \right]}_{=:\alpha} E_{j,l}$$
$$+ \underbrace{\left[ a_2 \langle e_k(\phi_2^\top e_j)^\top, \Delta\rangle + d_2 \langle e_k(\phi_2^\top e_l)^\top, \Delta\rangle \right]}_{=:\beta} E_{l,j} \in \operatorname{span}_{X', \Delta'} \{[\partial_{\phi_1} f(\phi(\theta), X')]^\top \cdot \Delta'\}.$$

To conclude it is enough to exhibit a choice of $\Delta$ such that $\alpha \neq 0$ and $\beta = 0$: this will indeed imply $E_{j,l} \in \operatorname{span}_{X', \Delta'} \{[\partial_{\phi_1} f(\phi(\theta), X')]^\top \cdot \Delta'\}$, hence the the existence of $A \in \mathbb{R}^{\dim \times \dim}$ such that $\begin{pmatrix} E_{j,l} \\ A \end{pmatrix} \in \operatorname{span}_{X', \Delta'} \{\partial_\phi f(\phi(\theta), X')^\top \cdot \Delta'\}$, and by (55) one will obtain (60) as claimed.

To ensure $\beta = 0$ is it enough to have $\Delta \phi_2^\top e_j = e_i \perp e_k$ (possible as $k \neq i$), as well as $\Delta \phi_2^\top e_l \perp e_k, e_i$ (possible as $l \neq j$) Such a choice also implies $\alpha = b_1$, which is indeed nonzero.

*3d step: Conclusion.* Finally by combining (55) and (60), one obtains that

$$\operatorname*{span}_{X \in \mathbb{R}^{N \times \dim}, \ \Delta \in \mathbb{R}^{N \times \dim}} \{[\partial_\phi f(\phi(\theta), X)]^\top \cdot \Delta\} = \mathbb{R}^{\dim \times (2\dim)},$$

which concludes the proof. $\qquad \square$

Note that in the case when $N = 1$, then (18) writes

$$g(\theta, x) = XV^\top O, \tag{61}$$

with $\theta = (Q, K, V, O)$. In particular the parameters $Q$ and $K$ remain constant during (4), so $\theta \mapsto Q$ and $\theta \mapsto K$ are conservation laws with respect to any loss $\ell$. Moreover (61) coincides with a 2-layer linear network $\tilde{g}(\tilde{\theta}, x)$ with $\tilde{\theta} := (V, O)$, whose exact independent conservation laws with respect to any loss $\ell$ such that $\mathcal{V}_\ell = \mathbb{R}^n$ have been studied in Proposition 3.5, and are thus obtained via the set of conservation laws $\tilde{\theta} \mapsto VV^\top - OO^\top$.

## G. Proof of Corollaries 3.9 and 3.10 for attention layers

**Corollary 3.9.** *Under the assumptions of Theorem 3.8 for each $\theta \in \Theta_{\texttt{att}}$ such that both horizontally concatenated matrices $(Q, K)$ and $(V, O)$ have full rank, there is a neighborhood of $\theta$ in which all conservation laws for (18) are functions of*

$$QQ^\top - KK^\top, \quad \text{and} \quad VV^\top - OO^\top \quad \text{and vice-versa.}$$

*Proof.* We recall that $\phi(\theta) = (Q^\top K, V^\top O)$ and we denote $\theta_1 = (Q, K) \in \mathbb{R}^{d_1 \times \dim} \times \mathbb{R}^{d_1 \times \dim}$ and $\theta_2 = (V, O) \in \mathbb{R}^{d_1 \times \dim} \times \mathbb{R}^{d_1 \times \dim}$ and $\psi : (U, V) \in \mathbb{R}^{d_1 \times \dim} \times \mathbb{R}^{d_1 \times \dim} \mapsto U^\top V \in \mathbb{R}^{\dim \times \dim}$. Let us define: $\psi^1 : \theta \mapsto \psi(\theta_1)$ and $\psi^2 : \theta \mapsto \psi(\theta_2)$. Then $\phi(\theta) = (\psi^1(\theta), \psi^2(\theta))$, that is to say $\phi$ can be decoupled into two functions depending on two separate blocks of parameters. Its Jacobian and Hessian matrices are thus block-diagonal. Denoting $\phi_i$, $\psi_i^1$, $\psi_i^2$ the coordinate functions of $\phi$, $\psi^1$, $\psi^2$, the Lie bracket $[\nabla \psi_i^1(\cdot), \nabla \psi_j^2(\cdot)] \equiv 0$ thus vanishes for every $i = (i_1, i_2)$ and $j = (j_1, j_2) \in \{1, \dots, \dim\}^2$ and as a consequence

$$\text{Lie}(\text{span}\{\nabla \phi_k(\cdot) : k\}) = \text{Lie}(\text{span}\{\nabla \psi_i^1(\cdot) : i\} \oplus \text{Lie}(\text{span}\{\nabla \psi_i^2(\cdot) : i\}).$$

Since $\mathcal{V}_\ell = \mathbb{R}^n$, by Theorem 3.8 and Proposition 2.7 we also have $\mathbb{W}^{g,\ell} = \text{span}\{\nabla \phi_i(\cdot) : i\}$, and thus for any $\theta$:

$$\dim(\text{Lie}(\mathbb{W}^{g,\ell})(\theta)) = \dim(\text{Lie}(\text{span}\{\nabla \psi_i^1(\cdot) : i\})(\theta)) + \dim(\text{Lie}(\text{span}\{\nabla \psi_i^2(\cdot) : i\})(\theta)).$$

Let us consider $\theta$ such that both horizontally concatenated matrices $(Q, K)$ and $(V, O)$ have full rank.

As $(Q, K)$ (resp $(V, O)$) has full rank, this remains locally the case. By Proposition 4.3 of (Marcotte et al., 2023), the dimension of $\text{Lie}(\text{span}\{\nabla \psi_i^1(\cdot) : i\})(\theta)$ (resp. of $\text{Lie}(\text{span}\{\nabla \psi_i^2(\cdot) : i\})(\theta)$) is locally constant, denoted $d_1(\theta)$ (resp. $d_2(\theta)$). Then, by Theorem 2.11, the exact number of independent conservation laws is equal to $D - d_1(\theta) - d_2(\theta)$. Finally, by Proposition 3.5, all conservation laws are obtained by the set $QQ^\top - KK^\top$, $VV^\top - OO^\top$ of conservation laws. $\square$

**Corollary 3.10.** *For any $h = 1, \cdots, H$, the functions*

$$Q_h Q_h^\top - K_h K_h^\top, \quad V_h V_h^\top - O_h O_h^\top,$$

*define conservation laws for (19) with respect to any loss $\ell$.*

*Proof.* We recall that $H$ is the number of attention heads and define $\theta_h := (Q_h, K_h, V_h, O_h)$ for any $1 \leq h \leq H$ as well as $\phi^h : \theta \mapsto \phi(\theta_h)$, where $\phi(\theta_h) = (Q_h^\top K_h, V_h^\top O_h)$. Then, we have using Proposition 2.7:

$$\mathbb{W}^{g,\ell} \subseteq \text{span}\{\nabla \phi_j^h(\cdot) : h \in \{1, \cdots, H\}, j \in \{1, \cdots, 2\dim \times \dim\}\},$$

where $(\phi_j^h)_j$ denote the coordinate functions of $\phi^h$. In particular by Corollary 3.9 (we do not need to assume the non-degeneracy condition on $\theta$ as in Corollary 3.9 as we just use the direct inclusion via the reparametrization $\phi$), the functions $H_1^h : \theta \mapsto Q_h Q_h^\top - K_h K_h^\top$ and $H_2^h : \theta \mapsto V_h V_h^\top - O_h O_h^\top$ are conservation laws (for any $\ell$ by using Corollary 3.9 and Proposition 2.7), hence by Proposition 2.3 they satisfy for all $\theta \in \Theta$, for all $h, j$, $\nabla H_1^h(\theta), \nabla H_2^h(\theta) \perp \nabla \phi_j^h(\theta)$ (as $H_1^h$ and $H_2^h$ only depend on $\theta_h$). And thus $\nabla H_1^h(\theta), \nabla H_2^h(\theta) \perp \mathbb{W}^{g,\ell}(\theta) = \mathcal{W}_\theta^g$. By Proposition 2.3, we conclude that indeed $H_1^h$ and $H_2^h$ are conservation laws for (19) with respect to any loss $\ell$. $\square$

## H. Proof of Proposition 3.11 about cross-entropy / classification layers

**Proposition 3.11.** *With respect to any loss $\ell$ such that $\mathcal{V}_\ell = \mathbb{R}^n$, there are exactly $m$ independent conservation laws for the classification layer given by* (20): $h_j(\theta) := \sum_i \theta_{i,j}, j = 1, \ldots, m.$

*Proof.* Using Corollary 2.5 and straightforward calculus involving the Jacobian of the softmax (that are left to the reader) we compute

$$\mathbb{W}^{g,\ell}(\theta) = \underset{x \in \mathbb{R}^m, w \in \mathcal{V}_\ell = \mathbb{R}^n}{\mathrm{span}} \{\partial_\theta g(\theta, x)^\top w\} = \underset{x \in \mathbb{R}^m, w \in \mathbb{R}^n : \sum_i w_i = 0}{\mathrm{span}} \{\mathrm{vec}(wx^\top)\} = \{\mathrm{vec}(Z) : Z \in \mathbb{R}^{n \times m} : 1^\top Z = 0\}.$$

This space is independent of $\theta$, so that $\mathrm{Lie}(\mathbb{W}^{g,\ell})(\theta) = \mathbb{W}^{g,\ell}(\theta)$. By Theorem 2.11, since $\dim(\mathrm{Lie}(\mathbb{W}^{g,\ell})(\theta)) = nm - m$, there are exactly $m$ independent conservation laws. One verifies directly that for the functions $h_j$ defined in Prop. 3.11, we have $\langle \nabla h_j(\theta), Z \rangle = 0, \quad \forall Z \in \mathbb{W}^{g,\ell}(\theta)$. $\qquad \square$

## I. Proof of Proposition 4.3 on "block" conservation laws

**Proposition 4.3.** *Consider a function $h \in \mathcal{C}^1(\Theta, \mathbb{R})$ that only depends on the coordinates $\theta_T$, and for each $\theta \in \Theta$ denote*

$$\Theta_{T^c}(\theta_T) := \{\eta \in \mathbb{R}^{T^c} : (\theta_T, \eta) \in \Theta\} \tag{24}$$

*Consider a loss that satisfies the assumptions of Proposition 2.3 as well as Assumption 2.4. The function $h$ is a conservation law of $g$ with respect to $\ell$ if, and only if, for every $\theta \in \Theta$ one has $\nabla_{\theta_T} h(\theta) \perp R_{\theta_T}(\mathbb{W}^{g,\ell})$, where:*

$$R_{\theta_T}(\mathbb{W}^{g,\ell}) := \underset{\substack{\eta \in \Theta_{T^c}(\theta_T) \\ w \in \mathcal{V}_\ell}}{\mathrm{span}} \underset{x \in \mathcal{X}_{(\theta_T, \eta)}}{\mathrm{span}} \{\partial_{\theta_T} g((\theta_T, \eta), x)^\top w\}. \tag{25}$$

*Proof.* By Corollary 2.5 and Proposition 2.3, $h$ is a conservation law of $g$ if, and only if, for any $\theta = (\theta_T, \theta_{T^c}) \in \Theta$:

$$\nabla h(\theta) \perp \underset{x \in \mathcal{X}_\theta, w \in \mathcal{V}_\ell}{\mathrm{span}} \{\partial_\theta g(\theta, x)^\top w\}. \tag{62}$$

Since $h$ only depends on $\theta_T$, we have $\nabla h(\theta) = \begin{pmatrix} \nabla_{\theta_T} h(\theta) \\ \nabla_{\theta_{T^c}} h(\theta) \end{pmatrix} = \begin{pmatrix} \nabla_{\theta_T} h(\theta) \\ 0 \end{pmatrix}$ hence the above orthogonality is equivalent to:

$$\nabla_{\theta_T} h(\theta) \perp \underset{x \in \mathcal{X}_\theta, w \in \mathcal{V}_\ell}{\mathrm{span}} \{\partial_{\theta_T} g(\theta, x)^\top w\}, \quad \forall \theta \in \Theta \tag{63}$$

Given any $\theta \in \Theta$ and any $\eta \in \Theta_{T^c}(\theta_T)$, the vector $\theta' := (\theta_T, \eta) \in \Theta$ also satisfies $h(\theta') = h(\theta)$ and $\nabla_{\theta_T} h(\theta') = \nabla_{\theta_T} h(\theta)$, hence the orthogonality condition (63) also holds at $\theta'$. As a result,

$$\nabla_{\theta_T} h(\theta) \perp \underset{\eta \in \Theta_{T^c}(\theta_T)}{\mathrm{span}} \underset{(x,w) \in \mathcal{X}_{(\theta_T, \eta)} \times \mathcal{V}_\ell}{\mathrm{span}} \{\partial_{\theta_T} g((\theta_T, \eta), x)^\top w\} =: R_{\theta_T}(\mathbb{W}^{g,\ell}). \tag{64}$$

Conversely, if $h$ satisfies (64), then $h$ also satisfies (63), which is equivalent to (62) and implies that $h$ is a conservation law of $g$ with respect to $\ell$. $\qquad \square$

## J. About the invariances of the neural network

### J.1. Conservation laws and invariances of the shallow case

**Definition J.1** (Invariant transformation on the cost (1))**.** A (one-parameter) transformation on an open set $\Omega \subseteq \Theta \subseteq \mathbb{R}^D$ is a map $T : \mathbb{R} \times \Omega \to \mathbb{R}^D$ such that $T(\cdot, \theta)$ is differentiable for each $\theta \in \Omega$ and $T(0, \cdot) = \mathrm{id}$. This transformation leaves invariant the cost (1) if for all $\theta \in \Omega$ and for all $\epsilon \geq 0$, $L_Z(T(\epsilon, \theta)) = L_Z(T(0, \theta)) = L_Z(\theta)$. When this holds, simple calculus yields for every $\theta \in \Omega$:

$$\left\langle \nabla L_Z(\theta), \frac{\partial}{\partial \epsilon} T(\epsilon, \theta) \Big|_{\epsilon=0} \right\rangle = 0. \tag{65}$$

We denote $\Delta_T(\cdot) := \frac{\partial}{\partial \epsilon} T(\epsilon, \cdot)\big|_{\epsilon=0}$.

*From loss invariance to conservation laws.* In the context of a gradient flow dynamic (4), implies

$$\langle \dot{\theta}(t), \Delta_T(\theta(t)) \rangle = 0. \tag{66}$$

**Definition J.2** (conservation law obtained via an invariance). In particular, in light of Proposition 2.3, if a function $h \in \mathcal{C}^1(\Omega, \mathbb{R})$ is such that for all $\theta$ one has $\nabla h(\theta) := \Delta_T(\theta)$, then $h$ is a conservation law. We say in that case that $h$ is a conservation law of $g$ obtained via the invariant transformation $T$.

**The case of a 2-layer linear network**    Consider $\theta := (U, V) \in \mathbb{R}^{n \times r} \times \mathbb{R}^{m \times r}$ and define $T^A$ as the linear transformation

$$T^A(\epsilon, U, V) = \left( U \exp(\epsilon A), V \exp(-\epsilon A^\top) \right). \tag{67}$$

Simple calculus yields $\Delta_{T^A}(U, V) = (UA, -VA^\top)$. Considering $g(\theta, x) := UV^\top x$ and any $A \in \mathbb{R}^{r \times r}$, $T^A$ is an invariant transformation on (1): for all $\epsilon$ and $\theta$, $g(T^A(\epsilon, \theta), \cdot) = g(\theta, \cdot)$ hence $L_Z(T^A(\epsilon, \theta)) = L_Z(\theta)$. As a particular consequence, for gradient flows with linear networks, since (66) holds for $T = T^A$ with *any* matrix $A \in \mathbb{R}^{r \times r}$, denoting $\langle M, N \rangle := \text{Tr}(M^\top N)$ we obtain $0 = \langle \dot{U}(t), U(t)A \rangle - \langle \dot{V}(t), V(t)A \rangle$ at each time and for any $A$. Specializing this to any *symmetric* matrix, we obtain that $0 = \langle \dot{U}, UA \rangle - \langle \dot{V}, VA \rangle = 1/2\frac{d}{dt}(\langle U, UA \rangle - \langle V, VA \rangle)$. Thus for every symmetric matrix $A$, $\langle U, UA \rangle - \langle V, VA \rangle$ is conserved, which coincides with all conservation laws in that case (cf Proposition 3.5).

**The case of a 2-layer ReLU network**    Consider $\theta := (U, V, v) \in \mathbb{R}^{n \times r} \times \mathbb{R}^{m \times r} \times \mathbb{R}^{1 \times r}$ and define $T^A$ as the linear transformation

$$T^A(\epsilon, U, V, b) := \left( U \exp(\epsilon A), V \exp(-\epsilon A^\top), b \exp(-\epsilon A^\top) \right).$$

Considering $g(\theta, x) := U\sigma(V^\top x + b^\top)$ and any diagonal matrix $A \in \mathbb{R}^{r \times r}$, $T^A$ is a linear transformation that leaves (1) invariant. Moreover, $\Delta_{T^A}(U, V, b) = (UA, -VA^\top, -bA^\top) = (UA, -VA, -bA)$, as diagonal matrices are symmetric. By restricting ourselves to *elementary diagonal matrices* $A = E_{i,i}$ where $E_{i,i}$ is the one-hot matrix in $\mathbb{R}^{r \times r}$ with the $(i,i)$-th entry being 1, $i = 1, \cdots, r$, we obtain that for all $i$, $0 = \langle \dot{u}_i(t), u_i(t) \rangle - \langle \dot{v}_i(t), v_i(t) \rangle - \dot{b}_i(t)b_i(t) = \frac{1}{2}\frac{d}{dt}(\langle u_i, u_i \rangle - \langle v_i, v_i \rangle - b_i^2)$. Thus for all $i$, $\|u_i\|^2 - \|v_i\|^2 - b_i^2$ is conserved, recovering all conservation laws (cf Proposition 3.5).

**The case of a 2-layer ReLU convolutive network**    Consider $\theta := ((u_{k,j})_{k,j}, (v_{j,i})_{j,i})$ the collection of all filters and denote $G(\theta, x)$ the convolutive 2-layer ReLU neural network defined by (42). As explained in (43), we express (42) as a 2-layer ReLU neural network (15) as follows. Let us write for $j = 1, \cdots, c_1$, $u^{(j)} = \text{vec}((u_{k,j})_k) \in \mathbb{R}^{n_u c_2}$ and $v^{(j)} = \text{vec}((v_{j,i})_i) \in \mathbb{R}^{n_v c_0}$. For any $x \in \mathbb{R}^m$ one has:

$$G(\theta, x) = \underbrace{\left( \overline{C}_1(u^{(1)}) \quad \cdots \quad \overline{C}_1(u^{(c_1)}) \right)}_{=: \overline{U}} \sigma(\underbrace{\left( \overline{C}_2(v^{(1)}) \quad \cdots \quad \overline{C}_2(v^{(c_1)}) \right)}_{=: \overline{V}}^\top x),$$

where $\overline{C}_1$ (resp. $\overline{C}_2$) is a linear operator defined in (44) (resp. (45)). We rewrite $\theta = (U, V)$, where $U := \left( u^{(1)} \quad \cdots \quad u^{(c_1)} \right)$ and $V := \left( v^{(1)} \quad \cdots \quad v^{(c_1)} \right)$. For any $j = 1, \cdots, c_1$, we define $T_j$ the linear transformation

$$T_j(\epsilon, U, V) := \left( U \exp(\epsilon E_{j,j}), V \exp(-\epsilon E_{j,j}^\top) \right).$$

The transformation $T_j$ leaves (1) invariant: for all $\epsilon$ and $\theta$, $g(T_j(\epsilon, \theta), \cdot) = g(\theta, \cdot)$ as $\overline{C}_1$ and $\overline{C}_2$ are linear operators, hence $L_Z(T_j(\epsilon, \theta)) = L_Z(\theta)$. Moreover, $\Delta_{T_j}(U, V) = (UE_{j,j}, -VE_{j,j})$ and thus (66) writes: $0 = \langle \dot{u}^{(j)}(t), u^{(j)}(t) \rangle - \langle \dot{v}^{(j)}(t), v^{(j)}(t) \rangle = \frac{1}{2}\frac{d}{dt}(\langle u^{(j)}, u^{(j)} \rangle - \langle v^{(j)}, u^{(j)} \rangle)$. Thus for all $j$, $\langle u^{(j)}, u^{(j)} \rangle - \langle v^{(j)}, u^{(j)} \rangle$ is conserved, recovering all conservation laws (cf Theorem 3.6).

**The case of an attention-layer**    We treat the case of the network (18) in the exact same way as for a 2-layer linear network by considering for any symmetric matrix $A \in \mathbb{R}^{d_1 \times d_1}$ the linear transformations $T^A$ and $T'^A$ defined by

$$T^A(\epsilon, Q, K, V, O) = \left( \exp(\epsilon A)Q, \exp(-\epsilon A^\top)K, V, O \right),$$

and

$$T'^A(\epsilon, Q, K, V, O) = \big(Q, K, \exp(\epsilon A)V, \exp(-\epsilon A^\top)O)\big),$$

and that leave (1) invariant. We conclude in the exact same way as for the 2-layer linear neural network case and we obtain that for every symmetric matrix $A$, $\langle Q, AQ \rangle - \langle K, AK \rangle$ and $\langle V, AV \rangle - \langle O, AO \rangle$ are conserved, which coincides with all conservation laws in that case (cf Corollary 3.9).

## J.2. A conservation law of a sub-network obtained via an invariance gives a conservation law of the global network

**Proposition J.3.** *Assume that $\Theta = \Theta_1 \times \Theta_2 \times \Theta_3$ and that for every $\theta = (\theta_1, \theta_2, \theta_3) \in \Theta$ we can write $g(\theta, x) = g_1(\theta_1, g_2(\theta_2, g_3(\theta_3, x)))$. If there is a conservation law $h : \Theta_2 \mapsto \mathbb{R}$ of $g_2$ obtained via an invariance of $g_2$ (cf Definition J.2), then $H : \theta \in \Theta \mapsto h(\theta_2)$ is a conservation law of the global neural network $g$.*

*Proof.* We assume there exists a transformation $T : (\epsilon, \theta_2) \in \mathbb{R} \times \Theta_2 \mapsto T(\epsilon, \theta_2)$ that leaves $g_2$ invariant, i.e.

$$g_2(T(\epsilon, \theta_2), \cdot) = g_2(T(0, \theta_2), \cdot), \ \forall \epsilon, \theta_2.$$

Let $h : \theta_2 \in \Theta_2 \mapsto h(\theta_2) \in \mathbb{R}$ be a conservation law of $g_2$ obtained via the transformation $T$. This means that (cf Definition J.2)

$$\nabla h(\theta_2) = \Delta_T(\theta_2), \quad \forall \theta_2 \in \Theta_2. \tag{68}$$

We now define $\tilde{T} : (\epsilon, \theta_1, \theta_2, \theta_3) \in \mathbb{R}_+ \times \Theta \mapsto (\theta_1, T(\epsilon, \theta_2), \theta_3)$. As $T$ leaves invariant $g_2$, $\tilde{T}$ is a transformation that leaves the global neural network $g$ invariant: for all $\epsilon \in \mathbb{R}$ one has $g(T(\epsilon, \theta), x) = g(T(0, \theta), x)$. Thus, $\tilde{T}$ leaves (1) invariant too. Moreover by (68) we have for any $\theta \in \Theta$:

$$\Delta_{\tilde{T}}(\theta) = \begin{pmatrix} 0 \\ \Delta_T(\theta_2) \\ 0 \end{pmatrix} = \begin{pmatrix} 0 \\ \nabla h(\theta_2) \\ 0 \end{pmatrix}. \tag{69}$$

Let us show that $H : (\theta_1, \theta_2, \theta_3) \in \Theta_1 \times \Theta_2 \times \Theta_3 \mapsto h(\theta_2)$ is a conservation law of the global network $g$. Let us $\theta(t)$ a solution of (4). Then for all $t$ such that it holds:

$$
\begin{aligned}
\frac{\mathrm{d}}{\mathrm{d}t}(H(\theta(t)) &= \langle \dot{\theta}(t), \nabla H(\theta(t)) \rangle \\
&\stackrel{(4)}{=} \langle \nabla L_Z(\theta(t)), \nabla H(\theta(t)) \rangle \\
&= \left\langle \nabla L_Z(\theta(t)), \begin{pmatrix} 0 \\ \nabla h(\theta_2(t)) \\ 0 \end{pmatrix} \right\rangle \\
&\stackrel{(69)}{=} \langle \nabla L_Z(\theta(t)), \Delta_{\tilde{T}}(\theta(t)) \rangle = 0,
\end{aligned}
$$

as $\tilde{T}$ leaves (1) invariant. $\qquad \square$

# K. Proof of Lemma 4.4 on the density of $\mathcal{X}_\theta$

Remember that we consider a deep network $g(\theta, x)$ composed of residual blocks denoted $g_{\theta_l}^l(x)$, corresponding either to a block of a convolutive ResNet (21), a residual MLP (22) or an attention layer (23), see Section 4.2 for the notations.

**Lemma 4.4.** *Denote $\Theta = \Theta_q \times \ldots \times \Theta_1$ (or $\Theta = \Theta_{q+1} \times \Theta_q \times \ldots \times \Theta_1$ with $\Theta_{q+1} = \mathbb{R}^{n \times m}$ when there is a last softmax layer) where for each layer $1 \le l \le q$, $\Theta_l$ is the set of parameters $\theta_l$ such that*

1. *$g_{\theta_l}^l$ is an open map[4];*

2. *all the rows of the matrix $V^l$ (resp. $U_l'$) from (21) (resp. (22)) are nonzero in the convolutive ResNet case (resp. in the Transformer case).*

---

[4]*i.e.*, it sends an open set to an open set

*For every $\theta \in \Theta$ we have $\overline{\mathcal{X}_\theta} = \mathbb{R}^m$.*

In particular, for an attention layer (23), Lemma 4.4 only requires the first assumption Item 1.

*Proof.* First we show that, without loss of generality, it is sufficient to prove the result in the absence of a last softmax layer (20). This is a simple consequence of the fact that, in the presence of such a layer, since $g_{\theta_{q+1}}^{q+1}: z \mapsto \mathrm{softmax}(\theta_{q+1}z)$ is infinitely smooth everywhere with respect to $\theta_{q+1} \in \mathbb{R}^{n \times m}$, the set $\mathcal{X}_\theta$ (which we recall is defined as the collection of all data points $x \in \mathbb{R}^m$ such that $\theta \mapsto g(\theta, x)$ is $C^2$ in a neighborhood of $\theta$, cf Proposition 2.3) for the model $g_{\theta_{q+1}}^{q+1} \circ g_{\theta_q}^q \circ \cdots \circ g_{\theta_1}^1$ is the same as with the "truncated" neural network $g_\theta(x) = g_{\theta_q}^q \circ \cdots \circ g_{\theta_1}^1(x)$.

To prove the result without a softmax layer, we define $G_\theta^l(x) = g_{\theta_l}^l \circ \cdots \circ g_{\theta_1}^1(x)$ the output after $l$ blocks with the convention $G_\theta^0(x) = x$.

Our goal is to show that $\overline{\mathcal{X}_\theta} = \mathbb{R}^m$, and we proceed by contradiction, assuming that there exists $x_0$ and $r > 0$ such that for any $x \in B(x_0, r)$, there is no neighborhood of $\theta$ in which $\theta' \mapsto g_{\theta'}(x)$ is $C^2$.

Since each residual layer $g_{\theta_l}^l$ is either defined as a convolutive 2-layer ReLU network (21), a dense 2-layer ReLU network (22), or an attention layer (23), a straightforward induction on $q$ shows that given any $x$, the function $\theta \mapsto g_\theta(x)$ is infinitely smooth in a neighborhood of $\theta$ unless there is at least one residual layer $1 \leq l \leq q$ such that $\theta^l \mapsto g_{\theta_l}^l(G_\theta^{l-1}(x))$ fails to be infinitely smooth. Since attention layers (23) are always infinitely smooth, the lack of smoothness can only come from a 2-layer ReLU residual block, convolutive or not. The lack of smoothness implies that the pre-activation of at least one of the hidden neurons of this layer must be zero, i.e., $\langle V_l[j,:], G^{l-1}(x) \rangle = 0$ and where we denote $V_l$ either the matrix $V^l$ from (21) or the matrix $U_l'$ from (22). Similarly we denote $U_l$ either the matrix $U^l$ from (21) or the matrix $U_l$ from (22).

Thus, for any $x \in B(x_0, r)$, there exists $1 \leq l \leq q$ such that $g_{\theta_l}^l : x' \mapsto x' + U_l \sigma(V_l x')$ corresponds to the 2-layer ReLU network (21) in the convolutive ResNet case (resp. (22) in the Transformer case), and such that there exists $j$ satisfying $\langle V_l[j,:], G^{l-1}(x) \rangle = 0$.

In other words, we have proved that $B(x_0, r) \subseteq \bigcup_{l,j} \mathcal{H}_{l,j}$ where each of the finitely many sets

$$\mathcal{H}_{l,j} := \{x' \in \mathbb{R}^n : \langle V_l[j,:], G^{l-1}(x') \rangle = 0\}$$

is a closed set of empty interior, as we now show. Indeed $\mathcal{H}_{l,j}$ is closed as the reciprocal image of a closed set ($\{0\}$) by a continuous function. To show that $\mathcal{H}_{l,j}$ is of empty interior we proceed by contradiction, assuming that there exists a non-empty open set $B$ such that any $x' \in B$ satisfies $G^{l-1}(x') \in (\mathbb{R}V_l[j,:])^\perp \subsetneq \mathbb{R}^m$, as $V_l[j,:] \neq 0$ by Item 2 ($\theta \in \Theta$). But as $x' \in B \mapsto G^{l-1}(x') = g_{\theta_{l-1}}^{l-1} \circ \cdots \circ g_{\theta_1}^1(x)$ is open as composition of open maps (thanks to Item 1) and as $B$ is open, the set $G^{l-1}(B)$ is an open set in $\mathbb{R}^m$: this is absurd as $G^{l-1}(B) \subset (\mathbb{R}V_l[j,:])^\perp \subsetneq \mathbb{R}^m$.

Overall, as $B(x_0, r)$ is covered by a finite union of closed sets with empty interior, by Baire's theorem $B(x_0, r)$ also has empty interior: this is absurd, which concludes the proof. $\qquad\square$

## L. Discussion on the genericity of Item 2 in the assumptions of Lemma 4.4

We discuss here why Item 2 is a generic condition on the set of parameters, for the example of a residual block associated with a 2-layer ReLU network: :

$$g_{\theta_l}^l : x \mapsto x + U_l \sigma(V_l x),$$

where $U_l$ and $V_l$ are defined via $\theta_l$ as in (21) or (22). Since $g_{\theta_l}^l$ is a continuous, piecewise linear, and homogeneous function of $x$, we can write $g_{\theta_l}^l(x) = J_l(x) \times x$, where $J_l(x)$ represents the Jacobian matrix given by: $J_l(x) = U_l D_l(x) V_l$. Here, $D_l(x)$ denotes a diagonal matrix whose entries are binary values (either 0 or 1) depending on the activation of the neurons at $x$. We denote by $\mathcal{D}$ the set of all such diagonal matrices. By construction, $\mathcal{D}$ is finite. Thus the Jacobian $J_l(x)$ belongs to the finite set

$$\mathcal{J}(U_l, V_l) := \{I_m + U_l D V_l : D \in \mathcal{D}\}.$$

If each element of this set is invertible (i.e. $\mathcal{J}(U_l, V_l) \subseteq GL_m(\mathbb{R})$, a generic condition), then $g_{\theta_l}^l$ is an open map.

## M. Finding parameters for simplified neural network forms

**Lemma M.1.** *Let us consider consider some $1 \leq l \leq q$ and the associated $\Theta_l$ defined in Lemma 4.4 and $g_{\theta_l}^l$ that can be either an attention layer with a skip connection (23), or a (convolutive (21) or not (22)) MLP. There exists $\theta_l \in \Theta_l$ such that for any $x \in \mathbb{R}^m$ one has $g_{\theta_l}^l(x) = x$.*

*Proof.* **1st case: $g_{\theta_l}^l$ is defined by (21) (resp. (22)).** Then by considering $V^l$ (resp. $U_l'$) such that all its rows are nonzero (so that Item 2 is satisfied) (it is possible by taking $v_{j,i}^{(l)}$ non equal to zero as the matrices $Q_i$ are injective), and by considering $U^l = 0$ by taking all $u_{k,j}^{(l)}$ equal to zero (resp. $U_l = 0$), then for any $x \in \mathbb{R}^m$, $g_{\theta_l}^l(x) = x$ (and in particular $g_{\theta_l}^l$ is open, so Item 1 is satisfied and thus $\theta_l = (U^l, V^l)$ (resp. $\theta_l = (U_l, U_l')$ is in $\Theta_l$).

**2d case: $g_{\theta_l}^l$ is defined by (23).** Then by considering $\theta_l = (Q_l, K_l, V_l, O_l) = (0, 0, 0, 0)$ one has for any $x \in \mathbb{R}^m$, $g_{\theta_l}^l(x) = x$, and so $g_{\theta_l}^l$ is open, implying that $\theta_l \in \Theta_l$. $\qquad\square$

## N. Proof of Theorem 4.6 on block laws for natural residual blocks

**Theorem 4.6.** *With $\Theta$ as in Lemma 4.4, consider the l-th residual block of Example 4.1 (resp. Example 4.2), and denote $\theta_T := \theta_l$ and $\theta_{T^c}$ the parameters of all other residual blocks. A function $H : \theta = (\theta_T, \theta_{T^c}) \in \Theta \mapsto h(\theta_T)$ that only depends on $\theta_T$ is a conservation law of $g$ with respect to a loss $\ell$ such that $\mathcal{V}_\ell = \mathbb{R}^n$ if and only if $h$ is a conservation law of the shallow residual network $g^l(\theta_l, x) := g_{\theta_l}^l(x)$ with respect to the Euclidean loss. The same result holds for $\theta_T := \theta_{q+1}$ when considering a last block (20).*

*Proof.* **Case $l = q + 1$.** First we treat the case where $\theta_T := \theta_{q+1}$. In this case, one has for any $\theta \in \Theta$ and for any $x \in \mathbb{R}^m$:

$$g(\theta, x) = \text{softmax}(\theta_{q+1}(g_{\theta_q}^q \circ \cdots \circ g_{\theta_1}^1(x)) = g_{\theta_{q+1}}^{q+1}(g_{\theta_q}^q \circ \cdots \circ g_{\theta_1}^1(x)).$$

As a consequence

$$\partial_{\theta_T} g(\theta, x) = \partial_{\theta_{q+1}} g^{q+1}(\theta_{q+1}, g_{\theta_q}^q \circ \cdots \circ g_{\theta_1}^1(x)), \quad \forall \theta \in \Theta, \quad \forall x \in \mathcal{X}_{\theta_{q+1}} := \mathbb{R}^m. \tag{70}$$

Then by using Lemma M.1, one can choose $\theta_{T^c} \in \Theta_{T^c}$ such that $g_{\theta_q}^q \circ \cdots \circ g_{\theta_1}^1(x) = x$ for any $x \in \mathbb{R}^m$, and thus one obtains $\mathbb{R}^m = \{g_{\theta_q}^q \circ \cdots \circ g_{\theta_1}^1(x) : \theta_{T^c} \in \Theta_{T^c}, x \in \mathbb{R}^m\}$. Therefore one has:

$$\operatorname*{span}_{x \in \mathbb{R}^m}\{\partial_{\theta_{q+1}} g^{q+1}(\theta_{q+1}, x)^\top\} = \operatorname*{span}_{\theta_{T^c} \in \Theta_{T^c}, x \in \mathbb{R}^m}\{\partial_{\theta_T} g(\theta, x)^\top\}. \tag{71}$$

Since we assume $\mathcal{V}_\ell = \mathbb{R}^n$, one has

$$
\begin{aligned}
\mathcal{W}_{\theta_{q+1}}^{g^{q+1},\ell} &\overset{\text{Corollary 2.5}}{:=} \operatorname*{span}_{x \in \mathcal{X}_{\theta_{q+1}},\, w \in \mathbb{R}^n}\{\partial_{\theta_{q+1}} g^{q+1}(\theta_{q+1}, x)^\top w\} = \operatorname*{span}_{x \in \mathbb{R}^m, w \in \mathbb{R}^n}\{\partial_{\theta_{q+1}} g^{q+1}(\theta_{q+1}, x)^\top w\} \\
&\overset{(71)}{=} \operatorname*{span}_{\theta_{T^c} \in \Theta_{T^c}, w \in \mathbb{R}^n} \operatorname*{span}_{x \in \mathbb{R}^m}\{\partial_{\theta_T} g^\top(\theta, x)w\} \\
&\overset{\text{Lemma 4.4}}{=} \operatorname*{span}_{\theta_{T^c} \in \Theta_{T^c}, w \in \mathbb{R}^n} \overline{\operatorname*{span}_{x \in \mathcal{X}_\theta}\{\partial_{\theta_T} g^\top(\theta, x)w\}} \\
&= \operatorname*{span}_{\theta_{T^c} \in \Theta_{T^c}, w \in \mathbb{R}^n} \operatorname*{span}_{x \in \mathcal{X}_\theta}\{\partial_{\theta_T} g(\theta, x)^\top w\} =: R_{\theta_T}(\mathbb{W}^{g,\ell}),
\end{aligned}
$$

where we recall that $R_{\theta_T}(\mathbb{W}^{g,\ell})$ is defined in Proposition 4.3. This concludes the proof in the case $\theta_T := \theta_{q+1}$.

**Case $1 \leq l \leq q$.** We now assume that $\theta_T := \theta_l$ for some $l = 1, \cdots, q$. Let us consider $h(\theta_l)$ a conservation law for $g^l$. Then as all conservation laws of $g^l$ are obtained via the rescaling invariances of $g^l$ as discussed in Appendix J.1 and by using Proposition 3.2, the function $H : \theta \in \Theta \mapsto h(\theta_l)$ is a conservation law of $g$ by Proposition J.3.

We now show the converse result: considering a function $H : \theta \in \Theta \mapsto h(\theta_l)$ that is a conservation law of $g$ we wish to show that $h$ is a conservation law of $g^l$.

**A. We first consider the case without a last block** (20)**, ie the case where:**

$$g(\theta, x) = g^q_{\theta_q} \circ \cdots \circ g^1_{\theta_1}(x) \in \mathbb{R}^m. \tag{72}$$

It will be convenient to rewrite this as the composition of three functions depending on three separate blocks of coordinates of the variable $\theta = (\tilde{\theta}_1, \tilde{\theta}_2, \tilde{\theta}_3) \in \tilde{\Theta}_1 \times \tilde{\Theta}_2 \times \tilde{\Theta}_3$ (with $\tilde{\Theta}_2 := \Theta_l$, and $\tilde{\Theta}_1, \tilde{\Theta}_3$ given by Cartesian products of adequate $\Theta_j$, $j \neq l$), the middle one corresponding to $\tilde{\theta}_2 = \theta_l$:

$$g(\theta; x) = g_1(\tilde{\theta}_1; g_2(\tilde{\theta}_2; g_3(\tilde{\theta}_3; x))). \tag{73}$$

If $l = q$ (resp if $l = 1$) corresponds to the parameters of the last (resp. first) block, then we directly can write $g(\theta; x) = g_2(\tilde{\theta}_2; g_3(\tilde{\theta}_3; x))$ (resp. $g(\theta; x) = g_1(g_2(\tilde{\theta}_2; x)))$, but we will use the formulation (73) in all cases informally by simplicity.

With these notations our goal is to show that a conservation law $H$ for $g$ in (73) that only depends on $\tilde{\theta}_2$ is a conservation law for $g_2$. A technical step, that we slightly postpone, is to prove that there are $\tilde{\theta}_1, \tilde{\theta}_3 \in \tilde{\Theta}_1 \times \tilde{\Theta}_3$ such that for every $\tilde{\theta}_2 \in \tilde{\Theta}_2$,

$$g(\theta, x) = g_2(\tilde{\theta}_2, x), \quad \forall x \in \mathbb{R}^m \tag{74}$$

with $\theta := (\tilde{\theta}_1, \tilde{\theta}_2, \tilde{\theta}_3)$. Using this result we proceed using a density argument. Denoting $\mathcal{X}_{\tilde{\theta}_2}$ the set of all data points $x \in \mathbb{R}^m$ such that $\theta'_2 \mapsto g_2(\theta'_2, x)$ is $\mathcal{C}^2$ in the neighborhood of $\tilde{\theta}_2$, we observe that $\mathcal{X}_\theta \subseteq \mathcal{X}_{\tilde{\theta}_2}$ with $\mathcal{X}_\theta$ defined as in Proposition 2.3. (Indeed if $x \in \mathcal{X}_\theta$, then $\theta' \mapsto g(\theta', x)$ is $\mathcal{C}^2$ in the neighborhood of $\theta := (\tilde{\theta}_1, \tilde{\theta}_2, \tilde{\theta}_3)$, so in particular $\theta'_2 \mapsto g((\tilde{\theta}_1, \theta'_2, \tilde{\theta}_3), x) = g_2(\theta'_2, x)$ is $\mathcal{C}^2$ in the neighborhood of $\tilde{\theta}_2$.) By (74), for any $\tilde{\theta}_2 \in \tilde{\Theta}_2$ and any $x \in \mathcal{X}_{\tilde{\theta}_2}$, $\partial_{\tilde{\theta}_2} g(\theta; x) = \partial_{\tilde{\theta}_2} g_2(\tilde{\theta}_2; x)$. Moreover $x \mapsto \partial_{\tilde{\theta}_2} g(\theta, x) (= \partial_{\tilde{\theta}_2} g_2(\tilde{\theta}_2, x))$ is continuous on $\mathcal{X}_{\tilde{\theta}_2}$. Indeed either $\mathcal{X}_{\tilde{\theta}_2} = \mathbb{R}^m$ (when $g_2$ is an attention layer) or $\mathcal{X}_{\tilde{\theta}_2}$ is the whole space outside a finite number of hyperplanes associated to the zeroes of the activation of each hidden neuron: in particular if $x \in \mathcal{X}_{\tilde{\theta}_2}$, then $g_2(\tilde{\theta}_2, x)$ does not have any activation that vanishes, so it remains the case in a neighborhood $B(x, r)$ of $x$, this implies in particular that $x' \in B(x, r) \mapsto g_2(\tilde{\theta}_2, x')$ is continuous. Finally, as $\overline{\mathcal{X}_\theta} = \mathbb{R}^n$ by Lemma 4.4, for any $x \in \mathcal{X}_{\tilde{\theta}_2}$ there exist $(x_N) \in (\mathcal{X}_\theta)^\mathbb{N}$ such that $x_N \longrightarrow x$. Thus, as $\mathcal{X}_\theta \subseteq \mathcal{X}_{\tilde{\theta}_2}$ and by continuity of $\partial_{\tilde{\theta}_2} g_2(\theta; x)$ on $\mathcal{X}_{\tilde{\theta}_2}$.

$$\partial_{\tilde{\theta}_2} g(\theta; x_N) = \partial_{\tilde{\theta}_2} g_2(\theta; x_N) \longrightarrow \partial_{\tilde{\theta}_2} g_2(\theta; x),$$

Since $\mathcal{V}_\ell = \mathbb{R}^m$ with $\ell$ the Euclidean loss in $\mathbb{R}^m$, we obtain with the considered triplet of parameter $\theta = (\tilde{\theta}_1, \tilde{\theta}_2, \tilde{\theta}_3)$

$$
\begin{aligned}
\mathcal{W}^{g_2, \ell}_{\tilde{\theta}_2} \overset{\text{Corollary 2.5}}{:=} & \underset{w \in \mathbb{R}^m}{\text{span}} \underset{x \in \mathcal{X}_{\tilde{\theta}_2}}{\text{span}} \{\partial_{\tilde{\theta}_2} g_2^\top(\tilde{\theta}_2; x) w\} \\
\subseteq & \underset{w \in \mathbb{R}^m}{\text{span}} \overline{\underset{x \in \mathcal{X}_\theta}{\text{span}} \{\partial_{\tilde{\theta}_2} g(\theta; x)^\top w\}} \quad (\text{as } \mathcal{X}_{\tilde{\theta}_2} \subseteq \mathbb{R}^m = \overline{\mathcal{X}_\theta}) \\
= & \underset{w \in \mathbb{R}^m}{\text{span}} \underset{x \in \mathcal{X}_\theta}{\text{span}} \{\partial_{\tilde{\theta}_2} g(\theta; x)^\top w\} \quad (\text{as every finite-dimensional space is closed}) \\
\subseteq & \underset{\theta'_1 \in \tilde{\Theta}_1, \theta'_3 \in \tilde{\Theta}_3, w \in \mathbb{R}^m}{\text{span}} \underset{x \in \mathcal{X}_{(\theta'_1, \tilde{\theta}_2, \theta'_2)}}{\text{span}} \{\partial_{\tilde{\theta}_2} g(\theta; x)^\top w\} =: R_{\tilde{\theta}_2}(\mathbb{W}^{g, \ell}).
\end{aligned}
$$

where we recall again that $R_{\theta_T}(\mathbb{W}^{g, \ell})$ is defined in Proposition 4.3. Thus by Proposition 4.3, if a function $H(\theta) = h(\tilde{\theta}_2)$ that only depends on $\tilde{\theta}_2$ is conserved for $g$ with respect to the Euclidean loss, then for every $\tilde{\theta}_2 \in \tilde{\Theta}_2$ the gradient $\nabla_{\tilde{\theta}_2} H(\theta) = \nabla h(\tilde{\theta}_2)$ is orthogonal to $R_{\tilde{\theta}_2}(\mathbb{W}^{g, \ell}) = \mathcal{W}^{g_2, \ell}_{\tilde{\theta}_2}$, hence by Proposition 2.3 $h$ is also conserved for $g_2$ with respect to the Euclidean loss.

**B. We finally consider the case with a last block** (20)**, ie the case where the neural network writes:**

$$g_\theta : x \in \mathbb{R}^m \mapsto g(\theta, x) = g^{q+1}_{\theta_{q+1}} \circ g^q_{\theta_q} \circ \cdots \circ g^1_{\theta_1}(x) \in \mathbb{R}^n.$$

We recall that $l \neq q + 1$. Denoting $\tilde{\theta} = (\theta_1, \cdots, \theta_q)$ and $\tilde{g}(\tilde{\theta}, x) = \tilde{g}_{\tilde{\theta}}$ the network (72), the analysis conducted in A above shows that there are $\tilde{\theta}_1 \in \tilde{\Theta}_1, \tilde{\theta}_3 \in \tilde{\Theta}_3$ such that for every $\tilde{\theta}_2 \in \tilde{\Theta}_2$ the parameter $\tilde{\theta} := (\tilde{\theta}_1, \tilde{\theta}_2, \tilde{\theta}_3)$ satisfies:

$$\tilde{g}(\tilde{\theta}, x) = g_2(\tilde{\theta}_2, x), \quad \forall x \in \mathbb{R}^m.$$

so that with $\theta = (\tilde{\theta}, \theta_{q+1})$ we have

$$g(\theta, x) = \text{softmax}(\theta_{q+1}\tilde{g}(\tilde{\theta}, x)) = \text{softmax}(\theta_{q+1}g_2(\tilde{\theta}_2, x)), \quad \forall x \in \mathbb{R}^m$$

As a result, reasoning as in A above, for any $x \in \mathcal{X}_{\tilde{\theta}_2}$ one has:

$$\partial_{\tilde{\theta}_2} g(\theta, x) = \partial\text{softmax}(\theta_{q+1}\tilde{g}(\tilde{\theta}, x))\theta_{q+1}\partial_{\tilde{\theta}_2} g_2(\tilde{\theta}_2, x).$$

Moreover we have

$$\underset{\theta_{q+1}}{\text{span}} \underset{x \in \mathcal{X}_{\tilde{\theta}_2}, w \in \mathbb{R}^n}{\text{span}} \left\{ \left( \partial\text{softmax}(\theta_{q+1}\tilde{g}(\tilde{\theta}, x))\theta_{q+1}\partial_{\tilde{\theta}_2} g_2(\tilde{\theta}_2, x) \right)^\top w \right\}$$

$$= \underset{\theta_{q+1}}{\text{span}} \underset{x \in \mathcal{X}_{\tilde{\theta}_2}, w \in \mathbb{R}^n}{\text{span}} \left\{ \partial_{\tilde{\theta}_2} g_2(\tilde{\theta}_2, x)^\top \theta_{q+1}^\top [\partial\text{softmax}(\theta_{q+1}\tilde{g}(\tilde{\theta}, x))]^\top w \right\}$$

$$= \underset{x \in \mathcal{X}_{\tilde{\theta}_2}}{\text{span}} \left\{ \partial_{\tilde{\theta}_2} g_2(\tilde{\theta}_2, x)^\top \underset{\theta_{q+1}, w \in \mathbb{R}^n}{\text{span}} \left\{ \theta_{q+1}^\top [\partial\text{softmax}(\theta_{q+1}\tilde{g}(\tilde{\theta}, x))]^\top w \right\} \right\}$$

$$= \underset{w' \in \mathbb{R}^m}{\text{span}} \underset{x \in \mathcal{X}_{\tilde{\theta}_2}}{\text{span}} \{\partial_{\tilde{\theta}_2} g_2^\top(\tilde{\theta}_2; x)w'\} =: \mathcal{W}_{\tilde{\theta}_2}^{g_2, \ell},$$

In the last line we used that

$$\underset{\theta_{q+1}, w \in \mathbb{R}^n}{\text{span}} \{\theta_{q+1}^\top [\partial\text{softmax}(\theta_{q+1}\tilde{g}(\tilde{\theta}, x))]^\top w\} = \mathbb{R}^m, \tag{75}$$

a property that we now show. Indeed, given any $y \in \mathbb{R}^m$ consider $\theta_{q+1} := e_1 y^\top$, where $e_1 \in \mathbb{R}^n$ the first canonical vector. Simple calculus yields $\theta_{q+1}^\top [\partial\text{softmax}(\theta_{q+1}\tilde{g}(\tilde{\theta}, x))]^\top = yz^\top$, where

$$z^\top = \left( \frac{\exp(\lambda)(n-1)}{(\exp(\lambda) + n - 1)^2}, -\frac{\exp(\lambda)}{(\exp(\lambda) + n - 1)^2}, \cdots, -\frac{\exp(\lambda)}{(\exp(\lambda) + n - 1)^2} \right) \neq 0, \quad \text{with } \lambda := \langle y, \tilde{g}(\tilde{\theta}, x) \rangle.$$

With $w := z/\|z\|_2^2$, we get $\theta_{q+1}^\top [\partial\text{softmax}(\theta_{q+1}\tilde{g}(\tilde{\theta}, x))]^\top w = y$. Since this holds for any choice of $y$, we get (75).

Finally, we can use the exact same previous proof by using Lemma 4.4 and conclude the proof.

**C. We finally prove** (74). By Lemma M.1, one can choose for all layer $p \neq l$ with $p \leq q$ some parameters $\theta_p$ such that for any $x \in \mathbb{R}^m$ one has $g_{\theta_p}^p(x) = x$. Thus with $\tilde{\theta}_1 := (\theta_{l+1}, \cdots, \theta_q) \in \tilde{\Theta}_1$ and $\tilde{\theta}_3 = (\theta_1, \cdots, \theta_{l-1}) \in \tilde{\Theta}_3$, one has $g_1(\tilde{\theta}_1, x) = x$ and $g_3(\tilde{\theta}_3, x) = x$ for any $x \in \mathbb{R}^m$. Finally one has $g(\theta, x) = g_2(\tilde{\theta}_2, x)$ for all $x \in \mathbb{R}^m$ which concludes the proof.

$\square$

## O. Additional figures

We illustrate the notion of blocks overlapping or not a residual connexion in Figure 3-Figure 2, and display experimental results on conserved functions during ResNet training in Figure 4 and during Transformer training in Figure 5.

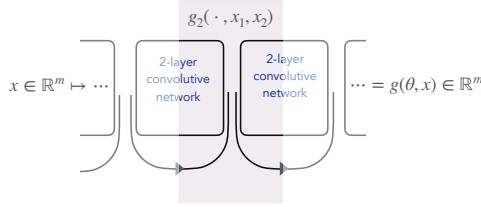

*Figure 2.* Block overlapping a residual connection

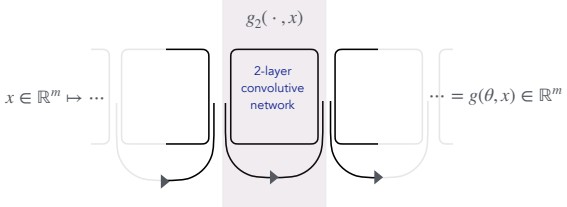

*Figure 3.* Block sharing a residual connection

## P. Proof of Theorem 4.7 : no block conservation laws for blocks overlapping a residual connexion

**Theorem 4.7.** *Consider a layer index* $1 \leq l \leq q - 1$ *and* $\Theta$ *defined as in Lemma 4.4 with the exception that for each* $\theta_{l+1} \in \Theta_{l+1}$, *we further require that the rows of* $V^{l+1}$ *are pairwise non-colinear. If* $n_v = p$ *then any conservation law of* $g$ *with respect to the Euclidean loss that only depends on* $\tilde{\theta}_2$ *is a constant function.*

While Theorem 4.7 is stated in the main text for the specific case of ResNets with residual blocks that involve a 2-layer ReLU network with matrices $U, V$ that are either dense or with a convolutive structure, we actually prove a more general version where these matrices have the generic structure associated to (42), with $C_1, C_2$ satisfying Assumption E.1 and two additional hypotheses on the matrices $P_i$ and $Q_i$ from Assumption E.1.

**Assumption P.1.** First we assume that $\bigcap \text{range}(Q_i) \neq \{0\}$, where $Q_i$ is defined in Assumption E.1.

**Link with the convolutive case.** Assumption P.1 (as well as Assumption E.1) is satisfied for the matrices $Q_i$ defined just after Assumption E.1, that are associated to the convolutive model (16) when $n_v = p$. Indeed in this setting, all $Q_i$ correspond to full circular shift operators, and notably $\mathbf{1} = Q_i \mathbf{1}$, where $\mathbf{1}$ denotes the all-ones vector.

*Proof of Theorem 4.7.* Here $\tilde{\theta}_2 = \theta_T = (V^{l+1}, U^l)$ corresponds to two consecutive parameter blocks before and after a skip connection and that can be written as the composition of $g_1$, $g_2$ and $g_3$ where

$$\begin{pmatrix} y_1 \\ y_2 \end{pmatrix} = g_2\left( (V^{l+1}, U^l); \begin{pmatrix} x_1 \\ x_2 \end{pmatrix} \right) := \begin{pmatrix} V^{l+1} \\ I_d \end{pmatrix} \begin{pmatrix} U^l & I_d \end{pmatrix} \begin{pmatrix} \sigma \\ id \end{pmatrix} \begin{pmatrix} x_1 \\ x_2 \end{pmatrix}, \tag{76}$$

$$\begin{pmatrix} x_1 \\ x_2 \end{pmatrix} = g_3(\tilde{\theta}_3, x) = \begin{pmatrix} V^l \\ I_d \end{pmatrix} (g_{\theta_{l-1}}^{l-1} \circ \ldots \circ g_{\theta_1}^1)(x), \tag{77}$$

$$\text{and } g_1\left( \tilde{\theta}_1, \begin{pmatrix} y_1 \\ y_2 \end{pmatrix} \right) = (g_{\theta_q}^q \circ \ldots g_{\theta_{l+2}}^{l+2}) \left( \begin{pmatrix} U^{l+1} & I_d \end{pmatrix} \begin{pmatrix} \sigma \\ id \end{pmatrix} \begin{pmatrix} y_1 \\ y_2 \end{pmatrix} \right). \tag{78}$$

The parameters $\tilde{\theta}_1, \tilde{\theta}_3$ gather all relevant parameters involved in the definitions of $g_1$ and $g_3$. We define $d_1$, $d_2$ such that $V^{l+1} \in \mathbb{R}^{d_1 \times n}, U^l \in \mathbb{R}^{n \times d_2}$ and will painlessly alternate between matrix and vector representations (in $\mathbb{R}^{n d_i}$) of such parameters. The core of the proof is to show that

$$R_{\theta_T}(\mathbb{W}^{g_2, \ell}) := \underset{\eta = (\tilde{\theta}_1, \tilde{\theta}_3) \in \Theta_{T^c}(\theta_T)}{\text{span}} \underset{(x,w) \in \mathcal{X}_{(\theta_T, \eta)} \times \mathbb{R}^n}{\text{span}} \{ \partial_{\theta_T} g^\top ((\theta_T, \eta); x) w \} = \mathbb{R}^{(d_1 + d_2)n} \tag{79}$$

with $\ell$ the Euclidean loss. By Proposition 4.3 this will imply as claimed that the only conservation laws $h$ of $g$ with respect to the Euclidean loss that only depend on $\theta_T$ are the constant ones.

To prove (79) we proceed in two steps, proving separately

$$\{0_{d_1 n}\} \times \mathbb{R}^{d_2 n} \subseteq R_{\theta_T}(\mathbb{W}^{g_2, \ell}) \tag{80}$$

$$\mathbb{R}^{d_1 n} \times \{0_{d_2 n}\} \subseteq R_{\theta_T}(\mathbb{W}^{g_2, \ell}). \tag{81}$$

By the definition of $R_{\theta_T}(\mathbb{W}^{g_2, \ell}) \subseteq \mathbb{R}^{(d_1 + d_2)n}$ as a linear span, this indeed implies (79).

As a warmup, let us characterize $\Theta_{T^c}(\theta_T) := \{\eta \in \mathbb{R}^{T^c} : (\theta_T, \eta) \in \Theta\}$ as defined in (24), where we recall that $\Theta = \Theta_q \times \cdots \times \Theta_1$ with the $\Theta_i$ defined in Lemma 4.4, and where $\Theta_{l+1}$ is also such that $V^{l+1}$ has all its rows that define distinct hyperplanes. One then has $\Theta_{T^c}(\theta_T) = \tilde{\Theta}_1 \times \tilde{\Theta}_3$, where

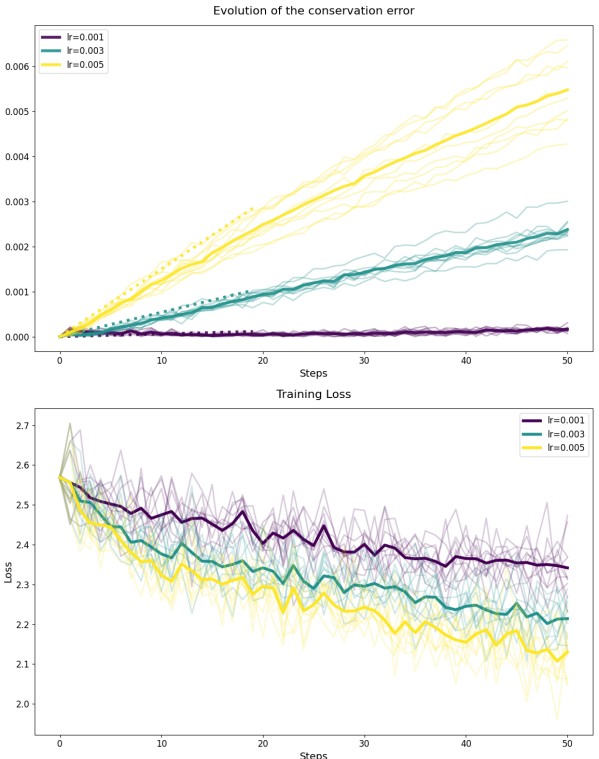

*Figure 4.* Tracking a conserved function and the loss during ResNet-18 training on CIFAR-10. For each learning rate between 1e-3 and 5e-3, we train 10 models for 50 steps using SGD without momentum or weight decay, with 10 different random seeds. For each configuration, we record both the loss evolution (bottom) and the evolution of the conservation error $\left| \frac{h(\theta_k) - h(\theta_0)}{h(\theta_0)} \right|$ (top) with $h_j$ defined in (17), where $\theta_T$ represents the parameters associated with the first residual block. The dotted lines show the theoretical slopes derived from bound (27), which scale quadratically with $\tau$ as $C\tau^2$. The empirical results confirm that the function is approximately conserved and that the slope coefficient maintains proportionality with $\tau^2$. For non-conserved functions, the evolution is in $\tau$ rather than $\tau^2$, since the first-order term in the Taylor expansion (used in the proof of Proposition 5.1) does not vanish in that case.

- $\tilde{\Theta}_1 := \Theta_q \times \ldots \times \Theta_{l+2} \times \hat{\Theta}_{l+1}$ with $\hat{\Theta}_{l+1} := \{U^{l+1} \in \mathbb{R}^{n \times d_1} : (U^{l+1}, V^{l+1}) \in \Theta_{l+1}\}$;

- $\tilde{\Theta}_3 := \hat{\Theta}_l \times \Theta_{l-1} \times \ldots \times \Theta_1$ with $\hat{\Theta}_l := \{V^l \in \mathbb{R}^{d_2 \times n} : (U^l, V^l) \in \Theta_l\}$.

It will also be useful to denote $\overline{\Theta}_1$ (resp. $\overline{\Theta}_3$) the set of all $\tilde{\theta}_1$ (resp. of all $\tilde{\theta}_3$) such that for each $k \geq l + 2$ (resp. all $k \leq l - 1$) we have: a) $U^k = 0$ (by taking all $u_{k',j}^{(k)}$ equal to zero), so that $(U^k, V^k)$ satisfies Item 1 ; and b) $V^k$ satisfies Item 2 of Lemma 4.4 (possible by taking $v_{j,i}^{(k)}$ non equal to zero as the matrices $Q_i$ are injective).

For every $\tilde{\theta}_1 \in \overline{\Theta}_1$ and $\tilde{\theta}_3 \in \overline{\Theta}_3$ we have

$$g_1\left(\tilde{\theta}_1; \begin{pmatrix} y_1 \\ y_2 \end{pmatrix}\right) = U^{l+1}\sigma(y_1) + y_2. \tag{82}$$

$$g_3\left(\tilde{\theta}_3; x\right) = \begin{pmatrix} V^l x \\ x \end{pmatrix}. \tag{83}$$

With the above notations, we highlight that if $\tilde{\theta}_1 \in \overline{\Theta}_1$ is such that $U^{l+1} \in \hat{\Theta}_{l+1}$ then $\tilde{\theta}_1 \in \tilde{\Theta}_1$, and if $\tilde{\theta}_3 \in \overline{\Theta}_3$ is such that $V^l \in \hat{\Theta}_l$ then $\tilde{\theta}_3 \in \tilde{\Theta}_3$. In particular such a choice of $\tilde{\theta}_1$ and $\tilde{\theta}_3$ implies that $\eta = (\tilde{\theta}_1, \tilde{\theta}_3) \in \Theta_{T^c}(\theta_T)$. In the rest of the proof we consider $\tilde{\theta}_1, \tilde{\theta}_3$ with these properties, this will enable us to leverage (82)-(83).

**1st step: proof of (80).**

In that part, we consider $\tilde{\theta}_1 \in \overline{\Theta}_1$ such that $U^{l+1} = 0$, and thus (cf (21)-(22)-(23) ) we have $g_{\theta_{l+1}}^{l+1} = \mathtt{id}$ and the block

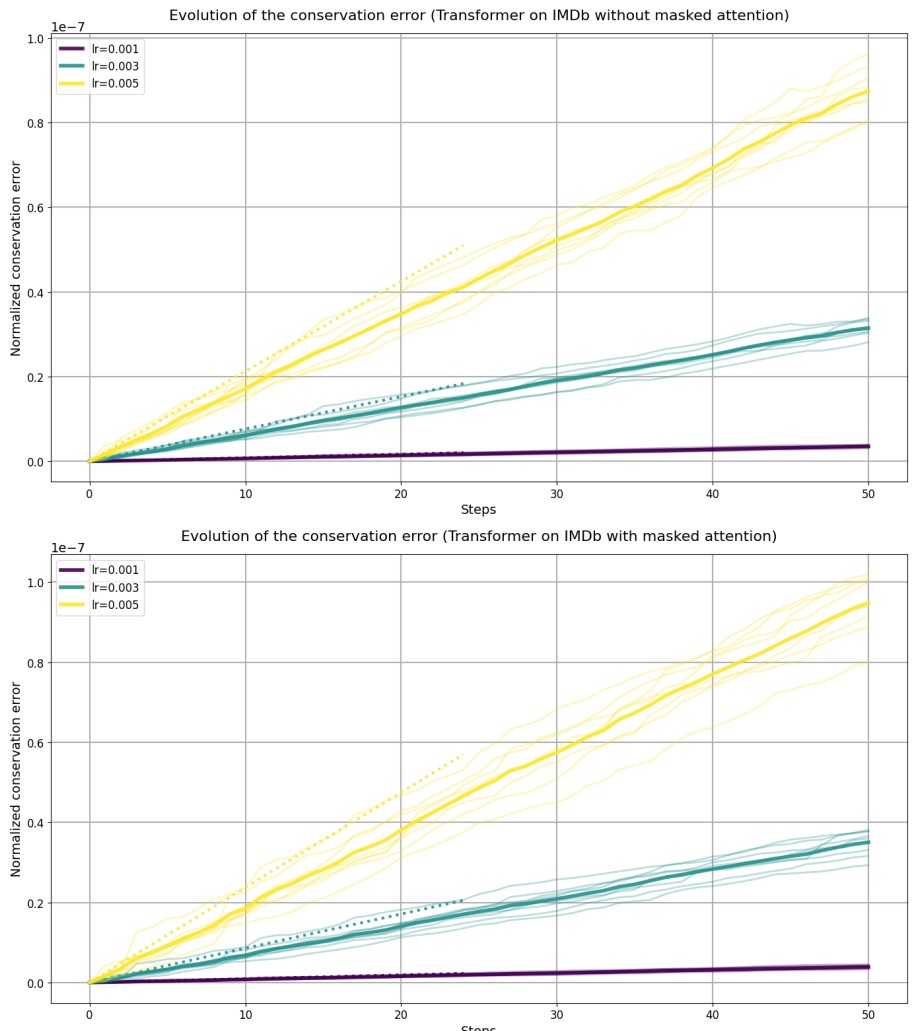

*Figure 5.* Tracking a conserved function and the loss during a Transformer training on IMDb dataset. For each learning rate between 1e-3 and 5e-3, we train 8 models for 50 steps using SGD without momentum or weight decay, with 8 different random seeds. For each configuration, we record both the evolution of the Frobenius norm of the conserved matrix identified in Corollary 3.10, specifically examining the query and key matrices from the first attention head in the first layer where masking is applied (bottom) or not (top). The dotted lines show the theoretical slopes derived from bound (27), which scale quadratically with $\tau$ as $C\tau^2$. The empirical results confirm that the function is approximately conserved and that the slope coefficient maintains proportionality with $\tau^2$.

parameter $\theta_{l+1} = (U^{l+1}, V^{l+1})$ satisfies Item 1 of Lemma 4.4 (i.e. $U^{l+1} \in \hat{\Theta}_{l+1}$ and thus $\tilde{\theta}_1 \in \tilde{\Theta}_1$). Since $U^{l+1} = 0$, combining (82)-(83) with the expression of $g_2$ (76) we have

$$g(\theta; x) = U^l \sigma(V^l x) + x. \tag{84}$$

Since the above expression of $g$ is independent of $V^{l+1}$, and $\theta_T = \tilde{\theta}_2 = (V^{l+1}, U^l) \in \mathbb{R}^{d_1 \times n} \times \mathbb{R}^{n \times d_2}$, for any $(H, K) \in \mathbb{R}^{d_1 \times n} \times \mathbb{R}^{n \times d_2}$, any $w \in \mathbb{R}^n$ and any $x \in \mathbb{R}^n$, at $\theta = (\tilde{\theta}_1, \tilde{\theta}_2, \tilde{\theta}_3) = (\theta_T, \theta_{T^c})$ we have:

$$\langle \partial_{\theta_T} g(\theta; x) \cdot (H, K), w \rangle = \langle K\sigma(V^l x), w \rangle = \langle (H, K), (0, w\sigma(V^l x)^\top) \rangle.$$

that is to say

$$\partial_{\theta_T} g(\theta; x)^\top w = \left( \begin{smallmatrix} 0 \\ \mathtt{vec}(w\sigma(V^l x)^\top) \end{smallmatrix} \right).$$

In light of the definition (79) of $R_{\theta_T}(\mathbb{W}^{g_2, \ell})$, since $\eta = (\tilde{\theta}_1, \tilde{\theta}_3) \in \Theta_{T^c}(\theta_T)$ for each of the parameters $\tilde{\theta}_1, \tilde{\theta}_3$ considered in this part of the proof, a sufficient condition to establish (80) is thus

$$\operatorname*{span}_{V^l \in \hat{\Theta}_l, w \in \mathbb{R}^n, x \in \mathcal{X}_\theta} \mathtt{vec}(w\sigma(V^l x)^\top) = \mathbb{R}^{d_2 n}$$

or equivalently (by simple linear algebra)

$$\operatorname*{span}_{V^l \in \hat{\Theta}_l, x \in \mathcal{X}_\theta} \sigma(V^l x) = \mathbb{R}^{d_2}.$$

For each $V^l \in \hat{\Theta}_l$, by definition the corresponding $\theta = (\tilde{\theta}_1, \theta_T, \tilde{\theta}_3)$ belongs to $\Theta$ hence by Lemma 4.4 $\mathcal{X}_\theta$ is dense. Since $x \mapsto \sigma(V^l x)$ is continuous, the above condition is also equivalent to

$$\operatorname*{span}_{V^l \in \hat{\Theta}_l, x \in \mathbb{R}^m} \sigma(V^l x) = \mathbb{R}^{d_2} \tag{85}$$

To prove (85) we simply exhibit for each canonical vector $e_i \in \mathbb{R}^{d_2}$ a matrix $V^l \in \hat{\Theta}_l$ and $x \in \mathbb{R}^m$ such that $\sigma(V^l x) = e_i$. For this simply consider $\lambda > 0$, $x = \lambda e_1 \in \mathbb{R}^m$, $V$ a matrix with its first column equal to $e_i$, and $V^l = V/\lambda$. If $\lambda > 0$ is large enough then one can easily check that $(U^l, V^l) \in \Theta_l$, hence $V^l \in \hat{\Theta}_l$ as claimed.

**2d step: proof of (81).**

We denote $v_1, \cdots, v_{d_1}$ the $d_1$ rows of the matrix $V^{l+1} \in \mathbb{R}^{d_1 \times n}$, and for each $j = 1, \cdots, d_1$ we denote

$$\mathcal{H}_j := \{x \in \mathbb{R}^n : v_j^\top x = 0\}.$$

Combining (82)-(83) with the expression of $g_2$ (76) we have for any $V^l$:

$$g(\theta; x) = U^{l+1}\sigma(V^{l+1} x) + x, \quad \text{for each } x \text{ such that } \sigma(V^l x) = 0. \tag{86}$$

As in the first step, since this expression of $g$ is independent of $U^l$, and as $\theta_T = \tilde{\theta}_2 = (V^{l+1}, U^l) \in \mathbb{R}^{d_1 \times n} \times \mathbb{R}^{n \times d_2}$, we obtain that for any $V^l$ and any $w \in \mathbb{R}^n$, at $\theta = (\tilde{\theta}_1, \tilde{\theta}_2, \tilde{\theta}_3) = (\theta_T, \theta_{T^c})$, denoting $D(x) := \operatorname{diag}((\mathbf{1}_{\langle v_j, x \rangle > 0})_j)$,

$$[\partial_{\theta_T} g(\theta; x)]^\top w = \left( \begin{smallmatrix} \mathtt{vec}(D(x)U^{l+1\top} wx^\top) \\ 0 \end{smallmatrix} \right), \quad \text{for each } x \in \mathbb{R}^n - \cup \mathcal{H}_j \text{ such that } \sigma(V^l x) = 0. \tag{87}$$

The claim (81) is a direct consequence of the following two inclusions that we prove below

$$\operatorname*{span}_{\tilde{\theta}_1 \in \tilde{\Theta}_1} \operatorname*{span}_{\substack{w \in \mathbb{R}^n \\ x \in (\mathbb{R}^n - \cup \mathcal{H}_j)}} \left\{ \left( \begin{smallmatrix} \mathtt{vec}(D(x)U^{l+1\top} wx^\top) \\ 0 \end{smallmatrix} \right) \right\} \subseteq R_{\theta_T}(\mathbb{W}^{g_2, \ell}) \tag{88}$$

$$\mathbb{R}^{d_1 n} \times \{0_{d_2 n}\} \subseteq \operatorname*{span}_{\tilde{\theta}_1 \in \tilde{\Theta}_1} \operatorname*{span}_{\substack{w \times \mathbb{R}^n \\ x \in (\mathbb{R}^n - \cup \mathcal{H}_j)}} \left\{ \left( \begin{smallmatrix} \mathtt{vec}(D(x)U^{l+1\top} wx^\top) \\ 0 \end{smallmatrix} \right) \right\} \tag{89}$$

*Proof of (88).* The main difference between the left-hand-side of (88) and the definition of $R_{\theta_T}$ in (79) is that in (88) $x$ can be freely chosen in the set $\mathbb{R}^n - \cup_j \mathcal{H}_j$ independently of $\tilde{\theta}_1$ (and of $\tilde{\theta}_3$). To prove (88) we exhibit below two vectors $\theta^+, \theta^- \in \Theta$ such that $\theta_T^+ = \theta_T^- = \theta_T$ and that for every $x \in \mathbb{R}^n - \cup_j \mathcal{H}_j$ and $w \in \mathbb{R}^n$, we either have

$$[\partial_{\theta_T} g(\theta^+; x)]^\top w = \left( \begin{smallmatrix} \mathtt{vec}(D(x)U^{l+1\top} wx^\top) \\ 0 \end{smallmatrix} \right) \in \overline{\operatorname*{span}_{x' \in \mathcal{X}_{\theta^+}, w \in \mathbb{R}^n} \{[\partial_{\theta_T} g(\theta^+; x')]^\top w\}} \tag{90}$$

or the equivalent with $\theta^-$ instead of $\theta^+$. By the closedness of finite-dimensional spaces and (79) we get

$$\begin{pmatrix} \mathtt{vec}(D(x){U^{l+1}}^\top wx^\top) \\ 0 \end{pmatrix} \in \operatorname*{span}_{x' \in \mathcal{X}_{\theta^+}, w \in \mathbb{R}^n} \{[\partial_{\theta_T} g(\theta^+; x')]^\top w\} \subseteq R_{\theta_T}(\mathbb{W}^{g_2, \ell}),$$

or of course the equivalent with $\theta^-$ instead of $\theta^+$, yielding the desired conclusion (88).

We now proceed to the construction of $\theta^+, \theta^-$. This is where we use Assumption P.1: this enables us to consider an arbitrary nonzero $a_0 \in \bigcap \operatorname{range} Q_i \subseteq \mathbb{R}^p$ and to denote $a = \begin{pmatrix} a_0 \\ \cdots \\ a_0 \end{pmatrix} \in \mathbb{R}^{pc_0} = \mathbb{R}^n$ as well as

$$\mathcal{A}^+ := \{x' \in \mathbb{R}^n : a^\top x' < 0\}, \quad \mathcal{A}^0 := \{x' \in \mathbb{R}^n : a^\top x' = 0\}, \quad \text{and} \quad \mathcal{A}^- := \{x' \in \mathbb{R}^n : a^\top x' > 0\}$$

We construct $\theta^+$ (resp. $\theta^-$) so that (90) or its equivalent with $\theta^-$ holds for every $x$ in the intersection of $\mathbb{R}^n - \cup \mathcal{H}_j$ with $\mathcal{A}^+$, with $\mathcal{A}^0$, or with $\mathcal{A}^-$.

Recall that we consider $\tilde{\theta}_1 \in \overline{\Theta}_1$ such that $U^{l+1} \in \hat{\Theta}_1$, and that this implies $\tilde{\theta}_1 \in \tilde{\Theta}_1$. We consider $\tilde{\theta}_3^+ \in \overline{\Theta}_3$ such that $V_+^l$ has all its rows that are equal[5] to $\epsilon a$ – in particular $V_+^l$ satisfies Item 2 in the assumptions of Lemma 4.4– where $\epsilon > 0$ is small enough to ensure that $(U^l, V_+^l) \in \Theta_l$ and hence $V_+^l \in \hat{\Theta}_l$ and $\tilde{\theta}_3^+ \in \tilde{\Theta}_3$. We denote $\theta^+ = (\tilde{\theta}_1, \theta_T, \tilde{\theta}_3^+)$. The construction of $\theta^-$ is similar, except that $V_-^l$ has all its rows equal to $-\epsilon a$ for small enough $\epsilon > 0$.

Considering $x \in (\mathbb{R}^n - \cup \mathcal{H}_j) \cap \mathcal{A}^+$ our goal is now to establish (90). As $\overline{\mathcal{X}_{\theta^+}} = \mathbb{R}^n$ by Lemma 4.4 (as $\theta^+ \in \Theta$), there exists a sequence of vectors $x_N \in \mathcal{X}_{\theta^+}$ such that $x_N \longrightarrow x$. By construction $\mathcal{A}^+$ is open, and $\mathcal{X}_{\theta^+} \subseteq \mathbb{R}^n - \cup \mathcal{H}_j$ (Indeed if $x \in \mathcal{X}_{\theta^+}$, then $\theta' \mapsto g(\theta', x)$ is $\mathcal{C}^2$ in the neighborhood of $\theta^+ := (\tilde{\theta}_1, \theta_T, \tilde{\theta}_3)$, so in particular $\theta'_T \mapsto g((\tilde{\theta}_1, \theta'_T, \tilde{\theta}_3), x) = U^{l+1} \sigma((V^{l+1})' x) + x$ (by (86)) is $\mathcal{C}^2$ in the neighborhood of $\theta_T$.) hence for $N$ large enough we have $x_N \in (\mathbb{R}^n - \cup \mathcal{H}_j) \cap \mathcal{A}^+$. As $\mathcal{A}^+ \subseteq \{x' : \sigma(V_+^l x') = 0\}$, the expression (87) of $[\partial_{\theta_T} g(\theta^+; \cdot)]^\top w$ is valid on $(\mathbb{R}^n - \cup \mathcal{H}_j) \cap \mathcal{A}^+$. Since it is continuous (because $D(\cdot)$ is locally constant on $\mathbb{R}^n - \cup \mathcal{H}_j$) we get

$$\begin{pmatrix} \mathtt{vec}(D(x){U^{l+1}}^\top wx^\top) \\ 0 \end{pmatrix} = [\partial_{\theta_T} g(\theta^+; x)]^\top w = \lim_{N \to \infty} [\partial_{\theta_T} g(\theta^+; x_N)]^\top w \in \overline{\operatorname*{span}_{x' \in \mathcal{X}_{\theta^+}, w \in \mathbb{R}^n} \{[\partial_{\theta_T} g(\theta^+; x')]^\top w\}} \quad (91)$$

therefore establishing (90) as claimed. The same reasoning holds for $x \in (\mathbb{R}^n - \cup \mathcal{H}_j) \cap \mathcal{A}^-$ using $\theta^-$ instead of $\theta^+$.

Finally for $x \in (\mathbb{R}^n - \cup \mathcal{H}_j) \cap \mathcal{A}^0$, one has in particular $V_-^l x = V_+^l x = 0$. As we have seen, $\overline{\mathcal{X}_{\theta^+}} = \overline{\mathcal{X}_{\theta^-}} = \mathbb{R}^n$, hence there exists a sequence of vectors $x_N \in \mathcal{X}_{\theta^+} \cup \mathcal{X}_{\theta^-}$ such that $x_N \longrightarrow x$. By extracting a subsequence if necessary we can assume that $\sigma(V_+^l x_N) = 0$ for every $N$ (or that $\sigma(V_-^l x_N) = 0$ for every $N$). Without loss of generality we assume $\sigma(V_+^l x_N) = 0$ for all $N$ (the other option is treated similarly). As proven above, we have $\mathcal{X}_{\theta^+} \cup \mathcal{X}_{\theta^-} \subseteq \mathbb{R}^n - \cup \mathcal{H}_j$, hence $x_N \in (\mathbb{R}^n - \cup \mathcal{H}_j) \cap \{x' : \sigma(V_+^l x') = 0\}$ for every $N$. Thus (91) remains valid and yields (90).

*Proof of* (89). First, observe that it is enough to prove that for each $k \in \{1, \cdots, d_1\}$ there exists $\tilde{\theta}_1 \in \tilde{\Theta}_1, \theta = (\tilde{\theta}_1, \theta_T, \tilde{\theta}_3) \in \Theta$, and $w \in \mathbb{R}^n$ such that

$$\mathtt{vec}(e_k x^\top, 0) \in \overline{\operatorname*{span}_{x' \in \mathbb{R}^n - \cup_j \mathcal{H}_j} \{\partial_{\theta_T} g^\top(\theta; x') w\}}, \quad \forall x \in \mathbb{R}^n \quad (92)$$

Indeed, by (87) and the closedness of finite-dimensional spaces this implies (89).

To prove (92), given an arbitrary $k \in \{1, \cdots, d_1\}$, the core of the proof is to exhibit below some $\tilde{\theta}_1 \in \overline{\Theta}_1$ such that the matrix $U^{l+1} \in \hat{\Theta}_{l+1}$ (recall that this implies $\tilde{\theta}_1 \in \tilde{\Theta}_1$) has a nonzero $k$-th column, and that for every $w \in \mathbb{R}^n$

$$\mathtt{vec}(E_{k,k}(U^{l+1})^\top wx^\top, 0) \in \overline{\operatorname*{span}_{x' \in \mathbb{R}^n - \cup_j \mathcal{H}_j} \{\partial_{\theta_T} g^\top(\theta; x') w\}}, \quad \forall x \in \mathbb{R}^n. \quad (93)$$

Since the $k$-th column of $U^{l+1}$ is nonzero, there exists an index $\ell$ such that $U_{\ell, k}^{l+1} = \alpha \neq 0$. Setting $w := e_\ell$ we have $E_{k,k}(U^{l+1})^\top w = \alpha e_k x^\top$, hence (93) implies (92).

Given an arbitrary $k \in \{1, \cdots, d_1\}$, we consider $k' = k \mod p_1$, by assumption Assumption E.1 one has $\operatorname{range}(P_{k'}) \neq \{0\}$, hence one can build $U^{l+1}$ as in (44) such that its $k$-th column is non equal to zero. To ensure that $(U^{l+1}, V^{l+1})$ satisfies Item 1 (hence $U^{l+1} \in \hat{\Theta}_{l+1}$, and hence $\theta \in \Theta$) we rescale $U^{l+1}$ such that its norm is small enough.

---

[5]The assumptions of Theorem 4.7 the matrix forbid $V^{l+1}$ to have colinear rows, but there is no such constraint on $V^l$.

With this choice of $U^{l+1}$, to prove (92) for every $x \in \mathbb{R}^n$ we first show it for $x$ living in $\mathcal{H}'_k := \mathcal{H}_k - \cup_{j \neq k} \mathcal{H}_j$ (note that this is a non empty set as the hyperplanes are pairwise distinct by definition of $\Theta$) before proving it for $x$ in the complementary direction. We denote

$$\mathcal{A}_k^+ := \{x \in \mathbb{R}^n : v_k^\top x > 0\}, \quad \text{and} \quad \mathcal{A}_k^- := \{x \in \mathbb{R}^n : v_k^\top x < 0\}$$

where we recall that $v_1, \ldots, v_{d_1}$ are the rows of $V^{l+1}$.

**We first show that** (92) **is satisfied on** $\mathcal{H}'_k$ **.**

Consider $x' \in \mathcal{H}'_k$. Denote $B(c, \eta)$ the open Euclidean ball of radius $\eta > 0$ centered at $c \in \mathbb{R}^n$. Given any $\eta > 0$, by continuity of $x \in \mathbb{R}^n \mapsto V^{l+1}x = (v_1^\top x, \cdots, v_{d_1}^\top x) \in \mathbb{R}^{d_1}$, there exists $x_\eta^+ \in B(x', \eta) \cap \mathcal{A}_k^+$ and $x_\eta^- \in B(x', \eta) \cap \mathcal{A}_k^-$ such that $1 = \mathrm{sign}(v_k^\top x_\eta^+) \neq \mathrm{sign}(v_k^\top x_\eta^-) = -1$ while for all $j \neq k$, $\mathrm{sign}(v_j^\top x_\eta^\pm) = \mathrm{sign}(v_j^\top x') \neq 0$ (as the hyperplanes are pairwise distinct by definition of $\Theta$). It follows that $x_\eta^\pm \in \mathbb{R}^n - \cup \mathcal{H}_j$. As a consequence, we have $D(x_\eta^+) - D(x_\eta^-) = \mathrm{diag}(e_k)$ hence for every $w \in \mathbb{R}^n$:

$$\begin{aligned}
\mathrm{vec}\left(E_{k,k}(U^{l+1})^\top w x'^\top, 0\right) &= \mathrm{vec}\left(\left(D(x_\eta^+) - D(x_\eta^-)\right)(U^{l+1})^\top w x'^\top, 0\right) \\
&= \lim_{\eta \to 0} \mathrm{vec}\left(D(x_\eta^+)(U^{l+1})^\top w(x_\eta^+)^\top - D(x_\eta^-)(U^{l+1})^\top w(x_\eta^-)^\top, 0\right) \\
&= \lim_{\eta \to 0}\left(\partial_{\theta_T} g^\top(\theta; x_\eta^+)w - \partial_{\theta_T} g^\top(\theta; x_\eta^-)w\right) \in \overline{\mathrm{span}_{x \in \mathbb{R}^n - \cup \mathcal{H}_j}}\{\partial_{\theta_T} g^\top(\theta; x)w\}.
\end{aligned}$$

This establishes (92) for any $x' \in \mathcal{H}'_k$.

**We finally show that** (92) **is satisfied for any** $x := v_k$**.**

With $x' \in \mathcal{H}'_k$ as above the continuity of $x \in \mathbb{R}^n \mapsto V^{l+1}x = (v_1^\top x, \cdots, v_{d_1}^\top x) \in \mathbb{R}^{d_1}$ also implies the existence of $\gamma > 0$ such that the vectors

$$x_K := x' + \gamma K v_k, \quad K \in \{-2, -1, 1, 2\},$$

satisfy $v_k^\top x_K \neq 0$ while for all $j \neq k$, $\mathrm{sign}(v_j^\top x_K) = \mathrm{sign}(v_j^\top x')$ (as the hyperplanes are pairwise distinct by definition of $\Theta$), so that $x_K \in \mathbb{R}^n - \cup \mathcal{H}_j$ and we similarly obtain

$$\begin{aligned}
&\gamma \mathrm{vec}\left(E_{k,k}(U^{l+1})^\top w v_k^\top, 0\right) \\
&= \gamma \mathrm{vec}\left(\underbrace{D(x_1)}_{=D(x_2)}(U^{l+1})^\top w v_k^\top - \underbrace{D(x_{-1})}_{=D(x_{-2})}(U^{l+1})^\top w v_k^\top, 0\right) \\
&= \mathrm{vec}\left(D(x_2)(U^{l+1})^\top w x_2^\top - D(x_1)(U^{l+1})^\top w x_1^\top - \left(D(x_{-1})(U^{l+1})^\top w x_{-1}^\top - D(x_{-2})(U^{l+1})^\top w x_{-2}^\top\right), 0\right) \\
&= [\partial_{\theta_T} g(\theta; x_2)]^\top w - [\partial_{\theta_T} g(\theta; x_1)]\top w - \{[\partial_{\theta_T} g(\theta; x_{-1})]\top w - [\partial_{\theta_T} g(\theta; x_{-2})]\top w\} \\
&\in \mathrm{span}_{x \in \mathbb{R}^n - \cup \mathcal{H}_j}\{\partial_{\theta_T} g^\top(\theta; x)w\},
\end{aligned}$$

which gives (92) for any $x \in \mathbb{R} v_k$.

We have thus proved that (92) holds for any $x \in \mathcal{H}'_k \cup \mathbb{R} v_k$. Since $\mathbb{R}^n = \mathrm{span}(\mathcal{H}'_k \cup \mathbb{R} v_k)$ this establishes (92) for every $x \in \mathbb{R}^n$. Since this holds for any $k$ this completes the proof of (89) and therefore of the theorem. $\qquad\square$

## Q. Proof of Proposition 5.1 (approximate conservation of laws under discrete dynamics)

**Proposition 5.1.** *Let $h(\theta)$ be a conservation law of the gradient flow with a bounded Hessian $\forall \theta$, $\quad \|\partial^2 h(\theta)\| \leq C_h$. Suppose that the gradients remain bounded in expectation throughout the algorithm:*

$$\mathbb{E}_{Z_k, \theta_k}\left\|\nabla L_{Z_k}(\theta_k)\right\|_2^2 \leq C_L. \tag{26}$$

*Then, we have* $\quad \mathbb{E}\left|h(\theta_k) - h(\theta_0)\right| \leq \frac{C_h C_L}{2}\sum_{i=0}^{k-1}\tau_i^2.$ $\tag{27}$

*Proof.* Since $h$ is a conservation law, by Proposition 2.3 we have $\langle \nabla h(\theta_k), \nabla L_{Z_k}(\theta_k) \rangle = 0$ for every $k$. Besides, a Taylor expansion yields

$$h(\theta_{k+1}) - h(\theta_k) = \frac{\tau_k^2}{2} \left[ \nabla L_{Z_k}(\theta_k) \right]^\top \partial^2 h(\xi) \nabla L_{Z_k}(\theta_k)$$

for some $\xi$ in the segment $[\theta_k, \theta_{k+1}]$. Applying the bounds on $\partial^2 h(\theta)$ and $\nabla L_{Z_k}(\theta_n)$, we get

$$\mathbb{E}|h(\theta_{k+1}) - h(\theta_k)| \leq \frac{\tau_k^2}{2} C_h C_L.$$

Summing over $k$ completes the proof. $\square$

## R. Conservation laws for Adam Flow

We recall that for Adam flow the space $\mathcal{W}_\theta^{g,\ell}$ is defined by

$$\mathcal{W}_\theta^{g,\ell} := \operatorname*{span}_{Z=(x_i,y_i)\in(\mathcal{X}_\theta\times\mathcal{Y})^N} \{\operatorname{sign}(\nabla L_Z(\theta))\}. \tag{94}$$

By directly adapting the results of Section 2.1 with $\mathcal{W}_\theta^{g,\ell}$ defined in (94), this leads to the direct corollary:

**Corollary R.1.** *Consider a loss $\ell(z,y)$ that satisfies Assumption 2.4. Under Assumption 2.6, then for all $\theta \in \Theta$:*

$$\mathcal{W}_\theta^{g,\ell} = \operatorname*{span}_{w\in\mathcal{W}_{\phi(\theta)}^{f,\ell}} \{\operatorname{sign}(\partial\phi(\theta)^\top w)\}. \tag{95}$$

*In particular if $\mathcal{W}_{\phi(\theta)}^{f,\ell} = \mathbb{R}^d$ one has*

$$\mathcal{W}_\theta^{g,\ell} = \operatorname*{span}_{w\in\mathbb{R}^d} \{\operatorname{sign}(\partial\phi(\theta)^\top w)\}. \tag{96}$$

Thus by Theorem 3.4 and Corollary R.1 one directly obtains:

**Theorem R.2.** *Under Assumption 2.4, if $\mathcal{V}_\ell = \mathbb{R}^n$, then considering $\Theta = \mathbb{R}^D$ and $\phi(\theta) := UV^\top$ for linear neural networks, one has: $\mathcal{W}_{\phi(\theta)}^{f,\ell} = \mathbb{R}^d$ and $\mathcal{W}_\theta^{g,\ell} = \operatorname*{span}_{w\in\mathbb{R}^d}\{sign(\partial\phi(\theta)^\top w)\}$ for Adam flows.*

Similarly, by applying Theorem 3.8 and Corollary R.1 one directly obtains the following theorem for an attention layer:

**Theorem R.3.** *Under Assumption 2.4, if $\mathcal{V}_\ell = \mathbb{R}^n$ and $N \geq 2$ then*

$$\mathcal{W}_{\phi(\theta)}^{f,\ell} = \mathbb{R}^d, \text{ and } \mathcal{W}_\theta^{g,\ell} = \operatorname*{span}_{w\in\mathbb{R}^d}\{\operatorname{sign}(\partial\phi(\theta)^\top w)\}, \forall\theta \in \Theta_{\mathtt{att}},$$

*for an attention layer and for Adam flows, where we recall that the reparametrization $\phi$ is defined by $\phi(\theta) = (\phi_1, \phi_2)$ with $\phi_1 = Q^\top K$ and $\phi_2 = V^\top O$.*

*Moreover as the parametrization $\phi_1$ and $\phi_2$ are separable, one has under the same assumptions:*

$$\mathcal{W}_\theta^{g,\ell} = \operatorname*{span}_{w=(w_1,w_2)\in\mathbb{R}^d} \left\{ \begin{pmatrix} \operatorname{sign}(\partial\phi_1(Q,K)^\top w_1) \\ \operatorname{sign}(\partial\phi_2(V,O)^\top w_2) \end{pmatrix} \right\}, \forall\theta \in \Theta_{\mathtt{att}}.$$

**Numerical experiments** The code provided in our GitHub repository numerically investigates the dimension of the space $\mathcal{W}_\theta^{g,\ell}$ defined in (96), i.e. the space spanned by the sign vectors of gradients $\partial\phi(\theta)^\top z$, where $\phi(U,V) = UV^\top$ and $\theta = (U,V) \in \mathbb{R}^{n\times r} \times \mathbb{R}^{m\times r}$. For various choices of $n, m, r$, the code: 1) generates random parameter matrices $U$ and $V$; 2) samples random vectors $z \in \mathbb{R}^{n\times m}$; 3) computes the gradients of $\phi$ with respect to $U$ and $V$; and 4) collects the sign patterns of the projected gradients $\partial\phi(\theta)^\top z$. It then estimates the dimension of the linear span of these sign vectors by calculating the rank of the resulting sign matrix. This provides a lower bound on the dimension of the sign-gradient space $\mathcal{W}_\theta^{g,\ell}$. In all tested configurations, this lower bound consistently equals the total parameter dimension $D = (n+m)r$, indicating that the sign vectors span the full parameter space, expect in the case $n = m = r = 1$ where in that case the dimension of the space is equal to 1. Consequently, this numerical observation suggests that there are no conservation laws for the mapping $\phi(U,V)$, except in the case where $n = m = r = 1$. In the latter case, there is indeed exactly one independent conservation law given by: $\theta \mapsto |U| - |V|$ as we now show.

**The special case** $n = m = r = 1$. We show that $h : \theta = (u, v) \in \mathbb{R}^2 \mapsto |u| - |v|$ is a conservation law for $g$ for the Adam flow (12).

By characterization of conservation laws we only need to show that for any $\theta \in (\mathbb{R}_*)^2$,

$$\nabla h(\theta) \perp \underset{w \in \mathbb{R}}{\text{span}}\{\text{sign}\left(\partial\phi(\theta)^\top w\right)\} = \mathbb{R}\text{sign}\left(\begin{pmatrix} v \\ u \end{pmatrix}\right),$$

which is always true as $\nabla h(\theta) = \begin{pmatrix} \text{sign}(u) \\ -\text{sign}(u) \end{pmatrix}$.

