# OpenReview forum: "Transformative or Conservative? Conservation laws for ResNets and Transformers"
_ICML.cc/2025/Conference — ICML 2025 oral_

### Official Review · Reviewer_iC26 · 2025-03-11

**Overall Recommendation:** 4

**Summary:**

This paper studies gradient flow conservation laws in neural networks of many architectures including linear networks, ReLU networks, networks with attention layers, and networks with residual skip connections. Using ideas from Lie Algebra (specifically Frobenius' theorem), they are able to characterize the number of conservation laws and in some cases prove that their provided conservation laws are complete. For transformers such conservation laws take the form of functions of $Q Q^\top - K K^\top$ and $VV^\top - OO^\top$, similar to the conservation laws of weights of deep linear networks.  They also characterize gradient flow conservation laws for subsets of the parameters such as blocks in a deep residual network. Lastly, they provide an error analysis for discrete time step GD and gradient flow, showing an error in the conservation law which accumulates over time on the order of $\text{learning rate}^2$.

**Claims And Evidence:**

This paper is primarily theoretical and provides many proofs and references to the very relevant prior work of Marcotte et al 2023.  The claim of stability of the conservation law at finite learning rate is verified in an experiment in Figure 1, where for very small learning rate, the conservation law holds well and for larger learning rates the error accumulates.

**Essential References Not Discussed:**

N/A

**Experimental Designs Or Analyses:**

The Figure 1 should state which conservation law is being tested. Some additional numerical tests, especially regarding the new conservation laws (attention, residual blocks, etc), could improve the paper.

**Methods And Evaluation Criteria:**

The paper is primarily theoretical so there is not much to check. Perhaps more numerical experiments verifying different conservation laws in the transformer could be interesting as these results appear to be novel contributions of this work.

**Other Comments Or Suggestions:**

See questions

**Other Strengths And Weaknesses:**

Another strength of the paper is its rigor and precision. However, the current writing could potentially be challenging to a machine learning theorist with less background in Lie Algebras and integrable systems. Some informal discussions after each result could make the paper more readable. In addition, it could improve the paper to test numerically some of the newer claims to demonstrate the power of this approach in the newer settings that go beyond Marcotte et al 2023.

**Questions For Authors:**

1. The authors write that all conservation laws in the transformer case are functions of $QQ^\top - KK^\top$. Is this full matrix difference conserved or just specific nonlinear functions of it? I ask because in the deep linear case, the conservation laws take the form $ W_{\ell} W_{\ell}^\top - W_{\ell+1}^\top W_{\ell+1}$ for **all matrix entries**.  Is there a method to extract which nonlinear functions of this difference are conserved?
2. What conserved quantity is being tracked in Figure 1 and the Appendix Figures?
3. Do the authors think that normalization layers (like layernorm) will make the number of conservation laws larger? If the gradient is independent of the scale / radial direction of the features, then it could be that the dynamics are confined to a lower dimensional subspace.

**Relation To Broader Scientific Literature:**

This paper provides a nice analytical toolkit to examine conservation laws in different neural network architectures. This is fundamental to deep learning theory since it aids our understanding of the implicit bias of gradient flows and the types of solutions networks converge to. While I was familiar with the two layer linear and ReLU conservation laws, I like that this approach is flexible enough to provide some insights into the residual networks and models with attention layers.

**Theoretical Claims:**

I checked and read Appendix A, B and C for the theoretical derivations. I also read some of Marcotte et al 2023 which contains some background material on this approach. Appendix B and C contains much of the technical machinery to prove properties and exhaustiveness of the conserved quantities. The result relies on the Frobenius theorem which is a fundamental tool in integrable systems theory. I also briefly skimmed Appendices E-I which derive the results for the attention layer which appeared correct.

---

> ### Author Rebuttal · Authors · 2025-04-01
>
> We would like to thank the reviewer for his positive comments and his insightful feedback.
>
> > **Q1d** "The Figure 1 should state which conservation law is being tested. Some additional numerical tests, especially regarding the new conservation laws (attention, residual blocks, etc), could improve the paper.” + “What conserved quantity is being tracked in Figure 1 and the Appendix Figures?”
>
> Thank you for pointing out that point. The caption will be fixed. As detailed in section 5.2, the conserved quantity that is tracked is the sum of all conserved quantities of equation 14 associated to the first convolutive layer. In particular, Figure 1 focuses on some new conservation laws of our paper (ResNet architecture, thus skip connection and convolutive layers). Concerning numerical experiments regarding new conservation laws associated to attention layers, see our answer to your question **Q3d**.
>
> > **Q2d** Some informal discussions after each result could make the paper more readable
>
> Thank you for the suggestion that will be taken care of in the final version.
>
> > **Q3d** In addition, it could improve the paper to test numerically some of the newer claims to demonstrate the power of this approach in the newer settings that go beyond Marcotte et al 2023.
>
> Beside numerically testing our new theoretical founding in the case of a ResNet’s training (see our answer to your question **Q1d**) in our paper (cf Fig. 1), we propose to add this experiment https://anonymous.4open.science/r/Numerical_experiments-2730/Transformer_IMDb.ipynb that tracks the evolution of a conservation law during the training of a transformer. More precisely, we train a transformer on the sentiment analysis dataset (IMDb dataset) and we track the evolution of $| h(\theta_n) - h (\theta_0)|$ where $h (\theta) = \| \| Q_1 Q_1^\top - K_1 K_1^\top\| \|_p$ (ie the norm of Frobenius of the conservation law given by Corollary 3.10, where $Q_1$ and $K_1$ are the query and key matrices of the first head and from the first layer. Similarly to Figure 1, we obtain the same bound O (step * learning_rate^2). This new experiment also focuses on new conservation laws (attention block and skip connection).
>
> > **Q4d** The authors write that all conservation laws in the transformer case are functions of $QQ^\top−KK^\top$. Is this full matrix difference conserved or just specific nonlinear functions of it? I ask because in the deep linear case, the conservation laws take the form $W_l W_l^\top−W_{l+1}^\top W_{l+1}$ for all matrix entries. Is there a method to extract which nonlinear functions of this difference are conserved?
>
> Yes, the full matrix difference is conserved, and this implies that any linear or nonlinear function of these matrix entries is also conserved: when a function $h(\theta)$ is conserved, any function$ \Psi(h(\theta))$ is also conserved. This is exactly why we need to define a notion of independent functions (Definition 2.10), so that we remove all these functional redundancies (see also our Proposition 2.12). We will stress on that point on the final version of our paper.
>
> > **Q5d** Do the authors think that normalization layers (like layernorm) will make the number of conservation laws larger? If the gradient is independent of the scale / radial direction of the features, then it could be that the dynamics are confined to a lower-dimensional subspace
>
> Since these layers are not parameterized and only marginally change the diversity of possible inputs to the subsequent layers, we expect them to have no effect on the analysis and, therefore, on the number of conserved laws.
> However, if you consider in other settings, another type of normalization that operates directly on the parameters, you should expect (as in the case of a cross-entropy layer) new conservation laws.

---

> > ### Comment · Reviewer_iC26 · 2025-04-04
> >
> > I thank the authors for their detailed responses. I will maintain my score.

---

### Official Review · Reviewer_Y1a9 · 2025-03-12

**Overall Recommendation:** 4

**Summary:**

The authors study conservation in more general networks than the previously studied ReLU and linear networks. In particular, they focus on convolutional ResNets and Transformers.

## Update after rebuttal
I appreciate the authors' detailed rebuttal, which has resolved my concerns. As a result, I have raised my score.

**Claims And Evidence:**

Yes the claims that the paper is the first to analyze conservation in Transformers is true to the best of my knowledge and the proofs are rigorous of the proposed theorems.

**Essential References Not Discussed:**

- Other good ODE-based references to add include  Ott et al., "ResNet After All: Neural ODEs and Their Numerical Solution", ICLR, 2021 and Krishnapriyan et al., "Learning continuous models for continuous physics", Nature Communications, 2022, which views a ResNet as a discrete Forward Euler discretization of a ODE and shows similar performance to the "continuous" NeuralODE. Onken et al., "Discretize-Optimize vs. Optimize-Discretize for Time-Series Regression and Continuous Normalizing Flows", 2020 shows that it may be preferable to first discretize and then optimize.

**Experimental Designs Or Analyses:**

The paper mainly consists of theoretical results. There are numerical results that demonstrate the the error scales numerically as the $O(step-size^2)$ in Figure 1. This numerical experiment studies ResNet-18 on CIFAR-10 dataset. It may be interesting to test different architectures on additional datasets as well so that the numerical results can be more thorough. I would be interested in seeing the CIFAR results on the Transformer architecture.

**Methods And Evaluation Criteria:**

Yes, the methods used in the proofs make sense for the gradient flow problem at hand. This work is mainly theoretical.

**Other Comments Or Suggestions:**

N/A

**Other Strengths And Weaknesses:**

## Weaknesses
- The authors mention a clear limitation in the conclusion that their work does not dynamics from Adam optimizer, which is the most commonly used optimizer for training Transformer and hence limits the practical aspects of the work. It would be interesting for the authors to discuss ideas of how to support Adam as future work.
### Minor
- Typo in "traing" on line 065 left column for training.
- \citep is also incorrect on line 062
- “overalapping” on line 102
- Increase x and yaxis label size, legend and titles in Figure 1

## Strengths
- Literature is well-summarized and reviewed.
- Section 2 provides a clear problem definition
- The authors are very clear on their contribution.
- Clear limitations and directions for future work are discussed in the conclusion.

**Questions For Authors:**

1. There is a clear connection to ODEs in the paper. Could the method also be connected to PDEs with a spatial component potentially in the image domain?
2. The problem definition defines the problem for tabular data with respect to classical regression or classification tasks. How could this method be extended to time series data?

**Relation To Broader Scientific Literature:**

- The paper is well-connected within the gradient flows literature and relevant to extend the work there to more modern architectures including transformers. The authors in the introduction describe conservation laws as a "balancedness condition". I think it would be interesting to see how this relates to conservation laws in PDEs and scientific machine learning literature, where a conservation law in integral form states that the rate of change of total mass of the system is balanced by the change in fluxes in the domain, which seems to be a related condition (see ansen et al., "Learning Physical Models that Can Respect Conservation Laws", ICML, 2023 and Richter-Powell et al., "Neural Conservation Laws: A Divergence-Free Perspective", NeurIPS, 2022).

**Theoretical Claims:**

The authors prove that conservation laws of gradient flows with weight decay are determined by their equivalent without weight decay in Theorem 2.1 and list this as a contribution. The emphasis on weight decay is unclear at this point in the introduction. The authors then discover new conservation laws for more modern architectures with the assumption that they have skip connections, i.e., shallow multi-channel convolution layers (Theorem 3.6), self-attention layers (Corollaries 3.9-3.10), cross-entropy classification layer (Prop 3.11) and MLP (Prop 3.2). Theorem 4.6 is the proof that these conservation laws match the laws of blocks in isolation. They also prove that the error bound scales as the step-size squared. The proofs seem correct to me.

---

> ### Author Rebuttal · Authors · 2025-04-01
>
> We thank the reviewer for the positive comments and insightful feedback.
>
> >  **Q1c** about structure theorem
>
> We stated theorem 2.1 right at the beginning to lighten the notation in the rest of the paper. It is new to the best of our knowledge.
>
> > **Q2c** CIFAR results on the Transformer architecture
>
> In this experiment https://anonymous.4open.science/r/Numerical_experiments-2730/Transformer_IMDb.ipynb, we train a transformer and track the evolution of the conservation law given by Corollary 3.10, confirming the O (step * learning_rate^2) behaviour. See also our answer to **Q3d** of reviewer iC26 for more detail.
>
> > **Q3c** how this relates to conservation laws in PDEs and scientific machine learning literature (...)
>
> Thank you for these references! The common technical aspect to our work and these references is the link between symmetries and conservation laws. However, while we *uncover* conservation laws satisfied by *the parameters of a neural network during the flow associated to their training*, these references *design neural networks implementing functions that satisfy certain physics-driven/PDE-driven conservation laws.*
>
> > **Q4c** in Essential References Not Discussed
>
> Thank you for these references! The 'gradient flow' ODE (3) describes *parameter* dynamics during training, while the Neural ODEs governs the *input* transformation $x$ through an infinitely deep network, *not the parameter evolution $\theta$*. Connections to Neural ODEs demonstrate the broader utility of Lie algebraic techniques for universality proofs in ML, see e.g. arXiv:1908.07838, arXiv:2404.08289. We will add this discussion to our related work section.
>
> > **Q5c** How to support Adam as future work
>
> Analyzing the original Adam algorithm is challenging due to the presence of two momentum parameters. However, a simplified version can be readily analyzed using our framework, and we will update the manuscript accordingly. A more detailed analysis of Adam itself remains an interesting direction for future work.
>
> The continuous limit of Adam yields an ODE where the gradient is replaced by its sign. $W(\theta)$ is spanned by signed gradients, it satisfies the assumptions of our results. In the simple case of a $2$-layer linear neural network we easily relate it with $\phi(\theta) = U V^\top$ and deduce in this code https://anonymous.4open.science/r/Numerical_experiments-2730/Adam.ipynb that $dim Lie (W(\theta)) = D$, so that there is no conservation law (except when $U$ and $V$ are scalars, in which case $|U|- |V|$ is conserved). For more detail, see our answer to **Q3b** of reviewer RSFD. We will mention this at the end of our paper.
>
> > **Q6c** (...) connected to PDEs with a spatial component (...)?
>
> This is a good question that relates closely to your earlier question **Q3c**. In response to **Q3c**, we mentioned that our analysis focuses on conservation with respect to the parameters $\theta$, rather than the input $x$ of the networks. However, we would like to point out that it is indeed possible to apply our approach to networks that include layers acting spatially on a signal or image $x$ (such as differential operators in the context of PDEs). In particular, one can design weight matrices in the network to operate directly on the spatial domain, typically corresponding to a discretization of a differential operator. This approach could provide a useful starting point for studying architectures such as PINNs or Neural Operators. After discretization, translation-invariant differential operators (e.g. Laplacians) become convolutions, which align naturally with our current methodology. That said, extending our results to infinite-dimensional operators remains an open and promising direction for future work. We will include these possibilities in the final version of the manuscript. If the reviewers have specific architectures or PDEs in mind, we would be very interested to hear more about them.
>
> > **Q7c** (...) extended to time series data?
>
> Transformers are indeed well-suited for time series data (as well as for NLP tasks involving next-token prediction). The only difference compared to the setup presented in our paper is the inclusion of masking to make the architecture causal. This modification does not affect our theoretical framework—the conservation laws remain unchanged. Additionally, the numerical code used to test the approximate conservation law under SGD can be applied in the causal setting simply by enabling the use_mask =True flag (cf https://anonymous.4open.science/r/Numerical_experiments-2730/Transformer_IMDb.ipynb). In practice, we observe that the predictions of our theory remain accurate even when accounting for discrete stepping in the algorithm. We will update the manuscript to reflect this. Extending our methods to other types of causal architectures, such as recurrent neural networks and Selective State Space Models, represents a promising direction for future work, which we will also mention.

---

> > ### Comment · Reviewer_Y1a9 · 2025-04-01
> >
> > I thank the reviewers for their detailed rebuttal.  I will increase my score.

---

### Official Review · Reviewer_RSFD · 2025-03-13

**Overall Recommendation:** 4

**Summary:**

**Summary of Contributions:**

The paper makes significant contributions by **deriving and analyzing conservation laws for modern deep learning architectures, specifically convolutional ResNets and Transformer networks**. It extends the understanding of conservation laws, which were previously mostly studied in shallow ReLU and linear networks, to these more practical architectures. The authors first show the conservation laws for basic building blocks and then analyze deeper networks, introducing the notion of conservation laws that depend only on a subset of parameters (e.g., a layer, a residual block, or an attention layer). A key finding is that **residual blocks have the same conservation laws as the block without the skip connection**. The paper also demonstrates that these conservation laws, initially established in the continuous gradient flow regime, are approximately preserved under discrete Stochastic Gradient Descent (SGD) dynamics.

*   **Literature:**
    *   The paper **appropriately cites relevant works** in the field of conservation laws in neural networks and the theory of deep learning. It clearly positions its contributions in relation to existing results, such as the completeness of conservation laws for shallow ReLU and linear networks and the analysis of implicit biases in ResNets and Transformers. The references are comprehensive and include both foundational works and recent advancements.

**3. Pros and Cons:**

*   **Pros:**
    *   **Novel and significant theoretical contributions** extending the understanding of conservation laws to modern deep architectures (ResNets and Transformers).
    *   **Rigorous mathematical framework** with formal definitions, theorems, and detailed proofs in the appendices.
    *   Introduction of the concept of **conservation laws depending on a subset of parameters** ("block" conservation laws).
    *   Important findings such as **residual blocks having the same conservation laws as their non-residual counterparts**.
    *   Theoretical analysis showing that conservation laws are **approximately preserved under SGD dynamics** with an error bound.
    *   **Empirical validation** of the approximate conservation under SGD through numerical experiments on ResNet-18.
    *   Completeness results for conservation laws in several basic building blocks with skip connections.

**Overall Assessment:**

This paper presents a **highly valuable and significant contribution** to the theoretical understanding of deep learning by extending the concept of conservation laws to modern neural network architectures. The rigorous mathematical derivations and the analysis of the behavior under SGD provide important insights into the training dynamics of ResNets and Transformers. While some limitations exist in terms of the architectures and optimizers covered, the novelty and soundness of the presented work are strong.

**Claims And Evidence:**

The paper "Transformative or Conservative?" claims to support its contributions with clear evidence through **mathematical derivations and proofs** presented within the main text and in the appendices. Several key findings are explicitly linked to theorems, propositions, and corollaries, with their proofs detailed in the appendices. For instance:

*   The structure theorem (Theorem 2.1) demonstrating the relationship between conservation laws with and without weight decay has its proof in Appendix A.
*   The conservation laws for shallow multi-channel convolution layers are presented in Theorem 3.6, with a more general proof in Appendix E.
*   The characterization of conservation laws for self-attention layers is given in Corollary 3.9 and 3.10, with proofs in Appendix G.
*   The existence of independent conservation laws for the cross-entropy classification layer is shown in Proposition 3.11, with a proof in Appendix H.
*   The proposition that residual blocks have the same conservation laws as their non-residual counterparts (Proposition 3.2) includes a proof in the main text.
*   Theorem 4.6, stating that conservation laws depending only on a block's parameters match those of the isolated shallow block, has its proof in Appendix M.
*   The absence of conservation laws for overlapping blocks in ResNets (Theorem 4.7) is proven in Appendix O.
*   The approximate preservation of conservation laws under SGD (Proposition 5.1) is supported by a proof in Appendix P.

Furthermore, the paper includes **numerical experiments** in Section 5.2 and Figure 1 that empirically evaluate the behavior of a conserved function during ResNet training with SGD. The results are presented as confirming the approximate conservation and aligning with theoretical expectations derived from Proposition 5.1.

**Essential References Not Discussed:**

NA

**Experimental Designs Or Analyses:**

Yes Experimental design is well versed.

**Methods And Evaluation Criteria:**

Here's a breakdown:

*   **Proposed Methods:** The core method employed in the paper is **mathematical derivation and analysis** of conservation laws. This is a fundamental approach for studying invariant quantities in dynamical systems, which aligns directly with the definition of conservation laws in the context of gradient flow training dynamics. By first establishing conservation laws for basic building blocks (like shallow networks and attention layers) and then extending the analysis to deeper architectures and considering subsets of parameters, the paper adopts a **structured and analytical approach** to a complex problem.

*   **Evaluation Criteria:** The evaluation strategy consists of two main components:
    *   **Mathematical Proofs:** The paper claims to support its theoretical contributions with **rigorous mathematical proofs** provided in the main text and, more extensively, in the appendices [the previous response]. Deriving these proofs is the primary way to validate the existence and form of the identified conservation laws. The presentation of findings as theorems, propositions, and corollaries indicates a commitment to mathematical rigor [the previous response].
    *   **Empirical Validation:** To examine the persistence of conservation laws under practical training scenarios with discrete dynamics like Stochastic Gradient Descent (SGD), the paper includes **numerical experiments**. These experiments track the evolution of a specific conserved quantity during the training of a ResNet-18 on the **CIFAR-10 dataset**. CIFAR-10 is a standard benchmark dataset in computer vision, making it a relevant choice for evaluating the behavior of ResNet architectures, which have been highly successful in this domain. The paper analyzes the conservation error in relation to the learning rate, comparing it with theoretical bounds derived for SGD. This empirical approach helps to bridge the gap between the theoretical continuous gradient flow regime and real-world discrete optimization.

In summary:

*   The use of **mathematical derivations** is a natural and appropriate method for investigating conservation laws, which are inherently mathematical properties of the training dynamics.
*   The **evaluation based on mathematical proofs** is the cornerstone of validating the theoretical claims made in the paper [the previous response].
*   The inclusion of **numerical experiments on a relevant benchmark dataset (CIFAR-10) with a widely used architecture (ResNet-18) and optimizer (SGD)** provides empirical support for the theoretical findings, particularly concerning the behavior of conservation laws in discrete optimization settings. Tracking a theoretically conserved quantity and comparing its empirical behavior with theoretical predictions is a sensible evaluation criterion.

**Other Comments Or Suggestions:**

**Detailed Evaluation:**

*   **Novelty, Relevance, and Significance:**
    *   **Novelty**: The paper is **highly novel** as it tackles a largely unexplored area of conservation laws in modern, deep architectures like ResNets and Transformers. The introduction of "block-wise" conservation laws and the analysis of their persistence under SGD are new contributions. The paper also provides new completeness results for several basic building blocks with skip connections.
    *   **Relevance**: Understanding the training dynamics of deep networks is a fundamental challenge in deep learning. Conservation laws offer valuable insights into the implicit bias of training algorithms and network architectures, as well as playing a role in theoretical convergence analyses. This research is therefore **highly relevant** to the deep learning research community. Furthermore, the potential application of these laws in designing new optimization schemes adds to the practical relevance.
    *   **Significance**: By extending the theory of conservation laws to ResNets and Transformers, the paper bridges an important gap in our theoretical understanding of these widely used architectures. The findings, such as the invariance of conservation laws with respect to skip connections and the approximate preservation under SGD, are **significant** for both theoretical analyses and potentially for guiding the development of new training techniques. The completeness results for basic building blocks provide a solid theoretical foundation.

*   **Soundness:**
    *   The paper appears to be **theoretically sound**. It builds upon existing frameworks for analyzing conservation laws in gradient flows. The paper provides formal definitions and propositions, with many proofs detailed in the appendices. The structure theorem (Theorem 2.1) establishing the relationship between conservation laws with and without weight decay seems crucial. The characterization of conservation laws for various building blocks (shallow convolutional ReLU networks, attention layers, cross-entropy classification layers) is supported by theorems and corollaries.
    *   The extension to deeper networks through the analysis of "block" conservation laws (Proposition 4.3 and Theorem 4.6) provides a rigorous way to connect the conservation laws of individual components to the global network behavior. The negative result regarding conservation laws overlapping residual connections (Theorem 4.7) is also an interesting and potentially important finding.
    *   The analysis of the persistence of conservation laws under discrete SGD dynamics (Proposition 5.1) provides a theoretical justification for the empirical observations. The error bound derived scales with the square of the step size, which is supported by the numerical experiments.
    *   The assumptions made throughout the paper (e.g., Assumption 2.4, 2.6, A.8, E.1) are clearly stated, and their implications are discussed. The paper acknowledges limitations, such as not covering Adam optimizer or normalization layers.

*   **Quality of Writing/Presentation:**
    *   The writing is generally **clear and well-structured**. The paper begins with a concise introduction and clearly outlines its contributions. The use of definitions, propositions, theorems, and corollaries provides a formal and logical flow.
    *   The organization of the paper, moving from basic building blocks to deeper networks and then to discrete dynamics, is logical. The appendices provide necessary details for the proofs, allowing the main body of the paper to remain focused on the key concepts and results.
    *   The inclusion of numerical experiments (Section 5.2 and Figure 1/4) helps to illustrate and validate the theoretical findings regarding the approximate conservation under SGD.
    *   Potential improvements could include:
        *   More intuitive explanations of the implications of the discovered conservation laws for the training process and the properties of the learned models.
        *   Further discussion on the intuition behind the absence of conservation laws overlapping residual connections.
        *   A more detailed explanation of the conditions under which the key assumptions (e.g., bounded Hessian, bounded gradients) hold in the context of deep networks.

**Other Strengths And Weaknesses:**

*   **Strengths**
    1.  Novel theoretical framework for analyzing conservation laws in ResNets and Transformers.
    2.  Rigorous mathematical derivations and completeness results for key building blocks.
    3.  Insightful findings regarding the impact of residual connections and the behavior under SGD.
    4.  Introduction of "block" conservation laws, offering a new perspective on analyzing deep networks.

**Questions For Authors:**

1.  **Completeness of Conservation Laws for Key Architectures:** The paper establishes conservation laws for several building blocks, including single attention layers, and shows that residual blocks inherit the laws of their non-residual counterparts. However, the completeness of conservation laws for deeper ResNet and Transformer architectures, as a whole, is not fully addressed, and completeness for multi-head attention remains an open problem.
    *   **Question:** Could the authors elaborate on the challenges in proving the completeness (or demonstrating the existence of additional laws) for deeper ResNet and Transformer architectures? What are the key obstacles compared to the analysis of the individual building blocks?

2.  **Practical Exploitation of Conservation Laws for Training:** The paper touches upon the potential for designing new optimization schemes that enforce or deviate from these conservation laws to potentially accelerate convergence. The numerical experiments demonstrate the approximate preservation under SGD.
    *   **Question:** Beyond observing their approximate preservation under SGD, do the authors have further insights or preliminary ideas on how the newly discovered conservation principles could be actively leveraged in practice? For example, could these laws inform initialization strategies, regularization techniques, or the design of more efficient optimizers for ResNets and Transformers?

3.  **Addressing Limitations in Future Work:** The conclusion explicitly mentions several limitations, including the absence of analysis for Adam optimizer, normalization layers, and the full complexity of multi-head attention in Transformers.
    *   **Question:** Could the authors briefly outline their planned approach or initial thoughts on how they intend to address these limitations in future research? What are the main challenges they foresee in extending their analysis to these more complex scenarios?

**Relation To Broader Scientific Literature:**

*   **Extending the Study of Conservation Laws to Modern Architectures:** The paper explicitly states that while conservation laws in gradient flow training dynamics are relatively well-understood for shallow ReLU and linear networks, their study has been **"largely unexplored for more practical architectures"** like convolutional ResNets and Transformer networks. This directly addresses a gap in the existing literature. Prior works like **Saxe et al. (2013)**, **Du et al. (2018)**, and **Arora et al. (2019)** established conservation laws primarily for these simpler architectures. This paper aims to bridge this gap by focusing on more complex, state-of-the-art models.

*   **Building Upon Completeness Results for Shallow Networks:** The paper mentions that **(Marcotte et al., 2023) demonstrated the "completeness" of conservation laws for shallow ReLU and linear networks** under Euclidean gradient flows, meaning no additional conservation laws exist in those cases. The current paper extends this line of inquiry to more complex architectures, not just characterizing laws but also considering their completeness for certain building blocks.

*   **Connecting to Work on Conservation Laws with Different Optimizers:** The paper cites **(Marcotte et al., 2024)** for unveiling novel conservation laws under alternative optimization algorithms like non-Euclidean gradient flows and momentum-based dynamics, highlighting their different characteristics compared to simple gradient flows. While the current paper primarily focuses on Euclidean gradient flows, it acknowledges this related work, suggesting a broader interest in the role of the optimization algorithm in determining conservation laws.

*   **Generalizing Prior Findings on Convolutional Networks:** The paper notes that **(Du et al., 2018) identified conservation laws for feed-forward networks with single-channel convolutions**. Theorem 3.6 in the current paper **generalizes these findings to multi-channel convolutional ReLU networks**, demonstrating that similar types of functions are preserved in this more complex setting.

*   **Investigating the Impact of Residual Connections:** Given the fundamental role of skip connections in ResNets **(He et al., 2016)**, the paper specifically examines their effect on conservation laws. Proposition 3.2, which shows that residual blocks have the same conservation laws as their non-residual counterparts, is a novel finding that directly relates to the architectural innovation of ResNets. This also contrasts with work like **(Marion et al., 2023)** which explores the implicit bias of ResNets towards Neural ODEs but doesn't focus on conservation laws in the same way.

*   **Analyzing Conservation Laws in Transformer Architectures:** Transformers **(Vaswani, 2017)** have become dominant in various domains. The paper's derivation and analysis of conservation laws for **single attention layers (Corollary 3.9) and partially for multi-head attention (Corollary 3.10)** contribute significantly to understanding the training dynamics of these complex models. This builds upon work like **(Vasudeva et al., 2024)** which focuses on the implicit bias and fast convergence of self-attention but through a different lens (convergence to SVM).

*   **Relating Conservation Laws to Network Invariances:** Remark 3.1 and Appendix J.1 discuss the intrinsic connection between the identified conservation laws and network invariances. This links the paper's findings to a broader theme in machine learning where symmetries and invariances of the model and loss function play a crucial role in the learning process.

*   **Examining the Persistence of Conservation Laws Under Discrete Optimization:** Section 5 investigates whether the conservation laws derived in the continuous gradient flow regime persist under **Stochastic Gradient Descent (SGD)**, a widely used discrete optimization method. Proposition 5.1 and the numerical experiments provide insights into the approximate nature of conservation in this practical setting, connecting the theoretical findings to real-world training scenarios. This relates to the vast literature on the behavior and convergence of SGD in deep learning **(e.g., Bach, 2024; Garrigos & Gower, 2023)**.

In summary, the paper's key contributions are deeply intertwined with the existing scientific literature by **extending established concepts of conservation laws to modern deep learning architectures**, building upon prior theoretical results for simpler models, and investigating the practical implications of these laws in the context of widely used optimization techniques and network design principles. The specificity of analyzing ResNets and Transformers, including the impact of residual connections and attention mechanisms, positions this work as a significant advancement in understanding the training dynamics of contemporary neural networks.

**Theoretical Claims:**

I haven't checked all the proofs of the paper, but some of them were inspired by previous literature.

---

> ### Author Rebuttal · Authors · 2025-04-01
>
> We would like to thank the reviewer for his constructive feedback and positive comments.
>
> > **Q1b** challenges in proving the completeness of laws beyond blockwise laws
>
> From a theoretical perspective, the Lie algebra for deeper cases becomes infinite dimensional, which makes the analysis significantly more involved than in block setting. Numerical experiments briefly described in Section 4.4 (see also the supplementary material), however, suggest that combining the laws associated with each block yields a complete set of laws for the deeper case. We will emphasize that theoretical challenge at the end of section 4.
>
> > **Q2b** leveraging newly discovered conservation principles in practice
>
> A first key application of conservation laws is to understand how initialization influences optimization dynamics—for example, by deriving an implicit Riemannian metric parameterized by the initialization. The results of arXiv:1910.05505 (in the case of linear neural networks) show that initializing with certain values of the conserved function can lead to accelerated convergence. Another application lies in the so-called “Edge of Stability” analysis, which, in the linear case, explains how increasing the step size in gradient descent leads to minimizers with small values of the conservation law (“balanced condition”). Extending this analysis beyond the linear case, as initiated in arXiv:2502.20531, is an interesting direction for future research.
>
> We will insist more on these practical applications in the introduction of our paper.
>
> > **Q3b** Could the authors briefly outline their planned approach (...)
>
> Regarding **normalization layers**, they can be treated as a new block and we expect much of the analysis from Appendix M to be adaptable. Then, the main challenge to fully address **multihead attention** is to prove the ``minimimality’’ of the associated reparameterization, which requires more geometric insight than the proof in Appendix F for the case of a single head. Finally, combining our findings on blockwise laws and the non-Euclidean setting of Marcotte et al. 2024 is likely to yield advances regarding **other optimizers**.  Indeed Marcotte et al. 2024 can deal with mirror gradient flow or with natural gradient flow. Analyzing the original **Adam algorithm** is challenging due to the presence of two momentum parameters. However, a simplified version can be readily analyzed using our framework, and we will update the manuscript accordingly. A more detailed analysis of Adam itself remains an interesting direction for future work.
>
> To make the analysis more tractable, we consider a limiting case of Adam where both momentum parameters tend to infinity. In this regime, the algorithm approaches sign gradient descent, and its continuous-time limit is given by the ODE:
> $ \dot \theta (t) = - \text{sgn} (\nabla L_Z(\theta(t))) \ (1).$
> In our framework, this corresponds to considering the linear space of signed gradients:
> $
> W(\theta) := \text{span} \\{ \text{sgn} (\nabla L_Z(\theta)) : Z \\}
> $
> for any parameter $\theta$. An interesting property of $W(\theta)$ is that, due to the quantization introduced by the sign function, $W(\theta)$ is locally constant around generic values of $\theta$. This leads to the identity $\text{Lie}(W)(\theta) = W(\theta)$. By Theorem 2.11, this implies that in a neighborhood of $\theta$, there are exactly
> $
> D - \text{dim} (\text{Lie}(W)(\theta)) = D - \text{dim} (W(\theta))
> $
> independent conservation laws. In particular, if $\text{dim} (W(\theta)) = D$, then no conservation law exists in that region.
>
> In the simple case of a $2$-layer linear neural network $g((U, V), x) = U V^\top x$, we easily show (direct with Theorem 3.4) that $W(\theta) = \text{span} \\{ \text{sgn} ( \partial \phi (\theta)^\top w): w \\}$ where $\phi(\theta) = U V^\top$,  which is easily computationaly tractable. In this code https://anonymous.4open.science/r/Numerical_experiments-2730/Adam.ipynb, we test it on different dimension settings and we obtain that, except in the case where $U$ and $V$ are both scalars, we have $\text{dim} W(\theta) = D$, so that there is no conservation law. In the special case where $U$ and $V$ are both scalars, we obtain that $\text{dim} W(\theta) = 1$, and thus there is only $2-1= 1$ conservation laws: indeed, $|U|- |V|$ is conserved.
>
> While we only provide numerical explanations in the $2$-layer linear case, we expect that it holds for all other architectures. This indicates a radically different geometric behavior of the Adam optimizer. It is worth recalling that adding a momentum term also leads to a loss of the total number of conservation laws (in most settings (cf Marcotte et al. 2024), there is no conservation law at all when considering a momentum term). A better understanding of these phenomena is an interesting future direction. We will mention this discussion at the end of our paper.

---

### Official Review · Reviewer_zuVu · 2025-03-13

**Overall Recommendation:** 2

**Summary:**

This paper studies conservation laws for gradient flow on a variety of modern neural network architectures, including ResNets and Transformers. The paper provides a characterization of all possible conservation laws for both shallow and deep architectures. Moreover, the paper quantifies the extent to which SGD satisfies an approximate conservation law.

**Claims And Evidence:**

For the most part, the main claims of the paper (a characterization of all possible conservation laws) are well supported.

On a more minor note, the abstract claims that "some deeper networks can [have] more conservation laws than intuitively expected." I do not see where this claim is proven in the paper.

**Essential References Not Discussed:**

N/A

**Experimental Designs Or Analyses:**

The experimental validation in Figure 1 appears to be sound.

**Methods And Evaluation Criteria:**

Yes, the methods in this paper make sense.

**Other Comments Or Suggestions:**

Typos:
- abstract: “can more” → “can have more”
- abstract: “the introduction the” → “the introduction of the”
- line 102, left column: “that “overalapping” two residual blocks”

**Other Strengths And Weaknesses:**

- One weakness of this paper is that the conservation law $h$ must hold for "any initialization and any dataset." In particular, $h$ cannot depend on the training dataset. This seems to be an overly strict condition. For example, in overparameterized linear regression (i.e L(w) = \|y - Xw\|^2), the quantity $P_{X^TX}^\perp w$, which does depend on the dataset, is conserved, and this is useful for characterizing the implicit bias of gradient descent.

- Next, while the paper builds significant technical machinery, the specific conservation laws for Transformers derived in Corollary 3.9 and 3.10 are already known in the matrix factorization literature; indeed, given any model with a parameter $W$ parameterized as $W = Q^TK$, it is well known that the quantity $QQ^T - KK^T$ will be conserved throughout the gradient flow trajectory. It is thus not clear to me what additional insights this paper provides.

- On the clarity front, I find Section 2 to be quite technically dense and hard to follow. The paper would improve with more exposition providing intuition for why each of these technical results are needed.

**Questions For Authors:**

- Can the authors please comment on my concerns in the "Strengths and Weaknesses" section?
- Many of the results in Section 2 and 3 seem to be restated results from (Marcotte et al., 2023, 2024). It is thus not clear to a reader unfamiliar with these prior works what the additional contribution of this current paper is. Could the authors please clarify in which ways their paper generalizes the results of these prior works?

**Relation To Broader Scientific Literature:**

This paper builds on prior works establishing a "balancedness" condition for ReLU and linear neural networks (Du et al., 2018; Arora et al., 2019), along with previous work on conservation laws of gradient flow (Marcotte et al., 2023, 2024).

**Theoretical Claims:**

I skimmed the proof of the main claims, and they appear to be sound.

---

> ### Author Rebuttal · Authors · 2025-04-01
>
> We thank the reviewer for his constructive feedback.
>
> > **Q1a** the abstract claims (...)
>
> Thank you for pointing this out, you’re right. Since this is not the focus of our paper as we don't have a detailed theoretical analysis of this phenomenon, we will remove this sentence from the abstract. Instead, we will add the following simple example at the end of section 4.4.
>
> Example: Consider a ReLU neural network with two-residual blocks, $g((u, v, s, t), x) = x + u \sigma (v x) + s  \sigma(t (x + u\sigma( v x))$ with $(u, v, s, t) \in \Omega \subseteq \mathbb{R}^4$ and $x \in \mathbb{R}$. While there are *two*  "block" conservation laws: $u^2 - v^2$ and $s^2 - t^2$, we exhibit a domain $\Omega$ where there are *three* conservation laws. For this, consider $\Omega$ the set of all parameters such that $sgn(t) = sgn(v) = sgn(u).$ Then, for any $x$:
>
> - either $vx, sx \leq 0$ and then $g(\theta, x) = x$ and thus $\nabla_{\theta} g = 0$;
> - or $vx, sx \geq 0$ and then $g(\theta, x) = x + uv x + stx +stuv x$ as $vx, sx, tu \geq 0$, and then $ 1/x \nabla_{\theta}  g=: \chi$ is a vector field that does not depend on $x$.
>
> Thus the space $W_{\theta}^{g, l} = \mathbb{R} \chi(\theta)$ (cf Proposition 2.3) is made by only one non-null vector field. Its associated Lie algebra is thus itself, and by Theorem 2.11, there are exactly 3 conservation laws as claimed.
>
> This phenomenon seems however restricted to the scalar case, as we empirically show in Section 4.4.
>
> > **Q2a** h cannot depend on the dataset (...)
>
> Thank you for this insightful comment. We view the dataset-independence condition as a deliberate design choice rather than a weakness, as it yields general insights on the implicit bias stemming solely from the architecture and optimization scheme.
>
> While we agree that introducing dataset dependency could enrich the analysis, it also opens the Pandora’s box of how to constrain this dependency to ensure the problem remains nontrivial and the conservation law remains informative. For instance, allowing unrestricted dataset dependency would make the problem on the number of independent conservation laws trivial, as we now explain.
>
> Indeed, the Lie algebra framework of Marcotte et al. can also be applied to vector fields associated with specific datasets (see Definition A.3). For a fixed dataset, the associated functional space (Definition A.4) becomes one-dimensional, and its generated Lie algebra is simply itself. By Frobenius' theorem, this yields exactly D-1 conserved functions which typically *exhibit non-trivial dataset dependencies*. Your example of overparameterized linear regression fits perfectly within this framework.
>
> Thus, finding the right constraints to keep the problem well-posed and meaningful becomes challenging. It seems feasible to adapt the algorithm that builds polynomial conservation laws (section 2.5 of Marcotte et al. 2023) to settings with fixed datasets by identifying polynomial conservation laws that are *also polynomial in the given dataset parameters* using formal calculus.
>
> We appreciate this suggestion and will add a discussion in the conclusion highlighting this as a promising direction for future research.
>
> > **Q3a** (...) conservation laws for Transformers (...)
>
> Indeed, this is a direct consequence of invariances, however, since we could not find any reference clearly stating this phenomenon for transformers, we believe it is worth highlighting. More importantly, our main contribution is, however, the nontrivial proof that *there is no other conservation law that only depends on the parameters of such attention blocks*. In particular, while finding the reparameterization $\phi(\theta)$ is relatively easy, proving its "minimality" (cf Appendix F) is much more involved. We will reword to insist on these facts.
>
> > **Q4a** Many of the results in Section 2 and 3 seem to be restated results (...)
>
> All results in section 3 are new except the one recalled in section 3.2 “ReLU and linear networks: known results”. In Section 2, novelties are notably Theorem 2.1, Corollary 2.8, Propositions 2.9 and 2.12. While Theorem 2.11 is not new, we provide a new simplified proof. We will clarify this.
>
> > **Q5a** in which ways does their paper generalize the results of these prior works?
>
> The only result that can be seen as a generalization of previous work (all other results are entirely new) is Theorem 3.6, as detailed in Remark 3.7. In particular, the function $f$ used to factorize the neural network $g_{conv}$ defined in equation (13) $g_{conv} = \psi \circ f$ is the same as for the 2-layer ReLU neural network $g$ defined in equation (12): one has $g = \phi \circ f$. Thus, to show Theorem 3.6, you can first directly use the result of Theorem 3.4 as it only depends on $f$ (see Lemma E.2). However, to complete the proof of Theorem 3.6, you also need to compute the generated Lie algebra associated to $\psi$, which is pretty involved and differs from Lie algebra computations associated to $\phi$.

---

### Decision · Program_Chairs · 2025-05-01

**Decision:**

Accept (oral)

**Comment:**

The paper extends an existing framework for studying conservation laws to key components of modern neural architectures, focusing in particular on convolutional ResNets and Transformers

The reviewers commended the paper for its thorough derivation of conservation laws in these settings, addressing an important gap in the literature, as well as for its detailed examination of how these theoretical principles manifest in practice, accounting for discrete effects and supported by empirical validation.